# High intrinsic phase stability of ultrathin 2M WS$_2$

Xiangye Liu[1,2,6], Pingting Zhang[1,2,6], Shiyao Wang[3,6], Yuqiang Fang[4,6], Penghui Wu[1,2], Yue Xiang[1,2], Jipeng Chen[1,2], Chendong Zhao[4], Xian Zhang [5], Wei Zhao[4], Junjie Wang[3], Fuqiang Huang [4] & Cao Guan [1,2] ✉

Metallic 2M or 1T′-phase transition metal dichalcogenides (TMDs) attract increasing interests owing to their fascinating physicochemical properties, such as superconductivity, optical nonlinearity, and enhanced electrochemical activity. However, these TMDs are metastable and tend to transform to the thermodynamically stable 2H phase. In this study, through systematic investigation and theoretical simulation of phase change of 2M WS$_2$, we demonstrate that ultrathin 2M WS$_2$ has significantly higher intrinsic thermal stabilities than the bulk counterparts. The 2M-to-2H phase transition temperature increases from 120 °C to 210 °C in the air as thickness of WS$_2$ is reduced from bulk to bilayer. Monolayered 1T′ WS$_2$ can withstand temperatures up to 350 °C in the air before being oxidized, and up to 450 °C in argon atmosphere before transforming to 1H phase. The higher stability of thinner 2M WS$_2$ is attributed to stiffened intralayer bonds, enhanced thermal conductivity and higher average barrier per layer during the layer(s)-by-layer(s) phase transition process. The observed high intrinsic phase stability can expand the practical applications of ultrathin 2M TMDs.

Phase engineering of group-VI transition metal dichalcogenides (TMDs) such as MoS$_2$ and WS$_2$ is important for acquiring novel physical and chemical properties[1–6]. Depending on coordination modes between the transition metal and chalcogen atoms, these TMDs present in either trigonal prismatic coordinated semiconductive phases (2H and 3R) or octahedral coordinated metallic phases (2M, T$_d$, 1T, 1T′, etc.)[1–3,6]. Bulk 2M and the monolayered (ML) 1T′ WS$_2$ have shown plenty of unique appealing properties, such as superconductivity[7–10], Weyl semimetal states[11,12], optical nonlinearity[13,14] and enhanced electrochemical activities[15–19]. However, the 1T′ TMDs are metastable and tend to transform to the thermodynamically stable 2H phase[1,3,4], hence their practical applications are significantly limited[1,3]. Although various strategies such as electron doping[20,21], strain effect[22,23], and

heterostructural interaction[24,25] can be adopted to stabilize the metallic phase TMDs, their intrinsic stabilities under practical operation conditions (e.g., temperature and atmosphere) remain unclear.

As van der Waals (vdW)-bonded layered materials, physicochemical properties of TMDs can dramatically change when their structures evolve from three-dimensional bulk to ultrathin two-dimensional (2D) regime. For example, band gaps of 2H MoS$_2$ and WS$_2$ turn from indirect to direct after thickness is reduced to monolayer[26]. Dielectric screening in gapped 2D crystals significantly reduces with decrease of layer number, leading to greatly enhanced Coulomb interactions[27]. The high Coulombic interactions have resulted in tightly bonded exitons and trions in ML MoS$_2$[27], as well as anormal thickness-dependent Raman shifts[28]. Atomically thin TMDs

[1]Institute of Flexible Electronics, Northwestern Polytechnical University, Xi'an 710072, China. [2]Key Laboratory of Flexible Electronics of Zhejiang Province, Ningbo Institute of Northwestern Polytechnical University, 218 Qingyi Road, Ningbo 315103, China. [3]State Key Laboratory of Solidification Processing, Northwestern Polytechnical University, Xi'an 710072 Shaanxi, China. [4]State Key Laboratory of High-Performance Ceramics and Superfine Microstructure, Shanghai Institute of Ceramics, Chinese Academy of Sciences Shanghai, Shanghai 200050, China. [5]Qian Xuesen Laboratory of Space Technology, China Academy of Space Technology, Beijing 100094, China. [6]These authors contributed equally: Xiangye Liu, Pingting Zhang, Shiyao Wang, Yuqiang Fang. ✉e-mail: iamcguan@nwpu.edu.cn

exhibit much higher electrocatalytic activities than bulk matrixes, because of "self-gating effect" induced high carrier densities[29,30]. The in-plane inversion symmetry of bulk TMDs is broken after they are reduced to ML, which produces out-of-plane spin polarization depending on the valley (K or −K point) in momentum space[2]. Thermal conductivity is found to decrease with increase of layer number for thin 2H $MoS_2$ and $T_d$ $WTe_2$[31,32], while the opposite trend is seen for thin indium selenides[33,34]. In a word, the layer number plays critical roles in both the electronic and phonon properties of 2D TMDs. As 2M to 2H phase transition of $WS_2$ (or 1T′ to 1H for ML $WS_2$) involves changes in energy band structures of both electron and phonon, the layer number could greatly affect the stability of 2M $WS_2$. However, the relation between layer number and stability of 2M $WS_2$ is barely known to the best of our knowledge.

Here, we investigate the thickness-dependent intrinsic phase stability of mechanically exfoliated 2M $WS_2$ and find that thinner samples have higher thermal stabilities. 2M to 2H phase transition temperature increases from 120 °C to 210 °C in the air as thickness of $WS_2$ is reduced from bulk to bilayer (2L). ML $WS_2$ can maintain 1T′ structure in the air and argon (Ar) atmosphere until temperature

reaches 350 °C and 450 °C, respectively, which are about two and three times higher than phase transition temperatures of bulk 2M $WS_2$. Raman spectroscopy reveals thinner 2M $WS_2$ has more stiffened intralayer W–W and W–S bonds and higher thermal conductance, which enables thinner samples with higher resistance to lattice deformation and higher efficiency heat dissipation under elevated temperature. Theoretical simulation and calculation further confirm thinner $WS_2$ has a higher average energy barrier per layer during the layer(s)-by-layer(s) phase transition process.

## Results

### Thickness-dependent phase stability of 2M $WS_2$

2M $WS_2$ is synthesized through solid-state chemical method as reported previously[7]. The as-synthesized 2M $WS_2$ shows distinctive Raman peaks at 110.9, 117.1, 177.2, 241.7, 268.6, 316.3, and 406.5 cm$^{-1}$ (black line in Fig. 1d), the same as previously reported results[3,7]. Rietveld refinement based on powder X-ray diffraction (XRD) (Supplementary Fig. 1 and Supplementary Table 1) reveals the synthesized $WS_2$ has monoclinic structure in $C_{2/m}$ space group with cell parameters $a$ = 12.8417 Å, $b$ = 3.2177 Å, $c$ = 5.6912 Å and $\beta$ = 112.8368°, which is in

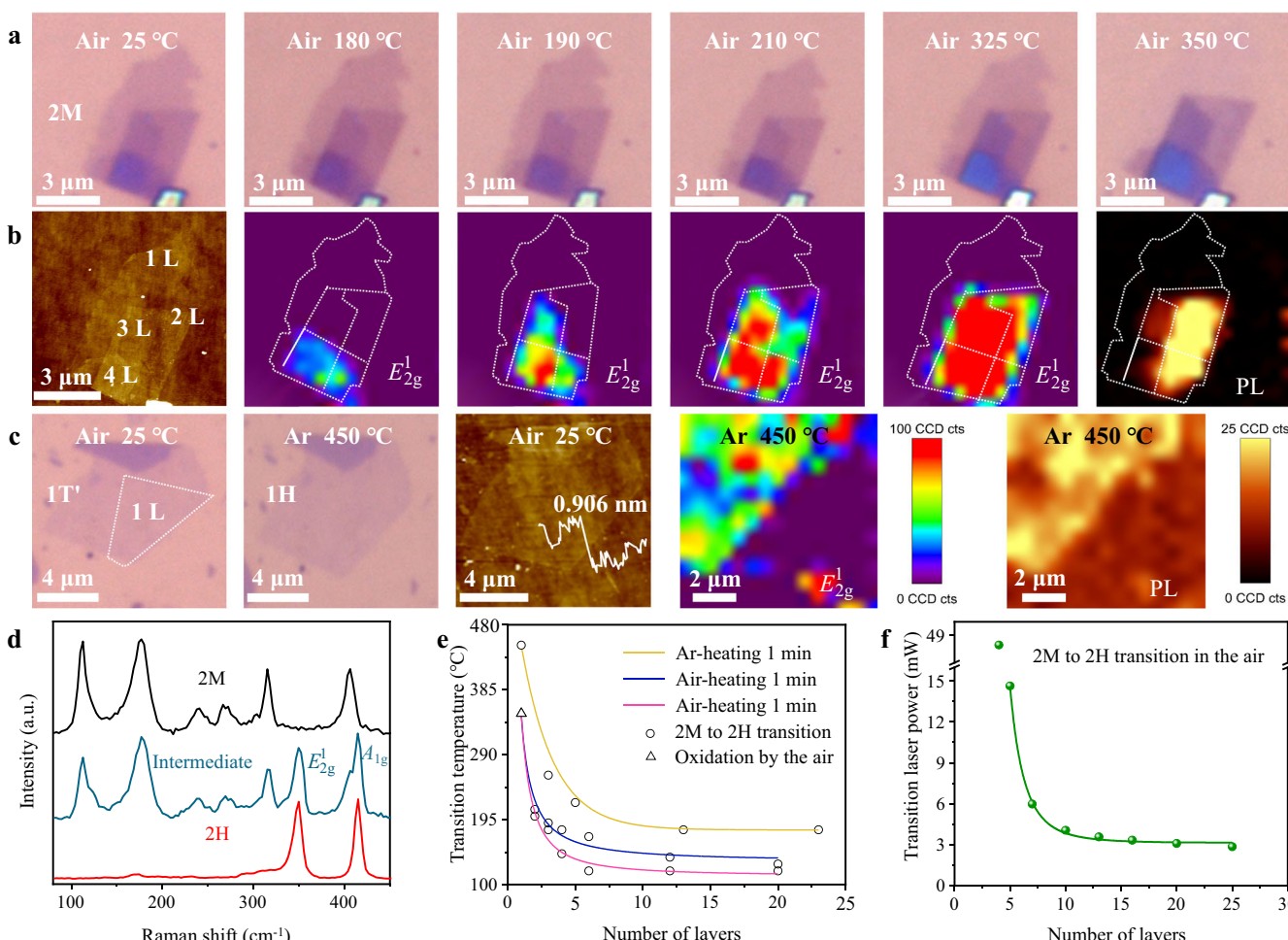

**Fig. 1 | Thickness-dependent phase stability of 2M $WS_2$ on Si/SiO$_2$ substrate.**
**a** Optical and **b** atomic force microscope (AFM) images and mappings of $E^1_{2g}$ Raman mode and photoluminescence (PL) intensities of a piece of exfoliated 2M $WS_2$ flake with different thicknesses areas at different temperatures. AFM image is measured at room temperature, and the corresponding height profile is shown in Supplementary Fig. 3f. **c** Optical images, AFM image, and mappings of $E^1_{2g}$ Raman mode and PL intensities of a piece of exfoliated monolayered (ML) 1T′ $WS_2$ flake at room temperature in the air and after heated at 450 °C in Ar atmosphere. Mappings measured area is outlined by dotted lines in the optical image. All the scale bars in (**a**–**c**) correspond to 2 μm. **d** Raman spectra of a 2M $WS_2$ and an intermediate phase and 2H $WS_2$ that were obtained by heating multilayered 2M $WS_2$ at 120 °C for 5 min and at 250 °C for 20 min, respectively. **e** 2M to 2H (or 1T′ to 1H) phase transition temperatures (circle points) and air oxidation temperature (triangle point) as functions of $WS_2$ layer thickness, measured in the air or Ar atmosphere with temperature elevated by 5 °C and held for 1 min or 15 min in each step. Solid lines are guides to the eye. **f** The power of laser required to activate 2M to 2H phase transition as a function of $WS_2$ layer thickness measured in the air. A solid line is a guide to the eye.

excellent agreement with the previously reported 2M WS$_2$ structure[3,7]. X-ray photoelectron spectroscopy (XPS) (Supplementary Fig. 2) further evidences the synthesized sample is composed of only W and S elements. All these results confirm the high purity of the synthesized 2M WS$_2$. Once 2M to 2H phase change is triggered by heat or oxidation, two additional Raman peaks appear at around 350.2 and 420.0 cm$^{-1}$ (blue line in Fig. 1d) that are attributed to the $A_{1g}$ and $E_{2g}^1$ vibration modes of 2H WS$_2$[3]. After the WS$_2$ flake is completely converted to 2H phase, all the 2M characteristic vibration modes disappear and only the $A_{1g}$ and $E_{2g}^1$ peaks can be seen (red line in Fig. 1d). Accordingly, the intensity of $E_{2g}^1$ mode can be used as an indicator for the extent of phase change within a 2M WS$_2$ flake.

Here, we define phase transition temperature as the one at which the 2M or 1T′ WS$_2$ starts to lose phase purity, i.e., the $E_{2g}^1$ Raman mode emerges, or the WS$_2$ sample begins to be oxidized by the air. To investigate the thickness-dependent phase stability, 2M WS$_2$ flakes are mechanically exfoliated into thin layers. Figure 1a, b shows optical and atomic force microscope (AFM) images of a piece of exfoliated 2M WS$_2$ flake with thickness varying from ML to 4L. This sample was treated with stepped heating in the air with temperature elevated by 5 °C and held for 1 min in each stage, during which optical images and mappings of $E_{2g}^1$ Raman mode and photoluminescence (PL) intensities are acquired to address the location of 2M to 2H (or 1T′ to 1H) phase change, as shown in Fig. 1a, b. It is found that the $E_{2g}^1$ mode can be observed at the 4L area when temperature goes up to 180 °C, whereas it does not appear at the 3L and 2L areas until the temperature goes to 190 °C, 210 °C, respectively. With temperature increasing from 180 °C to 325 °C, the $E_{2g}^1$ mode intensities of 3–4L areas keep increasing and the transparency is obviously decreasing, while the 2L area is too thin to tell transparency change. It is known that 2H WS$_2$ has higher refractive index and lower transparency than 2M WS$_2$[35]. PL peaks (Supplementary Fig. 3a–c) at ~640 nm emerge in 2–4L areas as temperature reaches 325 °C, further confirming the 2M to 2H phase transitions[36]. The PL intensities are significantly enhanced after heating to 350 °C, as shown in Supplementary Fig. 3a–c, consistent with the increase of $E_{2g}^1$ mode intensity at elevated temperature. These results indicate 2M to 2H phase transition is a slow process, and the extent of phase transition augments at elevated temperatures or after longer time of heating. PL intensity of WS$_2$ is sensitive to thickness[26], and the distribution of PL intensity (Fig. 1b) agrees to the AFM defined areas of various thicknesses from 2L to 4L. Comparatively, the ML WS$_2$ area preserves the typical 1T′ Raman pattern without emergence of $E_{2g}^1$ mode or PL until temperature reaches 350 °C (Fig. 1b and Supplementary Fig. 3d, e), evidencing the highest thermal stability among the four WS$_2$ areas. In agreement with this result, chemically exfoliated ML 1T′ WS$_2$ is also demonstrated to have good air-stability at room temperature[37,38]. Heated at 350 °C in the air, Raman signal of the ML 1T′ WS$_2$ fades away and the ML 1T′ WS$_2$ flake disappears, seen from Fig. 1a, which can be attributed to decomposition due to air oxidation. In our experiments of heating in the air, we never observe oxidation of 2L or multilayered 2M WS$_2$ flakes before 2M to 2H phase transition, while we never observe 1T′ to 1H phase transition on a ML 1T′ WS$_2$ flake before it was oxidized and decomposed. This means 1T′ to 1H phase transition temperature of ML 1T′ WS$_2$ is higher than the air oxidation and decomposition temperature. After moving to an inert Ar atmosphere, 1T′ to 1H phase transition temperature of ML WS$_2$ (Fig. 1c) is measured to be as high as 450 °C, confirmed by the reduced transparency and emergences of $E_{2g}^1$ mode and PL (Fig. 1c). These results strongly indicate thinner 2M WS$_2$ has higher phase stability.

To further confirm the thickness-dependent stability of 2M WS$_2$, we have prepared exfoliated samples with thickness covering a wide range (from 23L to ML) and systematically investigated their tolerance to heat in the air or Ar atmosphere. The optical, AFM images and Raman spectra of these samples at representative conditions can be found in Supplementary Figs. 4–6. Figure 1e summarizes phase transition temperatures of the investigated 2M WS$_2$ samples during stepped heating programs. When heating in the air with temperature held for 1 min in each stage as aforementioned, the measured phase transition temperature continuously increases from 120 °C to 350 °C as WS$_2$ is thinned from 20L to ML (blue line in Fig. 1e). When temperature is held for 15 min in each stage, the phase transition temperatures of thicker samples (≥6L) converge to a same level of 120 °C, while the phase transition temperatures of thinner samples (1–5L) still maintain the thickness dependence (red line in Fig. 1e). Regardless of temperature holding time, ultrathin samples (ML and 2L) exhibit significantly high phase transition temperatures that are over 200 °C. When heating in Ar atmosphere, the same trend of phase transition temperature as function of sample thickness is also observed, except all 2M (or 1T′) WS$_2$ samples show increased phase transition temperatures compared with heating in the air (yellow line in Fig. 1e). Previous X-ray absorption fine structure measurements have confirmed enrichment of electrons on 2M or 1T′ WS$_2$ surface[3,6]. It has also been demonstrated that donation of electrons helps to stabilize 1T′ TMDs, while extraction of electrons promotes 1T′ to 1H phase transition[20,21,39–41]. Thus, higher 2M to 2H (or 1T′ to 1H) phase transition temperature measured in the inert atmosphere can be attributed to getting rid of electron acceptors, such as O$_2$ and H$_2$O molecules in the air. High-resolution XPS (Supplementary Fig. 7) shows that W 4f, S 2p and valance band edge spectra of 2M WS$_2$ shift to higher energies after 2M to 2H phase transition is activated by heating in the air, and higher extent of phase transition leads to larger range of shifting. Meanwhile, no peak attributed to W–O or S–O chemical bond can be found in the W 4 f and S 2p spectra and O 1s spectra (probably contributed by adsorbed or intercalated H$_2$O and O$_2$ molecules) barely change during the 2M to 2H phase transition. These results indicate air speeds up 2M to 2H phase transition during heating processes by extracting electrons from 2M WS$_2$ without chemically oxidizing it. The thinner 2M WS$_2$ can also withstand higher intensity incident laser during Raman measurements (Fig. 1f and Supplementary Fig. 8). From 25L to 5L WS$_2$, the intensity of laser required to activate the 2M to 2H phase transition increases from 2.8 mW to 14.6 mW. A 48.3 mW laser, which is the maximum output of the used instrument, can turn a 4L WS$_2$ to 2H phase, but is not able to activate the phase transition for 1–3L 2M WS$_2$ samples.

## 2M to 2H phase transition mechanism

To understand how WS$_2$ transforms from 2M to 2H phase, we closely investigate crystal structures of the two phases. Each layer of 2M (or a layer of 1T′) WS$_2$ contains distorted [WS$_6$]$^{8-}$ octahedrons sharing edges along $bc$ plane, and W atoms form W−W zigzag chains along $b$ direction, resulting in nonuniform distances between W-atom lines, $d_1 = 2.28$ Å and $d_2 = 3.43$ Å (Fig. 2a and Supplementary Fig. 9a)[7]. Viewing from the $b$ direction, W and S atoms reside in six different planes and stack in an A/A′-B/B′-C/C′ model (Fig. 2a)[7]. Every two neighbored S atoms located in different planes (A and C or A′ and C′) share two W atoms (Supplementary Fig. 9a). Comparatively, each layer of 2H (or a layer of 1H) WS$_2$ is composed of standard [WS$_6$]$^{8-}$ trigonal prisms with uniform W-atom line distance of $d_3 = 2.70$ Å (Fig. 2b and Supplementary Fig. 9b)[42]. Viewing from the $b$ direction, W and S atoms stack in a simple A-B-A sandwiched model (Fig. 2b), and every two neighbored S atoms located in different planes share three W atoms (Supplementary Fig. 9b)[42].

Because of the structural differences between 2M and 2H WS$_2$, the 2M to 2H phase change involves reforming of intralayer bond network with displacements of W and S atoms and changing of interlayer spacing. As shown in Fig. 2c, during 2M to 2H phase change of WS$_2$, the bonded W–W chains break up and W atoms in these chains gradually separate along $c$ direction. The gliding of W atoms finally gives uniform W-atom line distance, approaching that in 2H WS$_2$. In the meantime,

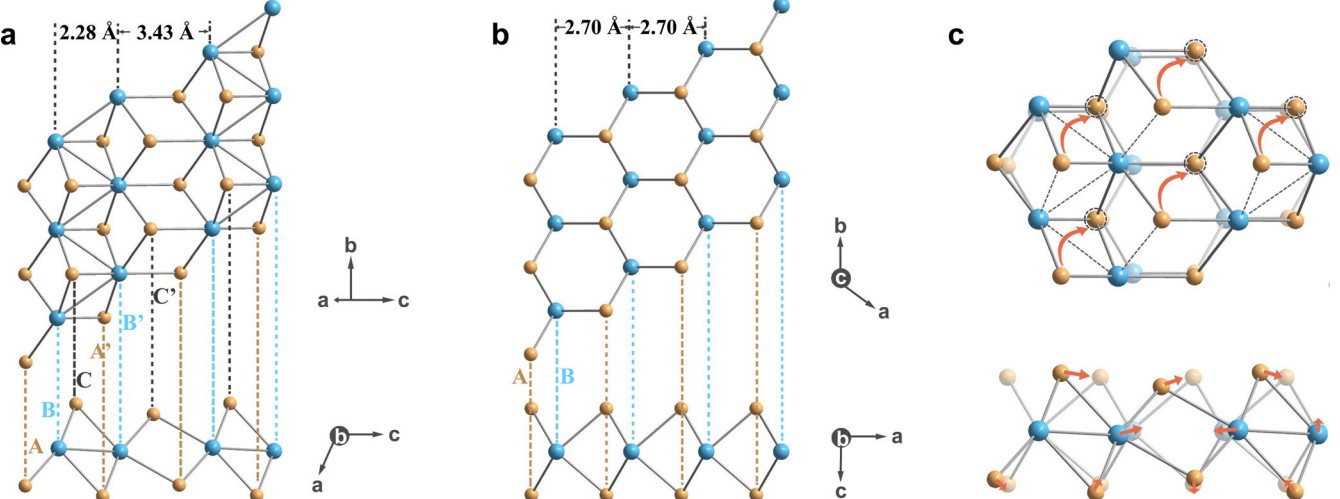

**Fig. 2 | 2M to 2H phase transition mechanism of WS₂. a** Crystal structures of 1T′ WS₂, where W and S atoms coordinate in an octahedral configuration and stack in an A/A′-B/B′-C/C′ mode. **b** Crystal structures of 1H WS₂, where W and S atoms coordinate in a trigonal prismatic configuration and stack in an A-B-A mode. **c** Top view (upper) and side view (lower) of crystal lattices when overlapping of a layer of

1T′ WS₂ on top of a layer of 1H WS₂. Color code: blue and orange spheres represent W and S, respectively. The broken W–W and W–S bonds are labeled by dash lines, and the directions of W and S atoms displacements are indicated by orange arrows. Each S atom that dissociates from a W atom after breaking the W–S bond moves to a new position (marked by a dotted line circle) and forms a new W–S bond.

S atoms residing in C and C′ planes glide along the *bc* direction to lay on top of adjacent S atoms in A and A′ planes, respectively. Subsequently, each S atom dissociates from the initially bonded W atom and forms a new covalent bond with another adjacent W atom, resulting in alteration of W−S coordination from octahedron to trigonal prism, as shown in Fig. 2c. Along with out-of-plane displacements of W and S atoms and adjusting of W−S bond lengths and W−S−W angles, each $[WS_6]^{8-}$ layer changes from the six-plane stack (2M) to the three-plane sandwiched stack (2H) (Fig. 2c). Since 2H WS₂ has smaller average W–S bond length, W−S−W bond angle and W-atom line distance but larger interlayer spacing than the 2M WS₂ (Supplementary Fig. 9), 2M to 2H phase transition experiences slight in-plane contraction and vertical expansion.

The slowness of 2M to 2H phase transition of WS₂ allows us to trace this process by investigating structures of intermediate phases after varied extents of 2M to 2H phase change. Intermediate phases WS₂ can be obtained by heating the 2M WS₂ above phase transition temperature within controlled time. As shown in Supplementary Figs. 4–6, all intermediate phases WS₂ flakes show superpositions of 2M and 2H characteristic Raman modes, indicating the intermediate phases of WS₂ are probably 1T′/1H heterostructures that can involve in-plane heterostructured areas and hybrid stacked layers. Figure 3a–c show high-resolution transmission electron microscopy (HRTEM) images of W-atom configurations in 2M and an intermediate phase. Zigzag chains of W atoms are observed in Fig. 3a, in consistency with the reported structure of 2M WS₂[3]. Area [1] of the intermediate phase WS₂ (Fig. 3b) reveals W-atom lines distances ranging from 0.26 to 0.31 nm, which are in between $d_1$ (0.23 nm) and $d_2$ (0.34 nm) of 2M WS₂ (Fig. 3a). Fourier transform of Area [1] displays irregular quadrilaterals (inset of Fig. 3b) that are distorted from the rectangular pattern of 2M WS₂ (inset of Fig. 3a). Area [2] of the intermediate phase WS₂ has a W-atom lines distance of 0.27 nm (Fig. 3c) and hexagonal pattern Fourier transform (inset of Fig. 3c), which are the same as 2H WS₂[42]. These results demonstrate that an intermediate phase WS₂ has an in-plane heterostructure of distorted 1T′ and 1H, which is formed due to gliding of W atoms.

To investigate layer stacking configurations of intermediate phases WS₂, AFM images are taken for different phases WS₂. It is observed that the thickness of a WS₂ flake (Fig. 3d–f) increases from 25.461 nm to 25.967 nm and 27.074 nm as it changes from 2M to an intermediate

and 2H phase, revealing the continuous change of interlayer spacing. Moreover, there is slight sliding at edge of the WS₂ flake from 2M to the intermediate and 2H phase (Fig. 3d–f and Supplementary Fig. 12), indicating generation of interlayer dislocation during the phase change. Since transforming from a 1T′ to 1H layer results in in-plane lattice contraction, the interlayer dislocation can be attributed to non-concerted 1T′ to 1H phase transition among different layers and the followed reconfiguration of layer stacking. XRD has been carried out on bulk 2M WS₂ powder during heating at 130 °C. With prolongation of heating time, XRD peaks attributed to 2M (200) diffractions are weakening and broadening, while the ones attributed to 2H (002) diffractions are intensifying and sharpening, and all these peaks are shifting to lower degrees but locating in between that of standard 2M (200) and 2H (002) diffractions, as shown in Fig. 3g, h. These results indicate ordered stacks of 1T′ and 1H layers are broken and built, respectively, with increase of phase change extent. An intermediate phase WS₂ has hybrid low-order stacks of 1T′ and 1H layers with average interlayer spacing in between that of pure 2M and 2H phase. In agreement with this result, Raman spectra (Supplementary Fig. 10) of intermediate phases WS₂ manifests that $A_{1g}$ mode (ascribed to out-of-plane vibration of 1H layer stacks) locates at lower wave number compared to 2H phase and keeps to blue-shift as the extent of 2M to 2H phase change increases. High-index XRD diffractions including 2M (11$\bar{1}$) (002) (31$\bar{1}$) (601) (311) are also disappearing with (11$\bar{1}$) diffraction shifting to higher degrees as heating time increases, as shown in Supplementary Fig. 11. This can be attributed to in-plane gliding and out-of-plane displacements of W and S atoms during the 2M to 2H phase transition process. The above XRD, Raman and AFM results hint that 2M to 2H phase transition of multilayered WS₂ probably takes place layer(s) by layer(s). To further confirm this mechanism, we tested Raman spectra on an intermediate phase multilayered WS₂ flake before and after punching with a tungsten probe. As shown in Fig. 3i, the exposed inner layers after punching (areas marked by red squares) exhibit significantly higher intensities of 2H $A_{1g}$ and $E_{2g}^1$ modes than top layers of the WS₂ flake, which strongly evidences non-concerted phase transition among different layers.

## Thickness-dependent thermal properties of 2M WS₂

As the 2M to 2H phase change of WS₂ occurs along with dissociations of W−W bonds and reconstructions of W−S bonds, strengths of these

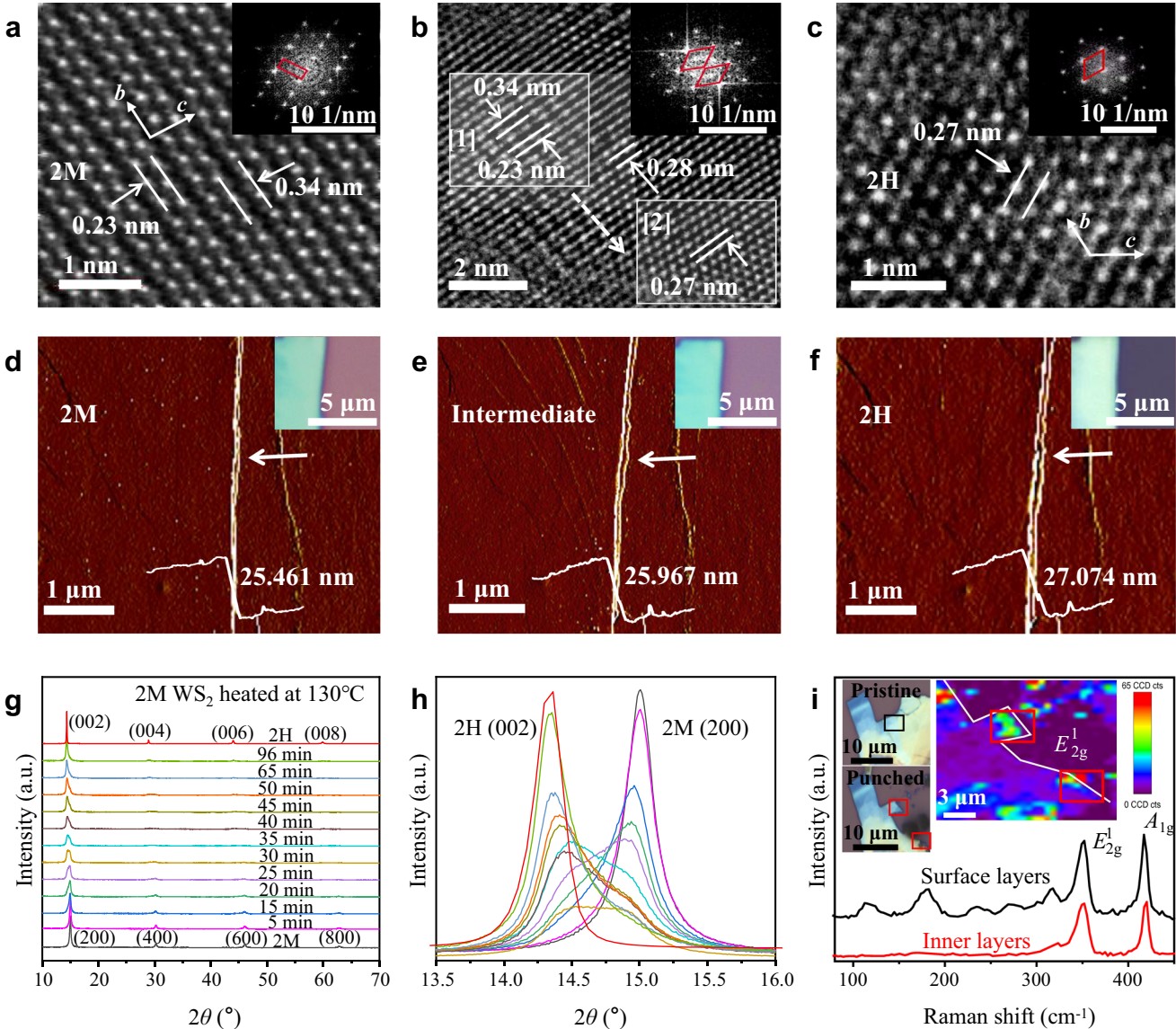

**Fig. 3 | Structural characterization of WS₂ during 2M to 2H phase transition.**
High-resolution transmission electron microscopy (HRTEM) images and the corresponding Fourier transforms of (**a**) 2M and (**b**) an intermediate phase WS₂. **c** A zoomed-in view and Fourier transform of area [2] in (**b**). **d** Powder X-ray diffraction (XRD) patterns of 2M, intermediate phases and 2H WS₂. 2H and intermediate phases WS₂ were obtained by heating 2M WS₂ in the air at 250 °C for 20 min and at 130 °C for various times, respectively. **e** A zoomed-in view of the XRD patterns in the range of 13.5°~16.0°. AFM images with height profiles and optical images (insets at top-right corners) of a piece of multilayered WS₂ flake in (**f**) 2M, (**g**) an intermediate and (**h**) 2H phases. Edge of the WS₂ flake is denoted by white arrow, where interlayer sliding can be seen. **i** Raman spectra of an intermediate phase multilayered WS₂ flake measured from the surface layers (black line) and the exposed inner layers (red lines) after punching with a tungsten probe. Insets show the corresponding optical images of the pristine WS₂ flake and the one after punching (left) and Raman $E_{2g}^1$ mode intensity mapping of the probe-punched WS₂ flake (right). Raman-measured areas are defined by squares.

intralayer chemical bonds play a critical role in phase stability of 2M WS₂. Characteristic vibration frequency ($\sigma$) of a chemical bond is in proportion to square root of the bond force constant ($k$), according to $\sigma = (N_A^{0.5}/2\pi)(k/M)^{0.5}$, where $N_A$ is Avogadro's constant, and $M$ is average atomic mass. Therefore, frequencies of Raman vibration modes of 2M WS₂ manifest strengths of the intralayer chemical bonds. Figure 4a shows Raman spectra of 2M WS₂ flakes with various thicknesses. All samples exhibit the same Raman pattern, indicating the exfoliated few-layered 2M WS₂ flakes maintain the same distorted octahedral coordination as that of bulk one. Previous theoretical calculation has demonstrated 2M WS₂ has nine active Raman vibration modes, including one $B_g$ mode, one $B_u$ mode and seven $A_g$ modes ($A_g^6$ at 327.5 cm⁻¹ is not detectable in our results), as denoted in Fig. 4a[3]. Interestingly, $B_g$, $A_g^1$ and $B_u$ modes are found to distinctly stiffen (blue

shift) with decrease of thickness. As shown in Fig. 4b, from 6L to ML 1T' WS₂, $B_g$ mode shifts from 110.9 to 121.1 cm⁻¹, with $A_g^1$ mode shifting from 117.1 to 125.5 cm⁻¹ and $B_u$ mode shifting from 177.2 to 184.7 cm⁻¹. For flakes of 6 or more layers, the frequencies of these modes converge to the bulk values. Other Raman modes of ML 1T' WS₂ were also reported to blue shift compared with that of bulk 2M one[3], but we are not able to see these shifts in our results, probably because the shifts are too small to be observed. As shown in Fig. 4c, $B_g$ mode corresponds to in-plane gliding of W and S atoms, while $A_g^1$ mode corresponds to distortion of W–S bonds and $B_u$ mode corresponds to stretching W–W bonds, along with out-of-plane displacements of S atoms[3]. Stiffening of Raman vibration modes with decrease of layer number indicates increases in W–W and W–S bond force constants, leading to more rigid S–W–S lattices in thinner 2M WS₂. This accounts for the

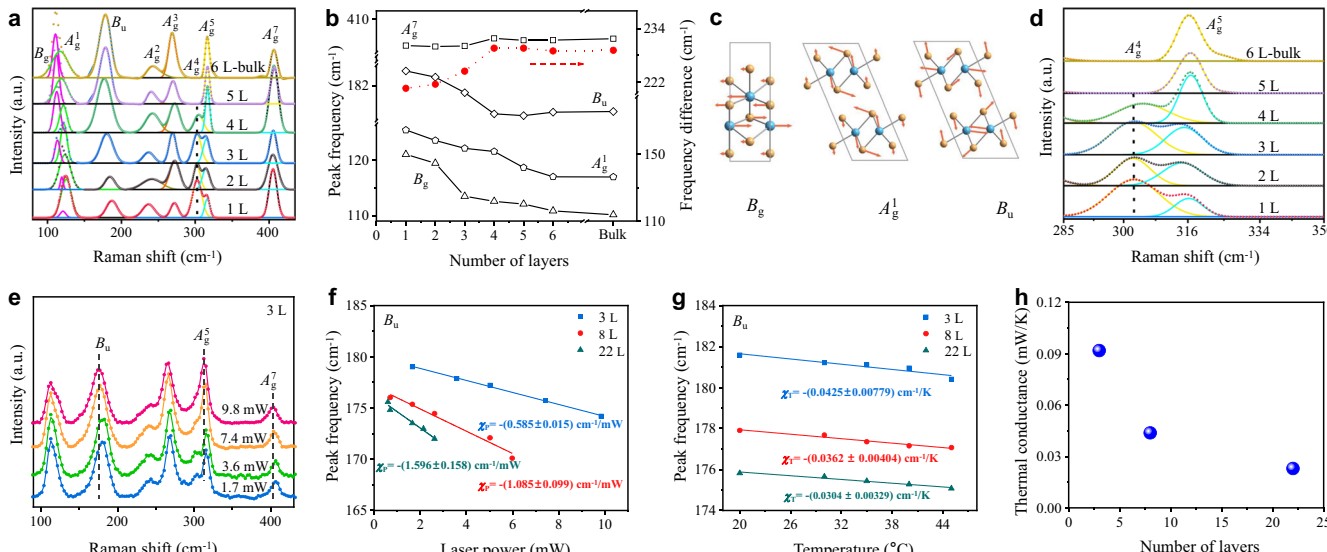

**Fig. 4 | Thickness-dependent intralayer bond strength and thermal conductance of 2M $WS_2$ on Si/SiO$_2$ substrate. a** Raman spectra of exfoliated thin (1 - 6L) and bulk 2M $WS_2$. The observed active Raman modes are labeled beside the corresponding peaks. **b** Frequencies of $B_g$, $A_g^1$, $B_u$ and $A_g^7$ Raman modes and the difference between frequencies of $B_u$ and $A_g^7$ modes (red dotted line) as a function of layer thickness of 2M $WS_2$. **c** Atomic displacements of the $B_g$, $A_g^1$ and $B_u$ Raman modes. $B_g$ mode is viewed in the $bc$ plane, while $A_g^1$ and $B_u$ modes are viewed in the $ac$ plane. **d** $A_g^4$ and $A_g^5$ Raman peaks of exfoliated thin (1–6L) and bulk 2M $WS_2$. **e** Raman spectra of a 3L 2M $WS_2$ acquired with different power laser excitation. Frequency of $B_u$ mode as functions of **f** laser power and **g** temperature for 3L, 8L and 22L 2M $WS_2$. Laser-power coefficient ($\chi_P$) and temperature coefficient ($\chi_T$) of $B_u$ mode frequencies are extracted from slopes of the corresponding plots. **h** Thermal conductivities of 3L, 8L and 22L 2M $WS_2$ estimated by $\chi_T$ divided by $\chi_P$. For each sample, an average thermal conductance is obtained based on three values calculated from three sets of $\chi_T$ and $\chi_P$ data that are extracted from $B_u$ mode plots in (**f**, **g**) and $A_g^5$ and $A_g^7$ modes plots in Supplementary Figs. 14, 15.

thickness-dependent stability of 2M $WS_2$. It should be noted that the high stability of ultrathin 2M $WS_2$ cannot be ascribed to substrate effect. The same trends of thickness-dependent Raman mode frequencies and phase stability of 2M $WS_2$ are also observed when the substrate is changed from Si/SiO$_2$ to polydimethylsiloxane (PDMS) that has minimum interaction with TMD flakes, as shown in Supplementary Fig. 13.

Additionally, the different thickness dependence of different Raman modes allows Raman frequency to be an indicator of the layer thickness. We have depicted the difference ($\Delta\omega$) between frequencies of $B_u$ and $A_g^7$ modes in Fig. 4b. For ML $WS_2$, $\Delta\omega$ is 220.5 cm$^{-1}$, which is smaller than the value of 221.5 cm$^{-1}$ for 2L and 224.5 cm$^{-1}$ for 3L $WS_2$. From ML to bulk $WS_2$, the $\Delta\omega$ increases from 220.5 cm$^{-1}$ to 229.2 cm$^{-1}$. This statistic of Raman mode frequencies provides a powerful tool for rapidly identifying thickness of 2M TMD, similar to that used for analyzing thickness of 2H TMD[28]. Another useful measure is the $A_g^4$ mode that turns to be detectable when 2M $WS_2$ is thinned to 4L as shown in Fig. 4d. $A_g^4$ mode has an intensity comparable with $A_g^5$ mode in 3L 2M $WS_2$ and exhibits higher intensity than $A_g^5$ mode in 2L 2M $WS_2$. Moreover, $A_g^4$ intensity is two times that of $A_g^5$ mode in ML 1T' $WS_2$ (Fig. 4d). The enhancing of $A_g^4$ mode with decreasing of layer number probably associates with change of vibration symmetries in ultrathin $WS_2$[2].

As 2M to 2H phase transition of multilayered $WS_2$ is probably a layer(s)-by-layer(s) slow process, thermal conductance plays an important role in phase stability of 2M $WS_2$. We then evaluate the thermal conductivities of 2M $WS_2$ flakes on SiO$_2$/Si substrates using temperature- and laser-power-dependent Raman spectroscopy[43]. With increase of temperature or laser power, the Raman vibration modes of all $WS_2$ flakes with various thicknesses are observed to soften (red shift), as shown in Fig. 4e and Supplementary Fig. 14. We have selected $B_u$, $A_g^5$ and $A_g^7$ modes to plot the peak frequency as functions of temperature or laser power, as shown in Fig. 4f, g and Supplementary Fig. 15a–d. The extracted temperature coefficients ($\chi_T$) of peak frequencies for different thicknesses are similar, in a range of 0.030–0.060 cm$^{-1}$/K. However, the extracted laser-power coefficients ($\chi_P$) of peak frequencies exhibit obviously thickness dependence, as

shown in Fig. 4f and Supplementary Fig. 15c, d. The $\chi_P$ of $B_u$ mode for 3L, 8L and 22L 2M $WS_2$ are 0.585, 1.085, and 1.596 cm$^{-1}$/mW, respectively. Thermal conductance can be estimated by $\chi_T$ divided by $\chi_P$[33,34,43,44]. Accordingly, the thermal conductance of 3L, 8L, and 22L 2M $WS_2$ are calculated to be 0.092, 0.044, and 0.023 mW/K (Fig. 4h), respectively, which clearly evidences thinner sample has higher thermal conductance. To deconvolute thermal conductance of the SiO$_2$/Si substrate from our measurements, we also tested Raman spectra of $WS_2$ at holey substrates (Supplementary Fig. 16). The as-obtained results also confirm thinner 2M $WS_2$ has higher thermal conductance, as shown in Supplementary Fig. 16, except that the calculated thermal conductivities at holey substrates are lower than the corresponding values at conventional SiO$_2$/Si substrates. Such reduction of thermal conductance with increase of thickness can be attributed to stronger phonon scattering and enhanced anharmonicity in thicker 2M $WS_2$[31]. The high thermal conductance of ultrathin 2M $WS_2$ can also be connected with the enhanced intralayer bonding that facilitates intralayer heat conduction. High thermal conductance enables high-efficiency heat dissipation during heating, leading to improved tolerance to elevated temperature for thinner 2M $WS_2$.

The different thermal conductivities between thick and thin $WS_2$ flakes can also be associated with their distinct phase distributions during the 2M to 2H phase change (Supplementary Fig. 17). Low thermal conductance of a thick 2M $WS_2$ flake results in nonuniform heat distribution during continuous heating process, and the 2M to 2H phase change occurs first at regions where heat is concentrated. Moreover, the non-concerted structural evolution among different layers can contribute to high interlayer phonon scattering and cause further low thermal conductance for the intermediate phase $WS_2$. Consequently, a sequential phase change at different regions leads to a 2M/2H stripes pattern on the $WS_2$ flake (Supplementary Fig. 17). In contrast, high thermal conductance of an ultrathin 2M $WS_2$ flake has uniform heat distribution, so the phase change homogeneously occurs at entire region of the flake when phase change temperature is reached (Supplementary Fig. 17).

## Phase transition simulation of different thicknesses WS$_2$

To study thickness-dependent phase transition kinetics, we have simulated the 2M to 2H phase change processes for 1L, 2L, and 3L WS$_2$ supercells (as shown in supplementary Fig. 18) by employing Vienna ab initio simulation package (VASP) and calculated energies of WS$_2$ lattices with different configurations by using climbing image nudged elastic band (CI-NEB) method. Like the transition state theory applied for chemical reactions, these calculations can find the transition state WS$_2$ lattice configuration during phase transition and identify the path passing through the transition state and connecting the initial and the final phase states. Such a path consists of a series of phase transition coordinates, including initial phase state (composed by 1T′ layers), final phase state (composed by 1H layers), transition state, and other intermediate lattice configurations. Method and principle used to find these coordinates are discussed in Part III of supporting information with Supplementary Fig. 19. The calculation output electronic files are attached as Supplementary Software 1. Configurations of all WS$_2$ lattices at the as-calculated coordinates (coord. I to coord. VII) are shown in Supplementary Figs. 20–26. During the simulated phase transition process, the octahedral coordinated lattices gradually deform until reach the transition states, which further evolve and relax to the trigonal prismatic coordinated lattices. Meanwhile, the WS$_2$ unit cell shrinks in $a$ and $b$ directions (Supplementary Fig. 27), which agrees to the afore-discussed phase transition mechanism.

During lattice deformation process of 1L 1T′ WS$_2$, W-atom line distance $d_1$ increases from 2.280 Å (coord. I) to 2.317 Å (coord. III), while W-atom line distance $d_2$ decreases from 3.340 Å (coord. I) to 3.342 Å (coord. III), as shown in Fig. 5a and Supplementary Fig. 20. These results reveal W-atom lines distances tend to be equalized along $b$ direction, in consistency with observations from TEM images (Fig. 5a–c) of WS$_2$ during 2M to 2H phase transition. With changing of W-atom line distances, S1 and S2 atoms are found to glide along $bc$ plane by large ranges from coord. I to III, rendering severe elongation of W2–S1 bonds to 2.558 Å and breakage of W4–S2 bonds (Fig. 5a). Meanwhile, the W1–S1 bond elongates from 2.404 Å (coord. I) to 2.517 Å (coord. III), while W3–S1 shortens from 2.438 Å (coord. I) to 2.319 Å (coord. III) and W3–S2 and W5–S2 bonds also shorten in coord. III. Similarly, W4–S3 and W4–S4 bond lengths are found to respectively increase and decrease from coord. I to III. Altering of W–S bonds lengths also results in out-of-plane displacements of S atoms, which makes distance between different S atomic planes reducing from 0.418 Å at coord. I to 0.273 Å at coord. III (Fig. 5b). The 1L WS$_2$ supercell reaches transition state at coord. III, with the highest lattice energy due to the significant lattice deformation. With further atomic gliding and reforming of W–S bonds, the lattice of 1L WS$_2$ turns to 1H type but with stretched W–S bonds in coord. IV (Fig. 5a). After further shortening of W–S bonds in the following three steps (Supplementary Fig. 20), the lattice of 1L WS$_2$ finally relaxes to a typical 1H structure in coord. VII (Fig. 5a).

The simulations of 2L and 3L WS$_2$ supercells reveal layer-by-layer phase transition mechanism, in good agreement with the XRD and Raman spectra of intermediate phases WS$_2$ (Fig. 3g–i). For 2L WS$_2$, the top layer changes from the initial 1T′ structure (coord. I) to a deformed structure at coord. IV, where the transition state is reached, and turns

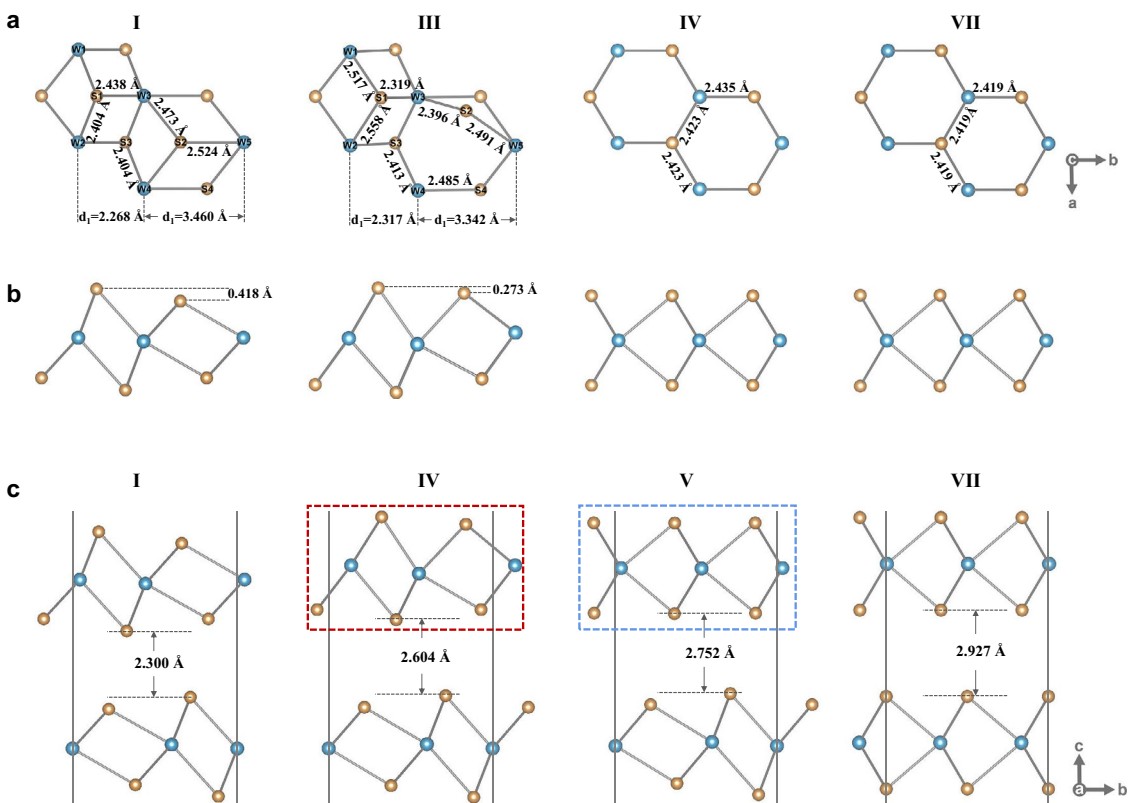

**Fig. 5 | Simulation of 2M to 2H phase transition of WS$_2$ using rectangular supercells.** Representative molecular geometries of the 1L WS$_2$ supercell in the initial 1T′ phase (coord. I), transition state (coord. III), an intermediate configuration (coord. IV) and the final 1H phase (coord. VII), viewing from (**a**) the $c$ direction and (**b**) the $a$ direction. Process from coord. I to coord. III corresponds to deformation of 1T′-type lattice and process from coord. IV to coord. VII corresponds to relaxation of 1H-type lattice. **c** Molecular geometries of the 2L WS$_2$ supercell at the initial phase state (coord. I), transition state (coord. IV), an intermediate configuration

(coord. V) and the final phase state (coord. VII), viewing from the $a$ direction. The highly deformed layer at the transition state is marked by a red dotted square. The layer turned from 1T′ to 1H type of structures is marked by a blue dotted square. Interlayer spacings are denoted in different states, showing interlayer expansion from coord. I to coord. VII. Lattice edges are depicted in different states, referring to which interlayer dislocation between top and bottom layers can be seen. Color code: blue and orange spheres represent W and S, respectively.

to 1H-type structure at coord. V that then relaxes to coord. VII. The bottom layer is gradually distorted from the initial 1T' structure until coord. IV and changes to 1H structure from coord. VI to VII, as shown in Fig. 5c and Supplementary Figs. 21, 22. Similarly, the bottom layer of 3L WS₂ first transforms to 1H structure, followed by the mid layer and top layer, seen from Supplementary Figs. 23–26. 1T'/1H heterostructures are seen at coord. V for 2L (Fig. 5c) and 3L WS₂ (Supplementary Fig. 26). At transition states, both 2L and 3L WS₂ contain a highly deformed layer, while other layers are also distorted from the initial 1T' phase, as shown in Supplementary Table 2. The deformed layers in 2L and 3L WS₂ have larger structural variations from the initial 1T' lattice compared to that in 1L WS₂. The two W-atom line distances get closer in transition state 2L 3L WS₂ than 1L WS₂ (Supplementary Table 2). In addition, the deformed layer in 3L WS₂ have two broken W–S bonds, while deformed layers in 1L and 2L WS₂ include only one (Supplementary Table 2).

The total phase transition barrier of a WS₂ supercell is defined as the energy difference between the corresponding transition and initial states. As thicker WS₂ supercell involves larger structural difference between the corresponding transition and initial states, the total phase transition barrier increases from 1.83 eV to 1.95 eV and 2.10 eV, as the number of layers increases from 1L to 2L and 3L, as shown in Fig. 6a. After normalizing the total phase transition barriers by the number of layers, average transition barriers per layer of 1L, 2L, and 3L WS₂ supercells are calculated to be 1.83 eV, 0.97 eV and 0.70 eV (Fig. 6b), respectively, exhibiting the average transition barrier decreases with the increase of layer number. This result theoretically confirms thinner 2M WS₂ has higher phase stability. It should be noted that the average transition barrier per layer decreases more and more slowly as the number of layers increases from 1L to 2L and 3L (Fig. 6c), implying the average transition barriers per layer for thick WS₂ flakes would converge to a certain value. This agrees to the fact that phase transition temperatures of WS₂ flakes thicker than 5L converge to 120 °C (Fig. 1e).

## Discussion

The higher intralayer bond strength in thinner 2M WS₂ can be associated with reduction of interlayer vdW interactions. As is known, higher stability of 1T' TMD than the corresponding 1T TMD originates from the distorted lattice that generates strong intralayer metal–metal bonds and prominent intralayer Coulombic interactions between metal and chalcogen atoms[4]. An interlayer Coulombic interaction, such as vdW force, brings screening effect to the intralayer interactions and weaken the intralayer bond strength[26–28]. As the interlayer vdW interaction decreases with reduction of thickness from bulk to a few layers, the intralayer bond strength accordingly increases and

reaches maximum in ML 1T' TMD, meaning that ML 1T' TMD has the highest intrinsic phase stability.

VdW interaction also plays an important role in the phase transition process of multilayered WS₂. One the one hand, vdW-interaction linked layers tend to deform simultaneously during phase transition, resulting in increasing of total transition barrier with the number of layers. One the other hand, the existence of vdW interaction can constrain the freedom of lattice deformation, to minimize the layer stacking energy. Consequently, phase transition of multilayered WS₂ occurs layer(s) by layer(s). In fact, WS₂ lattices always tend to minimize the vdW-interaction-associated stacking energy by optimizing the stack configuration. Interlayer expansion and dislocation during phase transition have been demonstrated by the simulated molecular geometries (Fig. 5c and Supplementary Fig. 26) and physical characterizations (Fig. 3d–f and Supplementary Fig. 12) of multilayered WS₂. For a relatively thick WS₂ (≥5L), transition state lattice might include two or more highly deformed layers that are widely separated since vdW interactions between them are small. Once the number of deformed layers existing at the transition state turns to be proportionate to the total number of WS₂ layers, the average phase transition barrier per layer will converge to a constant value for thick and bulk WS₂ flakes.

In summary, we have demonstrated ultrathin 2M WS₂ has significantly higher thermal stabilities than the bulk counterparts. Both phase transition temperature and durability of 2M WS₂ remarkably increase with thickness decreasing from bulk to ML. By analyzing 2M to 2H phase transition mechanism and Raman spectra of 2M WS₂ with different thicknesses, the higher stability of thinner 2M WS₂ is associated with stiffened intralayer bonds and enhanced thermal conductance, due to reduction of interlayer interactions. Theoretical calculation indicates average transition barrier increases with reduction of WS₂ layer number, since the phase transition occurs through a layer(s)-by-layer(s) mechanism. The high intrinsic phase stabilities of ultrathin 2M TMDs can inspire their tempting applications in various fields, including superconductor, electronics and energy conversion and storage.

## Methods

### Fabrication of different layered 2M WS₂
K₀.₇WS₂ crystals were synthesized as previously reported[7]. Specifically, K₂S₂ (prepared via liquid ammonia), W (99.9%, Alfa Aesar), and S (99.99%, Alfa Aesar) were mixed by the stoichiometric ratios and ground in an argon-filled glovebox. The mixtures were pressed into a pellet and sealed in the evacuated quartz tube. The tube was heated at 850 °C for 2000 min and slowly cooled to 550 °C at a rate of 0.1 °C min⁻¹. 2M WS₂ single crystals were obtained by oxidizing K₀.₇WS₂ in an aqueous solution containing 0.01 M K₂Cr₂O₇ and 0.02M H₂SO₄ at

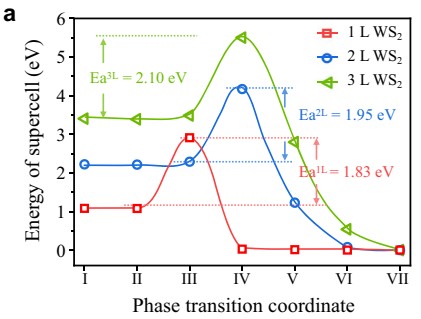
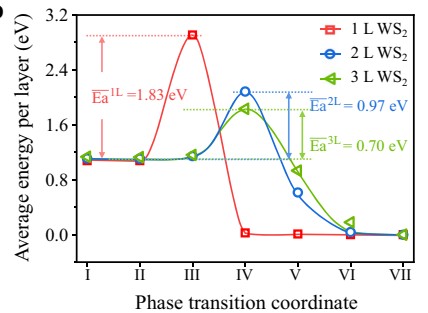
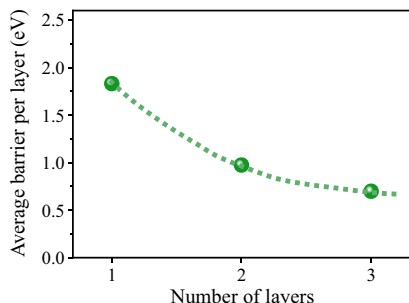

**Fig. 6 | Calculated energy profiles of 2M to 2H phase transition. a** Total energies and **b** average energies of 1L, 2L, and 3L WS₂ supercells at the initial state (coord. I), transition states (coord. III for 1L WS₂ and coord. IV for 2L and 3L WS₂), final states (coord. VII) and other intermediate configurations. Solid lines are guides to the eye. Energies in (**b**) are normalized by the corresponding numbers of WS₂ layers. Ea$^{1L}$, Ea$^{2L}$, and Ea$^{3L}$ are total phase transition energy barriers for the 1L, 2L, and 3L WS₂ supercells, respectively. $\overline{Ea}^{1L}$, $\overline{Ea}^{2L}$ and $\overline{Ea}^{3L}$ are average transition barriers per layer for the 1L, 2L, and 3L WS₂ supercells, respectively. **c** The average transition barrier per layer as function of the number of layers. A dashed line is a guide to the eye.

room temperature for 1 h. Different layered $WS_2$ flakes were mechanically exfoliated from the synthesized bulk 2M $WS_2$ onto a $SiO_2/Si$ substrate or a holey $SiO_2/Si$ substrate. To transfer the exfoliated $WS_2$ from $SiO_2/Si$ to PDMS, the $SiO_2/Si$ substrate was spin-coated with PMMA film and etched by KOH solution. A piece of PDMS film was used to pick up the $WS_2$/PMMA and the PMMA film was removed by soaking in acetone for an hour. More details about sample preparation are available in Supplementary Note 1.

### Temperature- and laser-power-dependent Raman measurements

Micro-Raman spectra were taken on the 2M $WS_2$ using a confocal microscope Raman system (WITec, Alpha300R) with an optical microscope (Nikon). For temperature-dependent Raman measurement, the exfoliated 2M $WS_2$ on a $Si/SiO_2$ or holey $Si/SiO_2$ or PDMS substrate was put on a hot plate, and temperature was increased by 5 °C and held for 1 min or 15 min in each heating step. The heating treatment was carried out either in the air or in a glovebox with an Ar atmosphere. After heating, the sample was transferred to the Raman instrument for Raman spectra acquisition. For laser-power-dependent Raman measurement, the powers of laser irradiating on the $WS_2$ flakes were quantified by a power meter (Thorlabs). More details about Raman measurements are available in Supplementary Note 2.

### Physical characterizations

The morphology and microstructure of the 2M $WS_2$ flakes were transferred to microgrid and examined by transmission electron microscopy (TEM, FEI Verios G4). The thickness and morphology were analyzed by using AFM images, which were captured with a Bruker Dimension Icon AFM under tapping mode. The samples were characterized by XRD pattern (Bruker D8 Advanced).

### Computational method

All the calculations were performed by the employee of the VASP[45]. The Perdew–Burke–Ernzerhof form of the generalized gradient approximation was employed to describe electronic exchange and correlation[46]. Lattice constants and atom positions of $WS_2$ in 2H and 2M phases, respectively, were optimized by using the conjugate gradient algorithm until the maximum force on a single atom is less than 0.02 eV/Å. The thickness of the vacuum layer exceeded 15 Å to eliminate the fictitious interaction caused by periodic cells. The van der Waals interactions were depicted in terms of the $D_3$ method of Grimme[47]. The CI-NEB method was adopted to locate the transition state from 2H to 2M[48]. More details about computational methods are available in Supplementary Note 3.

## Data availability

All data supporting the findings of this study are available within the paper and the supplementary information files. Additional data are available from the corresponding authors upon request.

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

## Acknowledgements
P.Z. acknowledges Dr. Jingzhi Shang for his assistance with Raman measurements. The authors acknowledge the financial support of the National Key Research and Development Program (2022YFE0121000), the National Natural Science Foundation of China (62288102), Natural Science Foundation of Shaanxi (Grant No. 5110210130), Key Research and Development Program of Shaanxi (Grant No. 5140220004) and Fundamental Research Funds for the Central Universities (Grant nos. G2022WD01007 and D5000230125), Science and Technology Commission of Shanghai Municipality (21ZR1473300).

## Author contributions
X. L., P. Z., S. W. and Y. F. contributed to this article equally. X. L. and C. G. conceived and proposed this project. Y. F. synthesized bulk 1T' WS2. P. Z. prepared exfoliated 1T' WS2 samples and carried out physical characterizations and stability measurements. S. W. carried out theoretical simulation. P. W. and Y. X. contributed to the collection of data for the revised manuscript. J. C., C. Z., X. Z. and W. Z. provided important insights in experimental design and manuscript writing. J. W., F. H and C. G. supervised the work. All the authors discussed the results and contributed to writing the manuscript.

## Competing interests
The authors declare no competing interests.
