## [Peer Review File · Nature Communications]

High Intrinsic Phase Stability of Ultrathin 2M WS₂Editorial Note: Parts of this Peer Review File have been redacted as indicated to remove third-party material where no permission to publish could be obtained.

REVIEWER COMMENTS

Reviewer #1 (Remarks to the Author):

The authors performed a systematic study on the phase transition stability of 1T' phase WS₂ with different layers. They found that the thin layer WS₂ is more resistant to phase transition, particularly single layer WS₂ can withstand temperatures up to 350°C. They attributed the higher stability of thinner 1T' WS₂ to the stiffening of intralayer bonds and increased thermal conductivity, resulting from the reduction of interlayer interactions, via thermal conductivity measurement and theoretical simulation. The simulation of the phase transition process of WS₂ from 1T' to 2H is kind of interesting, but thermal conductivity measurement and Raman spectroscopy analysis seem normal for lack of innovation. Furthermore, TMDs with 1T' phase have essential practical applications, which have not been examined in the article. In addition, the accuracy of some of the findings should be verified. As a result, I recommend that the authors revise their manuscript based on the following comments and submit it to a more specialized journal.

1 The optical image in Figure 1a is highly blurred, and the contrast between temperatures is not visible. Additionally, the boundaries of the Raman mapping images in Figures 1b and c are not distinct, making it hard to identify the area of the sample. The white outline provided by the author in the images is relatively faint.

2 At 350°C, the single WS₂ of the 1T' phase was no longer present, signifying that the sample was not stable. Despite this, the author maintained that the thin layer 1T' phase sample was more stable (On page 5, line 95 to 98), why?

3 How can one differentiate between shorter W-atom lines distance d_1 (0.23 nm) and longer W-atom lines distance d_2 (0.34 nm) in the high-resolution TEM image in Figure 2d, given that the author's current annotations do not correspond?

4 The author believes that the wrinkles on the surface of the sample in the intermediate state in Figure 2k are caused by interlayer dislocations due to molecular movement during the phase transition process. However, as we know, the microscopic molecular slip cannot be characterized by the surface morphology of AFM, so it is suggested to analyze the interaction and stress between the substrate and the substrate from a different perspective.

5 It is widely known that WS₂ of varying thicknesses have different absorption powers for lasers, thus, it is more appropriate to use a suspended substrate for thermal conductivity testing, similar to reference 34, in order to make the abscissa reference in Figure 3f meaningful.

6 The author's main focus in the article is WS₂, but it has been repeatedly mentioned in the introduction that the stability of 1T' phase TMDs is being studied. WS₂ is only one type of TMDs and cannot be representative of all TMDs. Thus, it is suggested that the author reorganize the introduction section to improve readability.

7 The photoluminescence spectrum of 2H WS₂ typically decreases in intensity as the number of layers increases. To further confirm the phase transition of the 1T' WS₂, it is suggested that the author observe the changes in PL peaks during the phase transition process more closely.

8 How can the author's statement on page 15, line 316 regarding the layer-by-layer phase transition of

WS2 of BL and TL be demonstrated through experiments?

Reviewer #2 (Remarks to the Author):

This manuscript reports on and investigates the higher phase stability of thin 1T'-WS2 as compared to bulk. The result is noteworthy and I think the core message is likely correct, I however have quite a few doubts about some specific conclusions. I further find the manuscript hard to follow and in large parts confusing, especially the later sections. I thus, at this stage, cannot recommend publication. I would be happy to look at the manuscript again after the authors addressed the following concerns. In general, for reviewing it would help if the figures are not placed at the end of the manuscript. Scrolling back and forth and trying to understand the phase transformation process was very tedious. If the editors agree, I would encourage the authors to include figures in the text when submitting the revised version.

1) I have strong doubts about the claim that bulk 1T'-WS2 (which by the way is usually called 2M-WS2 in the literature, although I don't insist to name it this way) will transform to 2H by itself without heating or laser power. My lab has worked with this phase for a while and we have a lot of experience handling the bulk crystal or powder. We never observed any phase transformation, not even partial, when the crystals, or powder were stored in either air or argon. We have, just as the authors report, observed the phase transformation with heating or with too high laser power when performing Raman experiments. I thus have a very hard time believing the authors report of a spontaneous phase transition that does not require any activation energy. I wonder if the Raman spectra of the samples that were measured after different times (I.e Fig S3) were always measured at the same sample spot. It could be that samples are locally different and thus they appear to start transforming. Also, the 0 days Raman spectrum of the 23K sample (Fig. S3 d) looks like it has a bad signal to noise ratio, maybe some 2H phase is already present. We have noticed before in bulk samples, that sometimes a phase mixture is present in fresh samples and the Raman spectrum can depend on the location of the measurement.

2) In conjunction with this I have some doubt with the proposed mechanism. I wonder if the higher stability stems from impurities in bulk samples which grow quickly, catalyzing the phase transformation. the authors mostly use Raman spectroscopy to analyze their samples, Raman can easily miss sample imperfections, small impurities etc. High-resolution powder diffraction together with Rietveld refinement might give more information. In our lab we found for example, that higher purity bulk samples can be heated to higher temperatures (~230 C) before they transform as compared to low quality bulk samples, which transform at 180 C (according to DSC).

3) I find the entire theory section highly confusing and would suggest a re-write. Maybe I understand the proposed mechanism better after. It is still not clear to me what exactly the different phases (I, II, etc) are and also where they come from. Did DFT suggest these as intermediates? Did the authors suggest them and then calculated then? I'm sorry but this part was very hard to follow.

4) I'm also confused about the oxidation discussion. How does oxidation speed up or cause the phase transition? the 1T'-2H phase transition does not involve any change of oxidation state so it has nothing to do with oxidation. If oxidizing environmental effects speed it up, I would argue even more that the

phase transformation is impurity driven and that the proposed mechanism about weaker vdW interactions is then false.

5) I'm also confused about Fig 2. I think this figure needs to be revised. These are the things that confuse me:

5i) in the HRTEM images which zone axis are we looking at? Can it be labeled? I assume we look at a layer (ie. Plan-view?) it would be good to clarify. And if we have plan view samples why is the resolution of Fig 2d so bad? I have seen many images of 1T'-WS₂ phases where the W-W zig-zag chains are very clearly resolved.

5ii) Fig 2h-j: what are the numbers (ie 25 nm etc) refer to? They are neither explained in the caption or text or I did not find the explanation.

5iii) I'm generally confused by the discussion of the intermediate phase, especially in respect to the layer distance. In the XRD, it is clear that the intermediate phase is a mixture of 2H and 1T'. There are clearly 2 peaks which overlap, each representing to one stacking distance. Nonetheless the authors keep discussing the intermediate phase as if it would be its own phase, with an average layer distance. I don't think this is correct. Fig. 2k makes no sense therefore. Also why does the intermediate phase have additional peaks? One appears clearly at ~ 34 degrees, but more minor ones are also visible.

6) in relation to above, I'm confused by the discussion about the intermediate phase in the text as well. How was the layer distance calculated? Why is the stacking peak in the x-ray called 200 if stacking is along b? Shouldn't it be 020?

Minor points:

a) Please plot all Raman spectra on the same x-axis (ie Figure S3 plots some panels to 500 and other to 450 wavenumbers which makes comparing the panels difficult).

b) in the end of the conclusion the authors generalize that the 1T' phase is more stable in thin TMDs. I would not make this claim, it might be very different in WTe₂ or other TMDs. In the case of WTe₂ the monolayer is known to be MUCH more air sensitive than the bulk (although neither does transform to 2H).

c) the authors use crystal settings in which the 2H-phase stacks along the c-axis but the 1T' along the b-axis. I know these are standard crystallographic settings but it might be easier to choose an orthorhombic setting for the 1T'-phase in which the layers also stack along c. makes comparison between the two and following hkl indices easier.

d) the authors might want to cite the following work where the phase transition in WS₂ was reported before: <https://pubs.aip.org/aip/apl/article-abstract/115/3/032102/37971/Evidence-for-a-narrow-band-gap-phase-in-1T-WS2?redirectedFrom=fulltext>

e) the authors might want to discuss the following report in which high phase stability of 1T'-ML is also reported: <https://www.science.org/doi/full/10.1126/sciadv.add6167>

To sum up, I think the general result might be valid but there are many unresolved issues with this manuscript. In addition, because parts were confusing, this review is not complete. I would look at the paper again but it will likely need a few rounds of revision before being suitable for publication.

Reviewer #3 (Remarks to the Author):

The authors report on the layer-number stability of metastable WS₂. They found that the atomically-thin 1T' WS₂ possesses higher phase and anti-oxidant stability compared to its bulk counterpart, confirmed by Raman and optical microscopy with annealing. They also demonstrated the existence of intermediate states between the 1T' and 2H phases, as revealed by Raman spectroscopy, XRD, TEM, and DFT calculations. The authors proposed that the enhanced thermal conductivity and the stiffening of intralayer bonds at the monolayer limit are the causes of this phenomenon.

Although the long-term stability of monolayer 1T' MoS₂ and WS₂ has been reported previously (e.g., L. Liu et al., *Nat. Mater.*, 17, 1108 (2018), and M. Okada et al., *ACS Nano*, 16, 13069 (2022)), the authors' findings concerning the enhanced phase stability of monolayer 1T' TMDs over the bulk form are intriguing. In general, the opposite is observed (F. Ye et al., *Small*, 42, 5802 (2016)). As such, I believe that these results will attract significant interest and propel the scientific study of metastable TMDs forward. However, I recommend a major revision prior to acceptance, as there are several points that need improvement. In particular, analysis of the intermediate state and the discussion of the mechanism are insufficient.

1. In Figure S2-5, the authors should display all the Raman spectra that were used to construct Fig. 1.
2. The authors should provide the definition of the transition temperature in Fig. 1e. For instance, this could be the temperature at which the intermediate state emerges or the temperature at which the phase completely transitions into the 2H phase.
3. At line 100, the authors should specify the number of samples they measured. The phrase "many samples" lacks precision.
4. The authors contend that an "intermediate state" exists. However, the structure is neither in the 2H phase nor the 1T' phase (Fig. 2c). If this is the case, it seems peculiar that the Raman spectra of the intermediate state are merely the sum of the spectra of the two crystalline phases. Are there any other alterations in the Raman spectra attributable to the formation of the intermediate state? Authors should discuss about Raman scattering of the intermediate state deeper.
5. In Fig. S6, is the crystallographic structure data sourced from previous reports or was it calculated by the authors? The authors need to clarify this point. If the structure data was referenced from a previous paper, the authors should cite the article.
6. Regarding the XRD (Fig 2g), what is the origin of the peak around 32 degrees in the intermediate sample? The authors should assign the peak.
7. The authors assert a layer-by-layer phase transition, but could this merely represent a reduced energy barrier due to the increased number of WS₂ formula unit in the unit cell, stemming from the increased number of layers? The actual value is approximately 1/2 or 1/3 of the monolayer. Thus, from a certain layer, the phase change barrier remains constant. The proposed mechanism should

therefore be reconsidered.

Other issues with the text structure include:

8. Figures should be numbered in the sequence in which they appear.

9. Excessive significant figures in the AFM heights (Fig. 2k), chemical bond angles/lengths in lines 138-145 and 267-282, etc., need to be reviewed and corrected.

10. The methods section lacks sufficient detail on the DFT calculation: information such as k-mesh, cut-off energy, etc., is missing. Please provide enough information to allow other researchers to reproduce the results.

11. Definitions of abbreviations must be given at their first usage. Several terms, such as "T-type", are used without definition. There are also instances where abbreviations are defined but not used subsequently (e.g., 3 L and TL). Please review the manuscript and rectify these issues.

12. There are numerous typographical errors in the manuscript that need to be corrected.

Point-by-Point Responses to Reviewers' Comments

Referee: 1

Comments: *The authors performed a systematic study on the phase transition stability of 1T' phase WS₂ with different layers. They found that the thin layer WS₂ is more resistant to phase transition, particularly single layer WS₂ can withstand temperatures up to 350°C. They attributed the higher stability of thinner 1T' WS₂ to the stiffening of intralayer bonds and increased thermal conductivity, resulting from the reduction of interlayer interactions, via thermal conductivity measurement and theoretical simulation. The simulation of the phase transition process of WS₂ from 1T' to 2H is kind of interesting, but thermal conductivity measurement and Raman spectroscopy analysis seem normal for lack of innovation. Furthermore, TMDs with 1T' phase have essential practical applications, which have not been examined in the article. In addition, the accuracy of some of the findings should be verified. As a result, I recommend that the authors revise their manuscript based on the following comments and submit it to a more specialized journal.*

Response: We appreciate the reviewer's comments and thanks for the positive feedback about the simulation of phase transition. We agree that the thermal and Raman measurements applied in our work are common experimental skills, but it is quite novel to analyze thickness dependent Raman spectra on single-crystalline 2M (or 1T') phase TMD to the best of our knowledge. Based on the obtained results, we have unambiguously concluded thinner 2M WS₂ has higher thermal stability, and we have also provided a useful measure to identify the thickness of ultrathin 2M WS₂ according to Raman spectra. There have been plenty of publications demonstrating potential applications of 2M (or 1T') phase TMDs, especially for the ultrathin ones, whereas a consensus is that the practical applications of 2M (or 1T') TMD are hindered by their metastability. Although we did not repeatedly examine the application of 2M (or 1T') TMD in our work, our findings could significantly abate the worry about 2M (or 1T') to 2H phase change and inspire practical applications of 2M (or 1T') TMD in various fields, such as superconductors, optical materials, and energy storage. Accordingly, our findings can attract strong interests from a broad readership, including materials scientists, nanotechnologists, and energy researchers. In the revised manuscript, we have carefully addressed all the reviewers' questions and strongly verified the accuracy of our findings by following all the

reviewers' suggestions. Therefore, we believe the revised manuscript deserves publication in *Nature Communications*.

Comment 1: *The optical image in Figure 1a is highly blurred, and the contrast between temperatures is not visible. Additionally, the boundaries of the Raman mapping images in Figures 1b and c are not distinct, making it hard to identify the area of the sample. The white outline provided by the author in the images is relatively faint.*

Response: To improve the image quality of Fig. 1, we have prepared new WS₂ flakes with relatively larger sizes and reprocessed in situ Raman measurements. The revised Fig. 1a-b (attached below) shows a piece of WS₂ flake that has four distinct areas with thicknesses of 4 L, 3 L, 2 L and 1 L. It is clear to see that transparency of 3-4 L areas are decreasing with increase of temperature (Fig. 1a). This is because higher temperature results in higher extent of 2M to 2H phase transition, and 2H WS₂ has higher infraction index and lower transparency than 2M WS₂. However, the 2 L area is too thin to tell transparency change during 2M to 2H phase change, but the phase change can be demonstrated by emergence of E_{2g}¹ Raman mode at 210 °C (Fig. 1 b4). The optical image of 1 L area doesn't change from RT to 325 °C, either, because 1T' to 1H phase transition doesn't occur in this range of temperature. However, 1 L area disappears at 350 °C (Fig. 1 a6) because of air oxidation, which we discuss in detail in the Response to Comment 2. To make the Raman mappings of different thickness areas easier to identify, we have used white dotted lines to outline the 1-4 L areas in the revised Fig. 1b.

Revised Fig. 1a-b. Thickness-dependent phase stability of 2M WS₂ on Si/SiO₂ substrate. (a1) Optical and (b1) AFM images of a piece of exfoliated 2M WS₂ flake with different thicknesses areas. (a2-6) Optical images and (b2-5) E_{2g}¹ Raman mode and (b6) PL intensities mappings of this flake at different temperatures in the air. Different thicknesses areas are outlined with dotted lines.

In the revised manuscript, we have modified the text discussing Fig. 1a-b as:

Here we define phase transition temperature as the one at which the 2M or 1T' WS₂ starts to lose phase purity, i.e., the E_{2g}¹ Raman mode emerges or the WS₂ sample begins to be oxidized by the air. To investigate the thickness-dependent phase stability, 2M WS₂ flakes are mechanically exfoliated into thin layers. Fig. 1 a1 and Fig. 1 b1 show optical and atomic force microscope (AFM) images of a piece of exfoliated 2M WS₂ flake with thickness varying from ML to 4 L. This sample was treated with stepped heating in the air with temperature elevated by 5 °C and held for 1 min in each stage, during which the E_{2g}¹ intensity mappings are acquired to address the location of 2M to 2H (or 1T' to 1H) phase change. It is found that the E_{2g}¹ mode can be observed at the 4 L area when temperature goes up to 180 °C (Fig. 1 b2), whereas it does not appear at the trilayer (TL) and BL areas until the temperature goes to 190 °C (Fig. 1 b3), 210 °C (Fig. 1 b4), respectively. With temperature increasing from 180 °C to 325 °C, the E_{2g}¹ mode intensities of 3-4 L areas keep increasing (Fig. 1 b2-5) and the transparency is obviously decreasing (Fig. 1 a2-5), while the BL area is too thin to tell transparency change. It is known that 2H WS₂ has higher infraction index and lower transparency than 2M WS₂.³⁵ Photoluminescence (PL) peaks (Supplementary Fig. 3a-c) at ~ 640 nm emerge in 2-4 L areas as temperature reaches 325 °C, further confirming the 2M to 2H phase transitions³⁶. The PL intensities are significantly enhanced after heating to 350 °C, as shown in Supplementary Fig. 3a-c, consistent with the increase of E_{2g}¹ mode intensity at elevated temperature. These results indicate 2M to 2H phase transition is a slow process, and the extent of phase transition augments at elevated temperatures or after longer time of heating. PL intensity of WS₂ is sensitive to thickness²⁶, and the distribution of PL intensity (Fig. 1 b6) agrees to the AFM defined areas of various thicknesses from BL to 4 L. Comparatively, the ML WS₂ area preserves the typical 1T' Raman pattern without

emergence of E_{2g}^1 mode or PL until temperature reaches 350 °C (Fig. 1b and Supplementary Fig. 3d-e), evidencing the highest thermal stability among the four WS₂ areas. In agreement with this result, chemically exfoliated ML 1T' WS₂ is also demonstrated to have good air-stability at room temperature^{37,38}. Heated at 350 °C in the air, Raman signal of the 1T' WS₂ fades away and the 1T' WS₂ flake disappears, seen from Fig. 1 a6, which can be attributed to decomposition due to air oxidation. This means the decomposition temperature of ML 1T' WS₂ in the air is lower than 1T' to 1H phase transition temperature and it is impossible to observe 1T' to 1H phase transition by heating ML WS₂ in the air.

Changes to the manuscript: Fig. 1a-b is revised and is discussed on Page 5-7.

Comment 2: At 350 °C, the single WS₂ of the 1T' phase was no longer present, signifying that the sample was not stable. Despite this, the author maintained that the thin layer 1T' phase sample was more stable (On page 5, line 95 to 98), why?

Response: We are aware that the description about the stability of monolayered WS₂ is confusing in the previous version manuscript. Although the monolayered 1T' WS₂ is oxidized and decomposed in the air at 350 °C before transforming to 1H phase, it has well preserved pure 1T' phase before 350 °C, evidencing by the absence of E_{2g}^1 Raman mode and photoluminescence (PL) (seen from the revised Supplementary Fig. 3d-e, attached below), while the 2 L, 3 L and 4 L WS₂ areas in Fig. 1a start to lose 2M phase purity and transform to 2H phase from 210 °C, 190 °C and 180 °C, respectively. Therefore, thinner 2M (or 1T') WS₂ sample has higher thermal stability. To clear up doubts from the corresponding statements, before discussing Fig. 1 in the revised manuscript, we define the phase transition temperature as the one at which the 2M or 1T' WS₂ starts to lose phase purity, involving both circumstances of 2M to 2H (or 1T' to 1H) phase transition and 1T' WS₂ being oxidized by the air.

Revised Supplementary Fig. 3d-e. (d) Raman and (e) PL spectra of ML 1T' WS₂ in Fig. 1a heated at different temperatures in the air, evidencing the ML 1T' WS₂ is stable before 350 °C in the air.

In the revised manuscript, the phase transition temperature is defined as:

Here we define phase transition temperature as the one at which the 2M or 1T' WS₂ starts to lose phase purity, i.e., the E_{2g}¹ Raman mode emerges or the WS₂ sample begins to be oxidized by the air.

In the revised manuscript, thermal stability of monolayered 1T' WS₂ in the air is discussed as:

Comparatively, the ML WS₂ area preserves the typical 1T' Raman pattern without emergence of E_{2g}¹ mode or PL until temperature reaches 350 °C (Fig. 1b and Supplementary Fig. 3d-e), evidencing the highest thermal stability among the four WS₂ areas. In agreement with this result, chemically exfoliated ML 1T' WS₂ is also demonstrated to have good air-stability at room temperature^{37,38}. Heated at 350 °C in the air, Raman signal of the 1T' WS₂ fades away and the 1T' WS₂ flake disappears, seen from Fig. 1 a6, which can be attributed to decomposition due to air oxidation. This means the decomposition temperature of ML 1T' WS₂ in the air is lower than 1T' to 1H phase transition temperature and it is impossible to observe 1T' to 1H phase transition by heating ML WS₂ in the air.

Changes to the manuscript: (1) Raman and PL spectra of monolayered 1T' WS₂ are added in Supplementary Fig. 3d-e; (2) Definition of phase transition temperature and stability of monolayered 1T' WS₂ in the air are discussed on Page 6.

Comment 3: How can one differentiate between shorter W-atom lines distance d_1 (0.23 nm) and longer W-atom lines distance d_2 (0.34 nm) in the high-resolution TEM image in Figure 2d, given that the author's current annotations do not correspond?

Response: Thanks for pointing out the inadequate annotations in Fig. 2d. In the revised figure (attached below), we have updated Fig. 2d with a higher-resolution TEM image and have clearly labeled d_1 (0.23 nm) and d_2 (0.34 nm).

Revised Fig. 2d. HRTEM images and the corresponding Fourier transforms of 2M WS₂.

Changes to the manuscript: Fig. 2d is updated and is discussed on Page 10-12.

Comment 4: The author believes that the wrinkles on the surface of the sample in the intermediate state in Figure 2k are caused by interlayer dislocations due to molecular movement during the phase transition process. However, as we know, the microscopic molecular slip cannot be characterized by the surface morphology of AFM, so it is suggested to analyze the interaction and stress between the substrate and the substrate from a different perspective.

Response: We realize wrinkles on the WS₂ surface can be attributed to non-homogeneous in-plane phase change but are not closely associated with interlayer dislocation. Moreover, wrinkles are not commonly seen in all the tested WS₂ flakes during 2M to 2H phase change. Therefore, we have removed the discussion about wrinkles from our revised manuscript.

We agree surface morphology of AFM is hard to characterize molecular slip if it has too small range. However, based on the lattice parameters shown in Supplementary Fig. 24 (attached below), a

unit cell of monolayered WS₂ contracts from 5.728 Å to 5.521 Å in the *b* direction as it transforms from 1T' to 1H phase, i. e. about 3.6% in-plane shrinkage. For a tested WS₂ flake with a side size of 10 μm, 1T' to 1H phase in a layer can result in about 360 nm lateral shrinkage. If the neighbor layers maintain the original 1T' structure, as phase change takes place layer by layer, the 3.6% lateral shrinkage can cause up to 360 nm sliding at edge of the phase-changed layer relative to other layers. Furthermore, the actual interlayer sliding can be even larger, because relaxation of interlayer stacking configuration can cause additional interlayer sliding, as shown in Fig. 5c (attached below). Accordingly, the range of interlayer sliding is large enough to be observed in the surface morphology of AFM, as shown in the revised Fig. 2i-k (attached below). To further confirm the interlayer sliding, we captured AFM images on another thinner WS₂ flake (about 6 L) with more distinct edges during 2M to 2H phase transition. As shown in the Supplementary Fig. 12 (attached below), interlayer sliding at the edge area can be clearly seen from 2M to intermediate and 2H phases, while the sliding direction is opposite to that in the revised Fig. 2i-k. Different sliding directions can be attributed to different relaxations of interlayer stacking configurations.

We understand the reviewer suggests analyzing the stress between the intermediate phase WS₂ and the substrate and watching the dislocation between WS₂ and the substrate from a cross section. We agree this is a good idea for investigating the effect of substrate interaction on 2M to 2H phase transition and we are interested to carry out these experiments in the future work, while the suggested characterization may not be as powerful as the presented AFM images to demonstrate the interlayer sliding.

Supplementary Fig. 24. Variation of calculated WS₂ unit cell parameter of (a) *a* and (b) *b* in the initial 2M phase (I), final 2H phase (VII), transition state and other intermediate configurations (II to VI), showing lattice shrinkages during 2M to 2H phase transition.

Fig. 5c. Snapshots of BL WS₂ molecular geometries at the initial 2M phase (coord. I), transition state (coord. IV), an intermediate configuration (coord. V) and the final 2H phase (coord. VII), viewing from the *a* direction. Interlayer spacings are denoted in different states, showing interlayer expansion from I VII. Lattice edges are depicted in different states, referring to which interlayer dislocation between top and bottom layers can be seen. Color code: blue and orange spheres represent W and S, respectively.

Revised Fig. 2i-k. AFM images and optical images (insets at top-right corners) of a piece of multilayered WS₂ flake in (i) 2M, (j) an intermediate and (k) 2H phases. Thicknesses of this flake at different phases are labeled. Edge of the WS₂ flake is denoted by white arrow, where interlayer sliding can be seen.

Supplementary Fig. 12. AFM images and optical images (insets at top-right corners) of a piece of a 6 L WS₂ flake in (a) 2M, (b) an intermediate and (c) 2H phases. Thicknesses of this flake at different phases are labeled. Edge of the WS₂ flake is denoted by white arrow, where interlayer sliding can be seen.

In the revised manuscript, the AFM images are discussed as:

AFM images (Fig. 2i-k) show the thickness of a WS₂ flake increases from 25.461 nm to 25.967 nm and 27.074 nm as it changes from 2M to an intermediate and 2H phase, further confirming the continuous change of interlayer spacing. Moreover, there is slight sliding at edge of the WS₂ flake from 2M to the intermediate and 2H phase (Fig. 2i-k and Supplementary Fig. 12), indicating generation of interlayer dislocation during the phase change. Since transforming from a 1T' to 1H layer results in in-plane lattice contraction, the interlayer dislocation can be attributed to non-concerted 1T' to 1H phase transition among different layers and the followed reconfiguration of layer stacking. The above XRD, Raman and AFM results hint that 2M to 2H phase transition of multilayered WS₂ probably takes place layer(s) by layer(s).

Changes to the manuscript: (1) Discussion about wrinkles on the WS₂ surface during phase change is removed; (2) Discussion about AFM images of WS₂ during phase change is in the manuscript on Page 13.

Comment 5: It is widely known that WS₂ of varying thicknesses have different absorption powers for lasers, thus, it is more appropriate to use a suspended substrate for thermal conductivity testing, similar to reference 34, in order to make the abscissa reference in Figure 3f meaningful.

Response: Thanks for this suggestion. We agree that WS₂ of varying thicknesses have different absorption powers. Therefore, thermal conductivity of the SiO₂/Si substrate used in our experiment affects the accuracy of the measured laser-power coefficients. To get rid of the substrate effect, we have run the temperature- and laser-power-dependent Raman spectroscopy at holey substrates where WS₂ flakes are suspended. The calculated thermal conductivities based on the as-measured results (Supplementary Fig. 16, attached below) also reveal thinner 2M (or 1T') WS₂ has higher thermal conductivity, the same trend as that obtained at the conventional SiO₂/Si substrate.

Supplementary Fig. 16. Thickness-dependent thermal conductivity of 2M WS₂ on holey Si/SiO₂ substrate. Raman spectra of (a) 4 L, (b) 10 L and (c) 16 L 2M WS₂ acquired with different-power laser excitation. Frequency of (d) B_u and (e) A_g⁷ Raman modes as functions of laser power for 4 L, 10 L and 16 L 2M WS₂. Laser-power coefficient (χ_p) of Raman frequencies are extracted from the slopes of the corresponding plots. Raman spectra of (f) 4 L, (g) 10 L and (h) 16 L 2M WS₂ acquired

at different temperatures. Frequency of (i) B_u and (j) A_g^7 Raman modes as functions of temperature for 4 L, 10 L and 16 L 2M WS_2 . Temperature coefficient (χ_T) of Raman frequencies are extracted from the slopes of the corresponding plots. (h) Thermal conductivities of 4 L, 10 L and 16 L 2M WS_2 estimated by χ_T divided by χ_p . For each sample, an average thermal conductivity is obtained based on two values calculated from three sets of χ_T and χ_p data that are extracted from B_u and A_g^7 modes plots.

Supplementary Fig. 16 is discussed in the revised manuscript as:

To deconvolute thermal conductivity of the SiO_2/Si substrate from our measurements, we also tested Raman spectra of WS_2 at holey substrates (Supplementary Fig. 16). The as-obtained results also confirm thinner 2M WS_2 has higher thermal conductivity, as shown in Supplementary Fig. 16, except that the calculated thermal conductivities at holey substrates are lower than the corresponding values at conventional SiO_2/Si substrates.

Since the phase stabilities of WS_2 samples discussed in this manuscript are all based on the conventional SiO_2/Si substrate, which is also the most widely used substrate in the field of TMD-based electronics, we keep the original Fig. 3e-h in our manuscript, but emphasize these measurements are taken at the conventional SiO_2/Si substrate in the corresponding figure caption and text. We have also added annotation in the caption of Supplementary Fig. 16 to highlight the results are measured on holey SiO_2/Si substrate:

Fig. 1. Thickness-dependent phase stability of 2M WS_2 on Si/SiO_2 substrate.

Fig. 3. Thickness-dependent intralayer bond strength and thermal conductivity of 2M WS_2 on Si/SiO_2 substrate.

Supplementary Fig. 16. Thickness-dependent thermal conductivity of 2M WS_2 on holey Si/SiO_2 substrate.

As 2M to 2H phase transition of multilayered WS_2 is a layer(s) by layer(s) slow process, thermal conductivity plays an important role in phase stability of 2M WS_2 . We then evaluate the thermal

conductivities of 2M WS₂ flakes using temperature- and laser-power-dependent Raman spectroscopy on SiO₂/Si substrate.

Changes to the manuscript: (1) Temperature- and laser-power-dependent Raman spectra of suspended WS₂ flakes are presented in Supplementary Fig. 16, which is discussed on Page 16; (2) Substrates used in various Raman measurements are explicitly defined in the revised manuscript.

Comment 6: The author's main focus in the article is WS₂, but it has been repeatedly mentioned in the introduction that the stability of 1T' phase TMDs is being studied. WS₂ is only one type of TMDs and cannot be representative of all TMDs. Thus, it is suggested that the author reorganize the introduction section to improve readability.

Response: Thanks for this good suggestion. We have reorganized the introduction section accordingly and made the discussion specifically focused on WS₂. The updated introduction section is copied below:

Phase engineering of group-VI transition metal dichalcogenides (TMDs) such as MoS₂ and WS₂ is important for acquiring novel physical and chemical properties¹⁻⁶. Depending on coordination modes between the transition metal and chalcogen atoms, these TMDs present in either trigonal prismatic coordinated semiconductive phases (2H and 3R) or octahedral coordinated metallic phases (2M, T_d, 1T, 1T', etc.)^{1-3,6}. Bulk 2M and the monolayered (ML) 1T' WS₂ have shown plenty of unique appealing properties, such as superconductivity⁷⁻¹⁰, Weyl semimetal states^{11,12}, optical nonlinearity^{13,14} and enhanced electrochemical activities¹⁵⁻¹⁹. However, the 1T' TMDs are metastable and tend to transform to the thermodynamically stable 2H phase^{1,3,4}, hence their practical applications are significantly limited^{1,3}. Although various strategies such as electron doping^{20,21}, strain effect^{22,23} and heterostructural interaction^{24,25} can be adopted to stabilize the metallic phase TMDs, their intrinsic stabilities under practical operation conditions (e.g., temperature and atmosphere) remain unclear.

As van der Waals (vdW)-bonded layered materials, physicochemical properties of TMDs can dramatically change when their structures evolve from three-dimensional bulk to ultrathin two-dimensional (2D) regime. For example, band gaps of 2H MoS₂ and WS₂ turn from indirect to direct

after thickness is reduced to monolayer (ML)²⁶. Dielectric screening in gapped 2D crystals significantly reduces with decrease of layer number, leading to greatly enhanced Coulomb interactions²⁷. The high Coulombic interactions have resulted in tightly bonded excitons and trions in ML MoS₂²⁷, as well as anormal thickness dependent Raman shifts²⁸. Atomically thin TMDs exhibit much higher electrocatalytic activities than bulk matrixes, because of “self-gating effect” induced high carrier densities^{29,30}. The in-plane inversion symmetry of bulk TMDs is broken after they are reduced to ML, which produces out-of-plane spin polarization depending on the valley (K or -K point) in momentum space². Thermal conductivity is found to decrease with increase of layer number for thin 2H MoS₂ and T_d WTe₂^{31,32}, while the opposite trend is seen for thin indium selenides^{33,34}. In a word, the layer number plays critical roles in both the electronic and phonon properties of 2D TMDs. As 2M to 2H phase transition of WS₂ (or 1T' to 1H for ML WS₂) involves changes of energy band structures of both electron and phonon, the layer number could greatly affect the stability of 2M WS₂. However, the relation between layer number and stability of 2M WS₂ is barely known to the best of our knowledge.

Here, we investigate the thickness-dependent intrinsic phase stability of mechanically exfoliated 2M WS₂ and find that thinner samples have higher thermal stabilities. 2M to 2H phase transition temperature increases from 120 °C to 210 °C in the air as thickness of WS₂ is reduced from bulk to bilayer (BL). ML WS₂ can maintain 1T' structure in the air and argon (Ar) atmosphere until temperature reaches 350 °C and 450 °C, respectively, which are about two and three times higher than phase transition temperatures of bulk 2M WS₂. Raman spectroscopy reveals thinner 2M WS₂ has more stiffened intralayer W-W and W-S bonds and higher thermal conductivity, which enables thinner sample with higher resistance to lattice deformation and higher efficiency heat dissipation under elevated temperature. Theoretical simulation and calculation further confirm thinner WS₂ has higher phase transition energy barrier, because lack of interlayer interaction in thinner WS₂ leads to higher-energy transition state lattice configuration during phase transition.

Changes to the manuscript: This content is mentioned in the manuscript on Page 3-4.

Comment 7: The photoluminescence (PL) spectrum of 2H WS₂ typically decreases in intensity as

the number of layers increases. To further confirm the phase transition of the 1T' WS₂, it is suggested that the author observe the changes in PL peaks during the phase transition process more closely.

Response: Thanks for this good suggestion. Accordingly, we did PL measurements on WS₂ flakes during heating. We observed the typical PL peak at ~ 640 nm that was attributed to 2H WS₂ (*Nano Lett.* 2013, 13, 8, 3447–3454) in the 2-4 L areas of a WS₂ flake (Fig. 1a) after they were heated to 325 °C in the air, and PL intensities were enhanced as temperature increased from 325 °C to 350 °C, as shown in the revised Supplementary Fig. 3a-c (attached below). These PL results have further confirmed 2M to 2H phase transition during heating and indicated higher temperature results in deeper extent of phase transition. PL intensity at 350 °C was also found to decrease from 2 L to 4 L areas, as shown in revised Fig. 1 b1, 6 (attached below). For monolayered WS₂ flake, PL emission was observed after it was heated to 450 °C in argon atmosphere (revised Fig. 1 c5, attached below), while it is impossible to see 1T' to 1H phase transition by heating it in the air.

Revised Supplementary Fig. 3a-c. PL spectra of (a) 2 L (b) 3 L (c) 4 L WS₂ in Fig. 1a heated at different temperatures in the air.

Revised Fig. 1b. (b1) AFM images of a piece of exfoliated 2M WS₂ flake with different thicknesses areas. (b6) PL intensities mappings of this flake at 350 °C in the air. Different thicknesses areas are outlined with dotted lines.

Revised Fig. 1c. (c1-2) Optical and (c3) AFM images of a piece of exfoliated ML 1T' WS₂ flake at room temperature and 450 °C in Ar atmosphere. (c4) E_{2g}¹ mapping and (c5) PL mapping of this ML WS₂ at 450 °C in Ar atmosphere. These mappings correspond to the dotted-line outlined area in c-1.

In the revised manuscript, the measured PL results are discussed as:

Photoluminescence (PL) peaks (Supplementary Fig. 3a-c) at ~ 640 nm emerge in 2-4 L areas as temperature reaches 325 °C, further confirming the 2M to 2H phase transitions³⁶. The PL intensities are significantly enhanced after heating to 350 °C, as shown in Supplementary Fig. 3a-c, consistent with the increase of E_{2g}¹ mode intensity at elevated temperature. These results indicate 2M to 2H phase transition is a slow process, and the extent of phase transition augments at elevated temperatures or after longer time of heating. PL intensity of WS₂ is sensitive to thickness²⁶, and the distribution of PL intensity (Fig. 1 b6) agrees to the AFM defined areas of various thicknesses from BL to 4 L. Comparatively, the ML WS₂ area preserves the typical 1T' Raman pattern without emergence of E_{2g}¹ mode or PL until temperature reaches 350 °C (Fig. 1b and Supplementary Fig. 3d-e), evidencing the highest thermal stability among the four WS₂ areas. In agreement with this result, chemically exfoliated ML 1T' WS₂ is also demonstrated to have good air-stability at room temperature^{37,38}. Heated at 350 °C in the air, Raman signal of the 1T' WS₂ fades away and the 1T' WS₂ flake disappears, seen from Fig. 1 a6, which can be attributed to decomposition due to air oxidation. This means the decomposition temperature of ML 1T' WS₂ in the air is lower than 1T' to 1H phase transition temperature and it is impossible to observe 1T' to 1H phase transition by heating ML WS₂ in the air. After moving to an inert Ar atmosphere, 1T' to 1H phase transition temperature of

ML WS₂ is measured to be as high as 450 °C (Fig. 1 c1-3), confirmed by emergences of E_{2g}¹ mode (Fig. 1 c4) and PL (Fig. 1 c5). These results strongly indicate thinner 2M WS₂ has higher phase stability.

Changes to the manuscript: This content is mentioned in the manuscript on Page 6.

Comment 8: How can the author's statement on page 15, line 316 regarding the layer-by-layer phase transition of WS₂ of BL and TL be demonstrated through experiments?

Response: We agree it is important to combine theoretical calculation and experiment to solidify the proposed layer(s)-by-layer(s) phase transition mechanism. To experimentally demonstrate this mechanism, we have employed Raman, XRD and AFM characterizations on intermediate phases WS₂ samples. In our experiment, we found 2M to 2H phase transition of WS₂ is a slow process. Different intermediate phases WS₂ could be obtained by heating the 2M WS₂ above phase transition temperature within controlled times. All the Raman (Supplementary Fig. 4-6) and XRD spectra (revised Fig. 2g-h, attached below) of intermediate phases WS₂ exhibit superpositions of 2M and 2H patterns, indicating the intermediate phases of WS₂ are 1T'/1H heterostructures.

Prolonging the heating time for bulk 2M WS₂, XRD peaks attributed to 2M (200) diffractions are weakening and broadening, while the ones attributed to 2H (002) diffractions are intensifying and sharpening, and all these peaks are shifting to lower degrees but locating in between that of standard 2M (200) and 2H (002) diffractions, as shown in the revised Fig. 2g-h. These results indicate ordered stacks of 1T' and 1H layers are broken and built, respectively, with increase of phase change extent. An intermediate phase WS₂ has hybrid low-order stacks of 1T' and 1H layers with average interlayer spacing in between that of pure 2M and 2H phase. In agreement with this result, Raman spectra (Supplementary Fig. 10, attached below) of intermediate phases WS₂ manifests that A_{1g} mode (ascribed to out-of-plane vibration of 1H layer stacks) locates at lower wave number compared to 2H phase and keeps to blue-shift as the extent of 2M to 2H phase change increases.

AFM images (revised Fig. 2i-k) confirm interlayer sliding during the phase change, which connects non-concerted 1T' to 1H phase changes among different layers, as discussed in the Response to Comment 4. We further punched an intermediate phase WS₂ flake using a probe to

expose inner layers at edge areas. We have tested Raman spectra and found that the exposed inner layers exhibit significantly higher intensities of 2H A_{1g} and E_{2g}^1 modes than the top layers, as shown in the revised Fig. 2l-m (attached below). This result means that inner layers of the intermediate phase WS_2 experience deeper extent 1T' to 1H phase transition than the top layers, strongly indicating 2M to 2H phase transition of multilayered WS_2 takes place layer(s) by layer(s).

Revised Fig. 2g-h. (g) Powder XRD patterns of 2M, intermediate phases and 2H WS_2 . 2H and intermediate phases WS_2 were obtained by heating 2M WS_2 in the air at 250 °C for 20 min and at 130 °C for various times, respectively. (h) A zoomed-in view of the XRD patterns in the range of 13.5°~16.0°.

Supplementary Fig. 10. Optical and AFM images and Raman spectra of a piece of multilayered WS₂ flake in 2M, intermediate and 2H phases. 2H and intermediate phases WS₂ were obtained by heating the 2M WS₂ in the air at 250 °C for 20 min and at 120 °C for various times, respectively. A_{1g} mode exhibits blue shifts as the extent of 2M to 2H phase change increases.

Revised Fig. 2l-m. (l) Raman spectra of an intermediate phase multilayered WS₂ flake measured from the top layers (black line) and the exposed inner layers (red lines) after punching with a probe. Insets show optical images of the pristine WS₂ flake (left) and the one after punching (right). Raman measured areas are defined by squares. (m) E_{2g}¹ Raman mode intensity mapping of the probe punched WS₂ flake in l. Red squares label the same areas with that in right inset of l.

Experimental results which support the layer(s)-by-layer(s) 2M to 2H phase transition mechanism are discussed in the revised manuscript as:

The slowness of 2M to 2H phase transition of WS₂ allows us to trace this process by investigating structures of intermediate phases after varied extents of 2M to 2H phase change. Intermediate phases WS₂ can be obtained by heating the 2M WS₂ above phase transition temperature within controlled time. As shown in Supplementary Fig. 4-6, all intermediate phases WS₂ flakes show superpositions of 2M and 2H characteristic Raman modes, indicating the intermediate phases of WS₂ are probably 1T'/1H heterostructures that can involve in-plane heterostructured areas and hybrid stacked layers.

To investigate layer stacking configurations of intermediate phases WS₂, XRD has been carried out on bulk 2M WS₂ powder during heating at 130 °C. With prolongation of heating time, XRD

peaks attributed to 2M (200) diffractions are weakening and broadening, while the ones attributed to 2H (002) diffractions are intensifying and sharpening, and all these peaks are shifting to lower degrees but locating in between that of standard 2M (200) and 2H (002) diffractions, as shown in Fig. 2g-h. These results indicate ordered stacks of 1T' and 1H layers are broken and built, respectively, with increase of phase change extent. An intermediate phase WS₂ has hybrid low-order stacks of 1T' and 1H layers with average interlayer spacing in between that of pure 2M and 2H phase. In agreement with this result, Raman spectra (Supplementary Fig. 10) of intermediate phases WS₂ manifests that A_{1g} mode (ascribed to out-of-plane vibration of 1H layer stacks) locates at lower wave number compared to 2H phase and keeps to blue-shift as the extent of 2M to 2H phase change increases. High-index XRD diffractions including 2M (11 $\bar{1}$) (002) (31 $\bar{1}$) (601) (311) are also disappearing with increasing of heating time, with 11 $\bar{1}$ diffraction shifting to higher degrees that can be attributed to in-plane atomic gliding and out-of-plane displacements, as shown in Supplementary Fig. 11. AFM images (Fig. 2i-k) show the thickness of a WS₂ flake increases from 25.461 nm to 25.967 nm and 27.074 nm as it changes from 2M to an intermediate and 2H phase, further confirming the continuous change of interlayer spacing. Moreover, there is slight sliding at edge of the WS₂ flake from 2M to the intermediate and 2H phase (Fig. 2i-k and Supplementary Fig. 12), indicating generation of interlayer dislocation during the phase change. Since transforming from a 1T' to 1H layer results in in-plane lattice contraction, the interlayer dislocation can be attributed to non-concerted 1T' to 1H phase transition among different layers and the followed reconfiguration of layer stacking. The above XRD, Raman and AFM results hint that 2M to 2H phase transition of multilayered WS₂ probably takes place layer(s) by layer(s). To further confirm this mechanism, we tested Raman spectra on an intermediate phase multilayered WS₂ flake before and after punching with a probe. As shown in Fig. 2l-m, the exposed inner layers after punching (areas marked by red squares) exhibit significantly higher intensities of 2H A_{1g} and E_{2g}¹ modes than top layers of the WS₂ flake, which strongly evidences non-concerted phase transition among different layers.

Changes to the manuscript: This content is mentioned in the manuscript on Page 12-13.

Referee: 2

Comments: *This manuscript reports on and investigates the higher phase stability of thin 1T'-WS₂ as compared to bulk. The result is noteworthy, and I think the core message is likely correct, I however have quite a few doubts about some specific conclusions. I further find the manuscript hard to follow and in large parts confusing, especially the later sections. I thus, at this stage, cannot recommend publication. I would be happy to look at the manuscript again after the authors addressed the following concerns. In general, for reviewing it would help if the figures are not placed at the end of the manuscript. Scrolling back and forth and trying to understand the phase transformation process was very tedious. If the editors agree, I would encourage the authors to include figures in the text when submitting the revised version.*

Response: Thanks for rating our results as noteworthy and happy to know the reviewer would like to look at the revised manuscript. We have carefully addressed all the confusing issues, especially for the sample purity and spontaneous phase transition. Details can be found below. As the reviewer suggested, we have also attached the figures in the text to make the revised manuscript more readable.

Comment 1: *I have strong doubts about the claim that bulk 1T'-WS₂ (which by the way is usually called 2M-WS₂ in the literature, although I don't insist to name it this way) will transform to 2H by itself without heating or laser power. My lab has worked with this phase for a while, and we have a lot of experience handling bulk crystal or powder. We never observed any phase transformation, not even partial, when the crystals, or powder were stored in either air or argon. We have, just as the authors report, observed the phase transformation with heating or with too high laser power when performing Raman experiments. I thus have a very hard time believing the authors report of a spontaneous phase transition that does not require any activation energy. I wonder if the Raman spectra of the samples that were measured after different times (i.e Fig S3) were always measured at the same sample spot. It could be that samples are locally different and thus they appear to start transforming. Also, the 0 days Raman spectrum of the 23K sample (Fig. S3 d) looks like it has a bad signal to noise ratio, maybe some 2H phase is already present. We have noticed before in bulk*

samples, that sometimes a phase mixture is present in fresh samples and the Raman spectrum can depend on the location of the measurement.

Response: We agree that it is appropriate to name the multilayered metastable phase as 2M and the monolayered one as 1T'. We have changed those names accordingly in the revised manuscript. Many thanks for pointing out the critical issue about spontaneous phase transition in the air. We appreciate that the reviewer provides an important clue that the 2H phase may already exist in the 2M WS₂ sample. We carefully checked the bulk WS₂ samples used to investigate the thickness dependent durability and found that these samples had indeed experienced 2M to 2H phase change. Both Raman and XRD spectra of these samples involve characteristic peaks of 2H phase WS₂, as shown in the Responsive Fig. 1 (attached below), meaning that there were 2H phase components present in these 2M WS₂ samples. These samples were stored in the air at room temperature for about 10 months. As a result, the data of Fig. 1f in our previous version manuscript is not reliable. Within the last three months, we prepared new pure 2M WS₂ crystals and exfoliated them into various thin flakes and placed them in the air at room temperature. Until the day we submitted this manuscript, no 2M to 2H phase change was observed for all those invested WS₂ flakes. Based on the above results, we can conclude that 2M to 2H phase transition of WS₂ can spontaneously occur in the air at room temperature, but this process is significantly slower than what we reported in previous version manuscript. At this moment, we don't have data to precisely describe the durability of 2M WS₂ in the air, so we have removed the related figures and discussions from the revised manuscript. We are interested in continuing the regarded investigation in our future work.

Responsive Fig. 1. (a) Raman spectra and (b) power XRD pattern (blue line) of the bulk 2M WS₂ used to investigate the thickness dependent durability in our previous version manuscript. E_{2g}¹ Raman mode emerges at point 2# and 4#. XRD pattern of this WS₂ sample involves peaks attributed to 2H (002) and 2M (200) diffractions.

The reason for 2M to 2H phase change in the air at room temperature is that air oxidation can remove the surface extra electrons from the 2M WS₂, resulting in reduction of stability and accelerated 2M to 2H phase change. The extra surface electrons can be well preserved in an inert argon atmosphere, so 2M WS₂ is more stable in the argon atmosphere. Our collaborators in another lab found a batch of their 2M WS₂ sample that had been stored in the argon atmosphere at room temperature for 12 months did not change to 2H phase. In our experiment, we have observed that 2M to 2H phase transition temperature is higher in the argon atmosphere than in the air, as shown in the revised Fig. 1d. Details about air-oxidation effect are discussed in the Response to Comment 4.

To address the concern whether repeated Raman measurements at the same spot of a WS₂ flake could induce 2M to 2H phase, we measured Raman spectra on WS₂ flakes by controlling power and time of laser irradiation. A 13 L 2M WS₂, which transforms to 2H phase under 7 mW laser irradiation, does not transform to 2H phase under a laser with lower power, even the irradiation time is prolonged to 180 s, as shown in the Responsive Fig. 2 (attached below). When we measured Raman spectra in the experiment about durability of 2M WS₂, we used a 0.8 mW laser that was absolutely not enough to activate phase change, and we set the irradiation time to be 20 s each time, repeating of which for up to 9 times would equal to 180 s of total irradiation time. However, we rarely measure Raman at the same spot for that many times in our experiments. Therefore, the observed spontaneous 2M to 2H phase change in the air was not caused by repeated Raman measurements at the same spot of a WS₂ flake.

Responsive Fig. 2. (a) Raman spectra of 13 L 2M WS₂ measured using various powers of incident laser. The total time of laser irradiation in each measurement is 20s. The 7 mW laser is capable to activate 2M to 2H phase transition, as the E_{2g}¹ Raman mode emerge. (b) Raman spectra of 13 L 2M WS₂ measured using 6 mW laser with various total times of laser irradiation. 2M to 2H phase transition is not activated with the time increasing from 20 s to 180 s.

In the revised Fig. 1, we add Fig. 1f (attached below) to present powers of laser required to activate 2M to 2H phase transition for different thicknessed WS₂ in the air, and this figure is discussed in the manuscript as:

The thinner 2M WS₂ can also withstand higher power of incident laser during Raman measurements (Fig. 1f and Supplementary Fig. 8). From 25 L to 5 L WS₂, the power of laser required to activate the 2M to 2H phase transition increases from 5.5 mW to 30 mW. A 100-mW laser, which is the maximum output of the used instrument, can turn a 4 L WS₂ to 2H phase, but is not able to activate the phase transition for ML 1T' and BL and TL 2M WS₂ samples.

Revised Fig. 1f. The power of laser required to activate 2M to 2H phase transition as a function of WS₂ layer thickness measured in the air.

Changes to the manuscript: (1) Figures and discussions about durability of WS₂ at room temperature are removed from the manuscript; (2) The power of laser required to activate 2M to 2H phase transition is included in the revised Fig. 1f and Supplementary Fig. 8, which is discussed in the manuscript on Page 9.

Comment 2: In conjunction with this I have some doubt about the proposed mechanism. I wonder if the higher stability stems from impurities in bulk samples which grow quickly, catalyzing the phase transformation. The authors mostly use Raman spectroscopy to analyze their samples, Raman can easily miss sample imperfections, small impurities etc. High-resolution powder diffraction together with Rietveld refinement might give more information. In our lab we found for example, that higher purity bulk samples can be heated to higher temperatures (~ 230 °C) before they transform as compared to low quality bulk samples, which transform at 180 °C (according to DSC).

Response: Thanks for the suggestion of investing in the purity of WS₂ sample. We carried out Rietveld refinement (Supplementary Fig. 1 and Supplementary Table 1, attached below), attached below) based on powder XRD of the synthesized 2M WS₂, and deduced the C_{2m} space group monoclinic structure with lattice parameters in good agreement with previously reported 2M WS₂

structure (*Nat. Mater.*, 2021, 20, 1113–1120). X-ray photoelectron spectroscopy (XPS) (Supplementary Fig. 2, attached below) confirms the synthesized 2M WS₂ does not have impurity elements. The updated TEM image (Fig. 2d) of 2M WS₂ shows obvious zigzag chains of W atoms, in consistency with the reported structure of 2M WS₂ (*Nat. Mater.*, 2021, 20, 1113–1120). These results strongly evidence the high purity of our synthesized 2M WS₂ sample. Additionally, Zhang group also reported the 2M to 2H phase transition temperature of bulk WS₂ in the air was ~117.3 °C (*Nat. Mater.*, 2021, 20, 1113–1120), in agreement with our result. The reviewer claims 2M WS₂ can withstand a temperature up to 180 °C or even 230 °C. We guess the 2M WS₂ samples were probably placed in an inert atmosphere in the mentioned experiments. We have observed that 2M to 2H phase transition temperature of WS₂ can be above 180 °C in argon atmosphere, as shown in the revised Fig. 1e. Additionally, sulfur vacancy, if existing in our WS₂ sample, is believed to serve as electron donor and stabilize the 1T' or 2M structure (*J. Mater. Chem. A*, 2018, 6, 23932–23977). 1T/1H heterostructure is also believed to be more stable than the pure 1T structure (*ACS Nano* 2018, 12, 12080–12088; *J. Mater. Chem. A*, 2018, 6, 23932–23977). In our experiment, we also found intermediate phases WS₂ that contained 1T'/1H heterostructures were quite stable, without changing of the Raman spectra for several months in the air at room temperature. To sum up, 2M to 2H phase transitions observed in our experiments are not likely induced by impurities or defects.

Supplementary Fig. 1. Rietveld refinement of powder XRD pattern in $C_{2/m}$ space group of 2M WS₂.

Supplementary Table 1. Structural parameters obtained from Rietveld refinements of the powder XRD of 2M WS₂.

R values	$R_p=10.16$	R_{wp}	R_{exp}	
	10.16	13.03	7.27	
Lattice parameters	a (Å)	b (Å)	c (Å)	β (°)
	12.8471	3.2177	5.6912	112.8368
Atom sites	x	y	z	
W	0.75566	0.50000	0.79548	
S1	0.13880	0.00000	0.32374	
S2	0.39484	0.00000	0.21906	

Supplementary Fig. 2. XPS survey of synthesized 2M WS₂.

The high purity of our synthesized 2M WS₂ sample is discussed as:

Rietveld refinement based on powder X-ray diffraction (XRD) (Supplementary Fig. 1 and Supplementary Table 1) reveals the synthesized WS₂ has monoclinic structure in C_{2/m} space group with cell parameters $a = 12.8417 \text{ \AA}$, $b = 3.2177 \text{ \AA}$, $c = 5.6912 \text{ \AA}$ and $\beta = 112.8368^\circ$, which is in excellent agreement with the previously reported 2M WS₂ structure^{3,7}. X-ray photoelectron spectroscopy (XPS) (Supplementary Fig. 2) further evidences the synthesized sample is composed of only W and S elements. All these results confirm the high purity of the synthesized 2M WS₂.

Changes to the manuscript: Rietveld refinement of powder XRD pattern and XPS survey are added in supporting information as Supplementary Fig. 1 and Supplementary Table 1 and Supplementary Fig. 2, respectively, which are discussed in the manuscript on Page 5.

Comment 3: I find the entire theory section highly confusing and would suggest a re-write. Maybe I understand the proposed mechanism better after. It is still not clear to me what exactly the different phases (I, II, etc) are and where they come from. Did DFT suggest these as intermediates? Did the authors suggest them and then calculate them? I'm sorry but this part was very hard to follow.

Response: Thanks for this good comment. We are aware that descriptions of the calculated states in our previous version manuscript are not accurate. The simulation of WS₂ lattices and calculation of free energies are based on the transition state theory, similar to that applied in simulation of chemical reactions. This calculation can identify the lowest-energy transition state molecular configuration, passing by which the WS₂ lattice evolves from 2M to 2H phase. Based on this, the theoretical simulation finally gives a path for 2M to 2H phase transition. Such a path consists of a series of phase transition coordinates, including initial 2M (or 1T'), final 2H (or 1H), transition state and other intermediate lattice configurations. We have changed the title of x-axis of Fig. 4 (attached below) to phase transition coordinate and named the data points as coord. I, coord. II, coord. III, and the like. Coord. I and VII correspond to 2M and 2H WS₂ lattices, respectively, of which structural parameters are known and used as the calculation base. The energy vertex coordinates, i.e., coord. III for monolayered and trilayered WS₂ and coord. IV for bilayered WS₂, are transition states. Others are

intermediate lattice configurations. In another word, except for the initial 2M (coord. I) and the final 2H structures (coord. VII), all the other coordinates are output by DFT calculations.

Revised Fig. 4. Energy profiles of 2M to 2H phase transition. The calculated energies of ML, BL and TL WS₂ lattices per unit cell in the initial 2M (or 1T') phase (coord. I), transition states (coord. III for ML and TL WS₂ and coord. IV for BL WS₂), final 2H (or 1H) phase (coord. VII) and other intermediate configurations. E_a^{ML} , E_a^{BL} and E_a^{TL} are estimated energy barriers of 2M to 2H (1T' to 1H) phase transitions for ML, BL and TL WS₂. E_a^{ML} , E_a^{BL} and E_a^{TL} are calculated to be 0.919 eV, 0.471 eV and 0.309 eV, respectively.

To make the theory section more readable, we have rewritten it as:

To study thickness-dependent phase transition kinetics, we have simulated the 2M to 2H phase change processes for rectangular ML, BL and TL WS₂ supercells by employing Vienna ab initio simulation package (VASP) and calculated energies of WS₂ lattices with different configurations by using climbing image nudged elastic band (CI-NEB) method. Like the transition state theory applied for chemical reactions, these calculations can find the transition state WS₂ lattice configuration during 2M to 2H phase transition, and identify the path passing through the transition state and connecting the initial 2M phase and the final 2H phase. Such a path consists of a series of phase transition coordinates, as shown in Fig. 4, including initial 2M (or 1T'), final 2H (or 1H), transition

state and other intermediate lattice configurations. Method and principle used to find these coordinates are discussed in Part III of supporting information with Supplementary Fig. 18. The energy difference between lattices of transition state and the initial 2M phase is the energy barrier for 2M to 2H phase transition. As shown in Fig. 4, ML WS₂ has an energy barrier of 0.919 eV, which is apparently higher than 0.471 eV of BL WS₂ and 0.309 eV of TL WS₂, theoretically confirming thinner 2M WS₂ has higher phase stability. Configurations of all WS₂ lattices at the as-calculated coordinates (coord. I to coord. VII) are shown in Supplementary Fig. 19-23. During the simulated phase transition process, the octahedral coordinated 2M lattices gradually deform until reach the transition states, which further evolve and relax to the trigonal prismatic coordinated 2H lattices. Meanwhile, the WS₂ unit cell shrinks in *a* and *b* directions (Supplementary Fig. 24), which agrees with the afore-discussed phase transition mechanism.

In Part III of the revised supporting information, we describe the principle of simulations as:

Theoretical simulation of 2M to 2H phase transition of WS₂ is based on the transition state theory². The initial 2M state can reach the final 2H state through many different pathways on the potential energy surface (PES), as shown in Supplementary Fig. 18. On any pathway connecting initial and final states on the PES, there will be a state corresponding to the energy maximum (EM). The transition state has an energy equal to the lowest EM. Therefore, the transition state can be located by finding the lowest EM on the connection pathways as shown in Supplementary Fig. 18. To locate the transition state from 2H to 2M, the climbing image nudged elastic band (CI-NEB) method was adopted³. Five images were used to build the elastic band between the given initial and final states along phase transition coordinates path. The energy and force convergence criteria were set to 10⁻⁵ eV and 0.05 eV/Å, respectively.

[REDACTED]

Supplementary Fig. 18. Illustration of different pathways connecting the initial state and the final state on a potential energy surface (PES), and the connection pathways and the corresponding energy maximum (EM) points are plotted in green lines and red dots, respectively. The EM point of pathway 3 has the lowest energy among all EM points of connection pathways, corresponding to the exact transition state (TS) connecting the initial state and the final state. (*J. Chem. Theory Comput.*, 2022, 18, 8, 5108–5115)

Changes to the manuscript: (1) The title of x-axis of Fig. 4 is changed to phase transition coordinate. (2) The revised theory section is in the manuscript on Page 17-18. (3) Supplementary Fig. 18 is added into supporting information to illustrate the principle of transition-state theoretical calculation. (4) Part III of the revised supporting information is modified.

Comment 4: I'm also confused about the oxidation discussion. How does oxidation speed up or cause the phase transition? The 1T'-2H phase transition does not involve any change of oxidation state, so it has nothing to do with oxidation. If oxidizing environmental effects speed it up, I would argue even more that the phase transformation is impurity driven and that the proposed mechanism about weaker vdW interactions is then false.

Response: 2M to 2H phase transition does not involve change of stoichiometry but does involve change of oxidation state. It has been demonstrated that W (or Mo) element in 2M WS₂ (or MoS₂)

has lower oxidation state than that in 2H WS₂ (or MoS₂), because 2M WS₂ (or MoS₂) possesses extra surface electrons (*Nat. Mater.*, 2021, 20, 1113–1120; *Nat. Chem.*, 2018, 10, 638–643). Previous reports also have proved that electron donors can stabilize 1T' WS₂ and MoS₂, while electron acceptors speed up 1T' to 2H phase transition (*Nat. Nanotech.*, 2014, 9, 391-396); *Nat. Mater.*, 2014, 13, 1128-1134); *Sci. Rep.*, 2017, 7, 3836; *J. Mater. Chem. A*, 2018, 6, 23932–23977). Sasaki group found that +3 Mo contained in chemical exfoliated 1T' MoS₂ was gradually oxidized to +4 when the 1T' MoS₂ was aged in the air (*Dalton Trans.*, 2018, 47, 3014–3021). To investigate the air-oxidation effect on the 2M to 2H phase transition, we analyzed XPS spectra of 2M, 2H and intermediate phases WS₂. The intermediate phases WS₂ were obtained by heating the 2M WS₂ in the air at 130 °C for different times. As shown in Supplementary Fig. 7 (attached below), W 4f peak and valence band edge shift to higher binding energies, as the extent of 2M to 2H phase transition gets larger. This result indicates the oxidation state of W increases during 2M to 2H phase change. Meanwhile, O 1s XPS peak (Supplementary Fig. 7, attached below) does not change during 2M to 2H phase transition, indicating there is no oxide species formed during the phase change, which does not support the conjecture about impurities induced phase change.

Supplementary Fig. 7. High-resolution XPS spectra of (a) W 4f and 5p (b) valence band and (c) O 1s of 2M, intermediate and 2H phases bulk WS₂. Intermediate phases WS₂ were obtained by heating 2M WS₂ in the air at 130 °C for 15 min, 30 min, and 45 min, respectively. 2H phases WS₂ were obtained by heating 2M WS₂ in the air at 250 °C for 20 min.

The air-oxidation effect on the 2M to 2H phase transition is discussed as:

Higher 2M to 2H (or 1T' to 1H) phase transition temperature in the inert atmosphere can be attributed to deconvolution of air oxidation effect. It is demonstrated that donation of electrons helps to stabilize 1T' TMDs, while extraction of electrons promotes 1T' to 1H phase transition^{20,21,39-41}. High-resolution XPS (Supplementary Fig. 7a-b) shows that W 4f peak and valance band edge of 2M WS₂ locate about 1.1 eV and 0.85 eV lower than that of 2H WS₂, respectively, indicating W element in 2M WS₂ has lower oxidation state than that in 2H WS₂. This is believed to be caused by existing extra electrons at 2M WS₂ surface^{3,6}. After 2M to 2H phase transition is activated by heating in the air, W 4f peak and valance band edge are observed to shift to higher energies, and higher extent of phase transition results in larger range of shifting, as shown in Supplementary Fig. 7a-b. Nevertheless, the O 1s peak (Supplementary Fig. 7c) barely changes during the 2M to 2H phase transition. These results indicate oxygen molecules speed up phase transition by extracting electrons from 2M WS₂ during heating in the air, without chemically oxidizing the WS₂.

Changes to the manuscript: This content is mentioned in the manuscript on Page 8-9.

Comment 5: I'm also confused about Fig 2. I think this figure needs to be revised. These are the things that confuse me:

(i) In the HRTEM images which zone axis are we looking at? Can it be labeled? I assume we look at a layer (ie. plan-view)? It would be good to clarify. And if we have plan view samples why is the resolution of Fig 2d so bad? I have seen many images of 1T'-WS₂ phases where the W-W zig-zag chains are very clearly resolved.

(ii) Fig 2h-j: what are the numbers (ie 25 nm etc) refer to? They are neither explained in the caption or text or I did not find the explanation.

(iii) I'm generally confused by the discussion of the intermediate phase, especially in respect to the layer distance. In the XRD, it is clear that the intermediate phase is a mixture of 2H and 1T'. There are clearly 2 peaks which overlap, each representing one stacking distance. Note the authors keep discussing the intermediate phase as if it would be its own phase, with an average layer distance. I don't think this is correct. Fig. 2k makes no sense, therefore. Also why does the intermediate phase have additional peaks? One appears clerklly at ~ 34 degrees, but more minor ones are also visible.

I'm also confused about the oxidation discussion. How does oxidation speed up or cause the phase transition? The 1T'-2H phase transition does not involve any change of oxidation state, so it has nothing to do with oxidation. If oxidizing environmental effects speed it up, I would argue even more that the phase transformation is impurity driven and that the proposed mechanism about weaker vdW interactions is then false.

Response: We have resolved the issues by modifying the Fig. 2d-m and the corresponding captions and texts mentioning this figure. The updated Fig. 2d-m is attached below:

Revised Fig. 2d-m. HRTEM images and the corresponding Fourier transforms of (d) 2M and (e) an intermediate phase WS₂. (f) Zoomed-in view and Fourier transform of area ② in e. (g) Powder XRD patterns of 2M, intermediate phases and 2H WS₂. 2H and intermediate phases WS₂ were obtained by heating 2M WS₂ in the air at 250 °C for 20 min and at 130 °C for various times, respectively. (h) A zoomed-in view of the XRD patterns in the range of 13.5°~16.0°. AFM images and optical images (insets at top-right corners) of a piece of multilayered WS₂ flake in (i) 2M, (j) an intermediate and (k) 2H phases. Thicknesses of this flake at different phases are labeled. Edge of the WS₂ flake is denoted by white arrow, where interlayer sliding can be seen. (l) Raman spectra of an intermediate phase multilayered WS₂ flake measured from the top layers (black line) and the exposed inner layers (red lines) after punching with a probe. Insets show optical images of the pristine WS₂ flake (left) and the one after punching (right). Raman measured areas are defined by squares. (m) E¹_{2g} Raman mode intensity mapping of the probe punched WS₂ flake in l. Red squares label the same areas with that in right inset of l.

(i) Fig. 2d corresponds to the bc plane of 2M WS₂, which is looked at from the a axis. Fig. 2f corresponds to the ab plane of 2H WS₂, which is looked at from the c axis. Lattice orientations have been labelled in the revised Fig. 2d and 2f. Additionally, Fig. 2d is updated with a new TEM image, where the W-W zig-zag chains of 2M WS₂ can be clearly seen.

(ii) The numbers labeled in Fig. 2h-j (Fig. 2i-k in the revised Fig. 2) are the thicknesses of the corresponding WS₂ flakes. We have explained these data in the revised figure caption and discussed them in the revised manuscript. In addition, we have removed the calculated thicknesses of per layer of different phases WS₂ (previous version Fig. 2k) and the associated text in the revised manuscript since these data are reporting similar information with the labeled thicknesses in the Fig. 2h-j (Fig. 2i-k in the revised Fig. 2).

(iii) Thanks for pointing out the confusing concept of “intermediate phase” in our previous version manuscript. To be precise, the intermediate phase WS₂ is not a mixture of 2H and 2M, but a heterostructure of 1T' and 1H. HRTEM images (revised Fig. 2e-f, attached above) show that an intermediate phase WS₂ has an area of distorted 1T' structure (Area ① in Fig. 2e) and an area of 1H structure (Area ② in Fig. 2f). Powder XRD (revised Fig. 2g-h, attached above) demonstrates intermediate phases WS₂ have 1T'/1H hybrid stacked layers and interlayer spacing is in between that of 2M and 2H phases WS₂. These results confirm intermediate phases WS₂ are heterostructures of 1T' and 1H, rather than physical mixture of them.

XRD of intermediate phases WS₂ does not have additional peaks compared to 2M and 2H WS₂. The peak at $\sim 34^\circ$ (actually 32.0°) can be attributed to $(11\bar{1})$ diffraction of 2M WS₂ (at 31.9°), except with a little shift to higher degree, as shown in Supplementary Fig. 11 (attached below). The peak of $(11\bar{1})$ diffraction is hard to observe in the XRD spectrum of pure 2M WS₂, because the (200) diffraction peaks are too strong since most of WS₂ flakes are lying along bc plane. These main peaks are significantly weakened in the intermediate phase WS₂ because of hybrid stacks of 1T' and 1H layers. Therefore, $(11\bar{1})$ peak turns manifest in some of the intermediate phases WS₂. Similar interoperation work for other small XRD peaks, including (002) $(31\bar{1})$ (601) (311) diffractions.

Additionally, we confirm that air-oxidation speeds up 2M to 2H phase transition, but this is not driven by impurity. Please find detailed discussion about air-oxidation effect in the Response to Comment 4.

Supplementary Fig. 11. A zoomed-in view of Fig. 2g in the range of $31^\circ \sim 43^\circ$.

The aforementioned HRTEM and XRD results about intermediate phases WS_2 are discussed in detail in the revised manuscript:

Fig. 2d-f show high-resolution transmission electron microscopy (HRTEM) images of W-atom configurations in 2M and an intermediate phase. Zigzag chains of W atoms are observed in Fig. 2d, in consistency with the reported structure of 2M WS_2 .³ Area ① of the intermediate phase WS_2 (Fig. 2e) reveals W-atom lines distances ranging from 0.26 to 0.31 nm, which are in between d_1 (0.23 nm) and d_2 (0.34 nm) of 2M WS_2 (Fig. 2d). Fourier transform of Area ① displays irregular quadrilaterals (inset of Fig. 2e) that are distorted from the rectangular pattern of 2M WS_2 (inset of Fig. 2d). Area ② of the intermediate phase WS_2 has a W-atom lines distance of 0.27 nm (Fig. 2f) and hexagonal pattern Fourier transform (inset of Fig. 2f), which are the same as 2H WS_2 .⁴² These results demonstrate that an intermediate phase WS_2 has an in-plane heterostructure of distorted 1T' and 1H, which is formed due to gliding of W atoms. To investigate layer stacking configurations of intermediate phases WS_2 , XRD has been carried out on bulk 2M WS_2 powder during heating at 130 °C. With prolongation of heating time, XRD peaks attributed to 2M (200) diffractions are

weakening and broadening, while the ones attributed to 2H (002) diffractions are intensifying and sharpening, and all these peaks are shifting to lower degrees but locating in between that of standard 2M (200) and 2H (002) diffractions, as shown in Fig. 2g-h. These results indicate ordered stacks of 1T' and 1H layers are broken and built, respectively, with increase of phase change extent. An intermediate phase WS₂ has hybrid low-order stacks of 1T' and 1H layers with average interlayer spacing in between that of pure 2M and 2H phase. In agreement with this result, Raman spectra (Supplementary Fig. 10) of intermediate phases WS₂ manifests that A_{1g} mode (ascribed to out-of-plane vibration of 1H layer stacks) locates at lower wave number compared to 2H phase and keeps to blue-shift as the extent of 2M to 2H phase change increases. High-index XRD diffractions including 2M (11 $\bar{1}$) (002) (31 $\bar{1}$) (601) (311) are also disappearing with increasing of heating time, with 11 $\bar{1}$ diffraction shifting to higher degrees that can be attributed to in-plane atomic gliding and out-of-plane displacements, as shown in Supplementary Fig. 11.

Changes to the manuscript: Fig. 2 is revised and is discussed in the manuscript on Page 12-13.

Comment 6: In relation to above, I'm confused by the discussion about the intermediate phase in the text as well. How was the layer distance calculated? Why is the stacking peak in the x-ray called 200 if stacking is along b? shouldn't it be 020?

Response: As discussed in the Response to Comment 5, the intermediate phases are 1T'/1H heterostructures. Fig. 2k in our previous version manuscript (attached below) shows average thicknesses per WS₂ layer of different phases. These data were obtained in this way: i) Thickness of per 2H WS₂ layer was assumed to be 0.90 nm, according to previously reported AFM data (*Nat. Comm.*, 2015, 6, 8569; *Appl. Surf. Sci.*, 2020, 533, 147479). Since number of layers was constant for the measured WS₂ flake in different phases shown in Fig. 2i-k, based on AFM-measured total thicknesses of the flake and the thickness of per 2H WS₂ layer, we could calculate the average per-layer thickness of 2M and the intermediate phase WS₂. The calculated per-layer thickness of intermediate phase WS₂ is in between that of 2M and 2H WS₂. This result is also strongly supported by XRD (revised Fig. 2g-h) of intermediate phases WS₂, as discussed in the Response to Comment 5.

In short, XRD of intermediate phases WS₂ exhibit broadened 2M (200) and 2H (002) diffraction peaks, while all these peaks locate in between that of standard 2M and 2H diffractions.

Previous version Fig. 2k. Calculated thickness per layer of 2M, an intermediate phase and 2H WS₂ based on their AFM images in previous version Fig. 2h-j.

Changes to the manuscript: The figure that presents the calculated thicknesses of per layer of different phases is removed from the revised manuscript.

Comment 7: Minor points:

(i) Please plot all Raman spectra on the same x-axis (i.e. Figure S3 plots some panels to 500 and other to 450 wavenumbers which makes comparing the panels difficult).

Response: X-axis ranges of all Raman spectra in the revised manuscript have been changed to from 80 to 450 cm⁻¹.

(ii) In the end of the conclusion the authors generalize that the 1T' phase is more stable in thin TMDs. I would not make this claim, it might be very different in WTe₂ or other TMDs. In the case of WTe₂ the monolayer is known to be MUCH more air sensitive than the bulk (although neither does transform to 2H).

Response: Thanks for this good suggestion. We have modified the “conclusion” accordingly. At the beginning of the revised “conclusion”, we specifically claim thinner 2M WS₂ has higher thermal

stability. In the end of the revised “conclusion”, we specifically mention the 2M TMDs. The revised sentences are copied below:

In summary, we have demonstrated ultrathin 2M WS₂ has significantly higher thermal stabilities than the bulk counterparts.

The high intrinsic phase stabilities of ultrathin 2M TMDs can inspire their tempting applications in various fields, including superconductor, electronics and energy conversion and storage.

Changes to the manuscript: This content is mentioned in the manuscript on Page 22-23.

(iii) *The authors use crystal settings in which the 2H-phase stacks along the c-axis but the 1T' along the b-axis. I know these are standard crystallographic settings, but it might be easier to choose an orthorhombic setting for the 1T'-phase in which the layers also stack along c, and make comparison between the two and following hkl incites easier.*

Response: Thanks for this good suggestion. To clear out doubts, specifically, we use the crystal setting in which 2H WS₂ stacks along the *c* axis and 2M WS₂ stacks along the *a* axis, rather than the *b* axis. These are customary settings for these kinds of TMDs in literatures (*Nat. Mater.*, 2021, 20, 1113–1120; *Adv. Mater.* **2019**, 31, 1901942; *Nat. Chem.*, 2018, 10, 638–643; *J. Am. Chem. Soc.*, 2019, 141, 790-793; *Angew. Chem. Int. Ed.*, 2018,57,1232–1235). To make the manuscript readable for most of the researchers in this field, we prefer to keep those settings when we discuss the lattice parameters, TEM images and XRD patterns. Whereas, following the reviewer’s suggestion, we have used the orthorhombic setting for both 2H and 2M WS₂ cells in our theoretical calculation, as shown in Fig. 5.

(iv) *The authors might want to cite the following work where the phase transition in WS₂ was reported before:* [https://pubs.aip.org/aip/apl/article-abstract/115/3/032102/37971/Evidence-for-a-narrow-band-gap-phase-in-1T-WS₂?redirectedFrom=fulltext](https://pubs.aip.org/aip/apl/article-abstract/115/3/032102/37971/Evidence-for-a-narrow-band-gap-phase-in-1T-WS2?redirectedFrom=fulltext).

Response: This work has been cited in the manuscript with a reference number of 37 on Page 6.

(v) *The authors might want to discuss the following report in which high phase stability of 1T'-ML is also reported:* <https://www.science.org/doi/full/10.1126/sciadv.add6167>

Response: This work reports chemical exfoliation of WS₂ and claims the prepared monolayered 1T' WS₂ is stable in the air at room temperature, which agrees to the conclusion in our manuscript that ultrathin 2M WS₂ (or 1T') has intrinsic high phase stability. This work is discussed in the manuscript as:

In agreement with this result, chemically exfoliated ML 1T' WS₂ is also demonstrated to have good air-stability at room temperature^{37,38}.

Changes to the manuscript: This work has been cited in the manuscript with a reference number of 38 on Page 6.

Comment 8: To sum up, I think the general result might be valid but there are many unresolved issues with this manuscript. In addition, because parts were confusing, this review is not complete. I would look at the paper again, but it will likely need a few rounds of revision before being suitable for publication.

Response: Thanks for voting our work generally valid. We have carefully addressed all the raised issues and revised our manuscript accordingly. We are also happy to make further revision if there is any unresolved issue.

Referee: 3

Comments: *The authors report on the layer-number stability of metastable WS₂. They found that the atomically thin 1T' WS₂ possesses higher phase and anti-oxidant stability compared to its bulk counterpart, confirmed by Raman and optical microscopy with annealing. They also demonstrated the existence of intermediate states between the 1T' and 2H phases, as revealed by Raman spectroscopy, XRD, TEM, and DFT calculations. The authors proposed that the enhanced thermal conductivity and the stiffening of intralayer bonds at the monolayer limit are the causes of this phenomenon. Although the long-term stability of monolayer 1T' MoS₂ and WS₂ has been reported previously (e.g., L. Liu et al., Nat. Mater., 17, 1108 (2018), and M. Okada et al., ACS Nano, 16, 13069 (2022)), the authors' findings concerning the enhanced phase stability of monolayer 1T' TMDs over the bulk form are intriguing. In general, the opposite is observed (F. Ye et al., Small, 42, 5802 (2016)). As such, I believe that these results will attract significant interest and propel the scientific study of metastable TMDs forward. However, I recommend a major revision prior to acceptance, as there are several points that need improvement. In particular, analysis of the intermediate state and the discussion of the mechanism are insufficient.*

Response: Thanks for the positive feedback and rating the significance and broad interest of our manuscript high. In the revised manuscript, we have carefully addressed the issues regarding analysis of the intermediate states and phase change mechanisms.

Comment 1: *In Figure S2-5, the authors should display all the Raman spectra that were used to construct Fig. 1.*

Response: Thanks for this good suggestion. We have presented all the regarded Raman spectra in Supplementary Fig. 4-6 and 8, which are attached in the Response to Comment 4.

Changes to the manuscript: Supplementary Fig. 4-6 and 8 are updated in the revised supporting information.

Comment 2: The authors should provide the definition of the transition temperature in Fig. 1e. For instance, this could be the temperature at which the intermediate state emerges or the temperature at which the phase completely transitions into the 2H phase.

Response: Thanks for this good suggestion. In the revised manuscript, we define the phase transition temperature as the one at which the 2M or 1T' WS₂ starts to lose phase purity, involving circumstances of 2M to 2H (or 1T' to 1H) phase transition and monolayered 1T' WS₂ being oxidized by the air. The transition temperature of monolayered WS₂ in the air depicted in the revised Fig. 1e specially means the temperature at which it starts to be oxidized and decomposed. This has been specified in the revised caption of the revised Fig. 1e that is attached below.

Revised Fig. 1e. 2M to 2H (or 1T' to 1H) phase transition temperatures as a function of WS₂ layer thickness measured in the air or Ar atmosphere. In the heating program, temperature was elevated by 5 °C and held for 1 min or 15 min in each step. The transition temperature of ML WS₂ in the air specially means the temperature at which it begins to be oxidized.

In the revised manuscript, the phase transition temperature is defined as:

Here we define phase transition temperature as the one at which the 2M or 1T' WS₂ starts to lose phase purity, i.e., the E_{2g}¹ Raman mode emerges or the WS₂ sample begins to be oxidized by the air.

Changes to the manuscript: This content is mentioned in the manuscript on Page 5.

Comment 3: At line 100, the authors should specify the number of samples they measured. The phrase "many samples" lacks precision.

Response: We agree it is not appropriate to use the phrase “many samples”. In this sentence, we attempt to emphasize we have prepared WS₂ samples with thickness covering a wide range, while the exact number of samples is not so important to our discussion about the trend of phase stability. Therefore, we choose to remove the phrase “many samples” from the sentence:

To further confirm the thickness-dependent stability of 2M WS₂, we have prepared exfoliated samples with thickness covering a wide range (from 23 L to ML) and systematically investigated their tolerance to heat in the air or inert atmosphere.

Changes to the manuscript: This content is mentioned in the manuscript on Page 8.

Comment 4: The authors contend that an "intermediate state" exists. However, the structure is neither in the 2H phase nor the 1T' phase (Fig. 2c). If this is the case, it seems peculiar that the Raman spectra of the intermediate state are merely the sum of the spectra of the two crystalline phases. Are there any other alterations in the Raman spectra attributable to the formation of the intermediate state? Authors should discuss Raman scattering of the intermediate state deeper.

Response: Thanks for this good comment. In the revised manuscript, we precisely describe the intermediate phases WS₂ as ones which experience various extents of 2M to 2H phase transition. To thoroughly investigate the Raman scattering of intermediate phase WS₂, we have prepared a series of intermediate phases WS₂ by heating 2M WS₂ flakes above phase transition temperatures within controlled times. Raman spectra (Supplementary Fig. 4-6 and Supplementary Fig. 10 attached below) of these intermediate phase WS₂ samples exhibit sums of 2M and 2H characteristic Raman modes without additional peaks compared to the Raman spectra of pure 2M and 2H WS₂. This is because intermediate phases WS₂ have 1T'/1H heterostructures. The HRTEM image (revised Fig. 2d-f) of an intermediate phase WS₂ shows in-plane 1T'/1H hetero-structure with slight lattice distortion in the 1T' areas. XRD (revised Fig. 2g-h) reveals an intermediate phase WS₂ has hybrid low-order stacks of 1T' and 1H layers. Because of the hybrid 1T'/1H stack, Raman spectra (Supplementary Fig. 10, attached below) of intermediate phases WS₂ manifests that A_{1g} mode (ascribed to out-of-plane

vibration of 1H layer stacks) locates at lower wave number compared to 2H phase and keeps to blue-shift as the extent of 2M to 2H phase change increases.

Supplementary Fig. 10. Optical and AFM images and Raman spectra of a piece of multilayered WS₂ flake in 2M, intermediate and 2H phases. 2H and intermediate phases WS₂ were obtained by heating the 2M WS₂ in the air at 250 °C for 20 min and at 120 °C for various times, respectively. A_{1g} mode exhibits blue shifts as the extent of 2M to 2H phase change increases.

In the revised manuscript, the intermediate phases WS₂ are explicitly described and the associated HRTEM, XRD and Raman characterizations are discussed as:

The slowness of 2M to 2H phase transition of WS₂ allows us to trace this process by investigating structures of intermediate phases after varied extents of 2M to 2H phase change. Intermediate phases WS₂ can be obtained by heating the 2M WS₂ above phase transition temperature within controlled time. As shown in Supplementary Fig. 4-6, all intermediate phases WS₂ flakes show superpositions of 2M and 2H characteristic Raman modes, indicating the intermediate phases of WS₂ are probably 1T'/1H heterostructures that can involve in-plane heterostructured areas and hybrid stacked layers. Fig. 2d-f show high-resolution transmission electron microscopy (HRTEM) images of W-atom configurations in 2M and an intermediate phase. Zigzag chains of W atoms are observed in Fig. 2d, in consistency with the reported structure of 2M WS₂.³ Area ① of the intermediate phase WS₂ (Fig. 2e) reveals W-atom lines distances ranging from 0.26 to 0.31 nm, which are in between d_1 (0.23 nm)

and d_2 (0.34 nm) of 2M WS₂ (Fig. 2d). Fourier transform of Area ① displays irregular quadrilaterals (inset of Fig. 2e) that are distorted from the rectangular pattern of 2M WS₂ (inset of Fig. 2d). Area ② of the intermediate phase WS₂ has a W-atom lines distance of 0.27 nm (Fig. 2f) and hexagonal pattern Fourier transform (inset of Fig. 2f), which are the same as 2H WS₂.⁴² These results demonstrate that an intermediate phase WS₂ has an in-plane heterostructure of distorted 1T' and 1H, which is formed due to gliding of W atoms. To investigate layer stacking configurations of intermediate phases WS₂, XRD has been carried out on bulk 2M WS₂ powder during heating at 130 °C. With prolongation of heating time, XRD peaks attributed to 2M (200) diffractions are weakening and broadening, while the ones attributed to 2H (002) diffractions are intensifying and sharpening, and all these peaks are shifting to lower degrees but locating in between that of standard 2M (200) and 2H (002) diffractions, as shown in Fig. 2g-h. These results indicate ordered stacks of 1T' and 1H layers are broken and built, respectively, with increase of phase change extent. An intermediate phase WS₂ has hybrid low-order stacks of 1T' and 1H layers with average interlayer spacing in between that of pure 2M and 2H phase. In agreement with this result, Raman spectra (Supplementary Fig. 10) of intermediate phases WS₂ manifests that A_{1g} mode (ascribed to out-of-plane vibration of 1H layer stacks) locates at lower wave number compared to 2H phase and keeps to blue-shift as the extent of 2M to 2H phase change increases.

Changes to the manuscript: This content is mentioned in the manuscript on Page 11-12.

Comment 5: In Fig. S6, is the crystallographic structure data sourced from previous reports or was it calculated by the authors? The authors need to clarify this point. If the structure data was referenced from a previous paper, the authors should cite the article.

Response: The crystal structure in the previous version Fig. S6 is referenced from a previous report (*J. Solid State Chem.*, 1987, 70, 207-209). This figure has been modified and entitled with Supplementary Fig. 9 (attached below), and the mentioned report is cited in the caption:

Revised Supplementary Fig. 9. (a) Distorted octahedral coordinated W and S atoms and representative W–S bond lengths and W–S–W angles and the average values and (c) the interlayer spacing of 2M WS₂. S1 atoms locate in C or A' planes and S2 atoms locate in C' or A planes as defined in Fig. 2a. (b) Trigonal prismatic coordinated W and S atoms and W–S bond length and W–S–W angle and (d) the interlayer spacing of 2H WS₂ (*J. Solid State Chem.*, 1987, 70, 207-209). Color code: blue and orange spheres represent W and S, respectively.

Changes to the manuscript: *J. Solid State Chem.*, 1987, 70, 207-209 is cited in the caption of Supplementary Fig. 9 with a reference number of 9 and in the manuscript with a reference number of 42 on Page 2.

Comment 6: Regarding the XRD (Fig. 2g), what is the origin of the peak around 32 degrees in the intermediate sample? The authors should assign the peak.

Response: Thanks for this good suggestion. The peak at $\sim 32^\circ$ is attributed to $(11\bar{1})$ diffraction of 2M WS₂ (actually at 31.9°), except with a little shift to higher degree. The peak of $(11\bar{1})$ diffraction is hard to observe in the XRD spectrum of pure 2M WS₂, because the (200) diffraction peaks are too strong, since most of WS₂ flakes are lying along bc plane. These main peaks are significantly weakened in the intermediate phase WS₂ because of hybrid stacks of 1T' and 1H layers. Therefore, $(11\bar{1})$ peak turns manifest in some of the intermediate phases WS₂. Similar interoperation works for other small XRD peaks, including (002) $(31\bar{1})$ (601) (311) diffractions. In the revised manuscript, we

present XRD spectra of a series of intermediate phases WS₂ in revised Fig. 2g-h (attached below) and the zoomed-in view of the small peaks in supplementary Fig. 11 (attached below), and discuss these peaks as:

High-index XRD diffractions including 2M (11 $\bar{1}$) (002) (31 $\bar{1}$) (601) (311) are also disappearing with increasing of heating time, with 11 $\bar{1}$ diffraction shifting to higher degrees that can be attributed to in-plane atomic gliding and out-of-plane displacements, as shown in Supplementary Fig. 11.

Revised Fig. 2g-h. (g) Powder XRD patterns of 2M, intermediate phases and 2H WS₂. 2H and intermediate phases WS₂ were obtained by heating 2M WS₂ in the air at 250 °C for 20 min and at 130 °C for various times, respectively. (h) A zoomed-in view of the XRD patterns in the range of 13.5°~16.0°.

Supplementary Fig. 11. A zoomed-in view of Fig. 2g in the range of 31° ~ 43°.

Changes to the manuscript: This content is mentioned in the manuscript on Page 12-13.

Comment 7: The authors assert a layer-by-layer phase transition, but could this merely represent a reduced energy barrier due to the increased number of WS₂ formula units in the unit cell, stemming from the increased number of layers? The actual value is approximately 1/2 or 1/3 of the monolayer. Thus, from a certain layer, the phase change barrier remains constant. The proposed mechanism should therefore be reconsidered.

Response: Thanks for this good comment. We agree that the calculated phase transition energy barriers for bilayered and trilayered WS₂ are respectively about 1/2 and 1/3 of that of monolayered WS₂, as shown in Fig. 4. It should be noted that these energy barriers values have been normalized by the number of WS₂ unit cells. That is to say, the total energy barriers for phase transitions of bilayered or trilayered WS₂ would be almost the same as that of monolayered WS₂, if all different layers in bilayered or trilayered WS₂ experience 1T' to 1H concertedly, rather than layer by layer. However, all the XRD, AFM, Raman and theoretical calculation results indicate phase transition of multilayered WS₂ occurs layer(s) by layer(s). As a result, compared to the 1T' to 1H phase transition of a monolayered WS₂, a multilayered WS₂ is easier to have one layer transforming to 1H phase with other layers maintaining the 1T' structure. In another word, a multilayered WS₂ has smaller energy barrier than the monolayered WS₂ to start the 1T' to 1H phase transition within one layer, rather than entirely transform from the 2M to 2H phase with all layers.

Comment 8: Figures should be numbered in the sequence in which they appear.

Response: In the revised manuscript, all figures have been numbered in the sequence in which they appear, except for Fig. 1d. The Raman spectra (Fig. 1d) have to be firstly discussed to introduce the E_{2g}¹ mode as an indicator for emergence of phase change, followed by discussion about optical images, AFM images and Raman mappings of exfoliated samples at different temperatures. However, to make the entire Fig. 1 tidily arranged, it is better to align the images and plots respectively, i.e., the Raman spectra (Fig. 1d) are placed together with the statistical plots (Fig. 1e-f) after the images

(Fig. 1a-c) of exfoliated samples. Therefore, it is more appropriate to keep the layout of Fig. 1 and discuss Fig. 1d before Fig. 1a-c.

Comment 9: Excessive significant figures in the AFM heights (Fig. 2k), chemical bond angles/lengths in lines 138-145 and 267-282, etc., need to be reviewed and corrected.

Response: Thanks for this good suggestion. We have carefully checked all these figures and made corrections accordingly. Both Fig. 5a and Fig. 2a present W-atom line distances of 2M WS₂, and both Fig. 5a and Supplementary Fig. 9a present W–S bonds lengths of 2M WS₂. Structural data in Fig. 2a and Supplementary Fig. 9a are referenced to the articles (*Adv. Mater.* 2019, 31, 1901942; *Nat. Mater.*, 2021, 20, 1113–1120) that are obtained based on measured XRD results, while structural data in Fig. 5a are output from our theoretical calculations. The corresponding data in those figures have a little difference. To make them consistent with each other in the revised manuscript, we changed the data in Fig. 5a and the related texts to be the same with that in Fig. 2a and Supplementary Fig. 9a.

Changes to the manuscript: (1) Thickness of per WS₂ layer in the previous version Fig. 2k is removed from the manuscript; (2) d_1 and d_2 of the coord. I in the revised Fig. 5a and in the text mentioning them are changed to 2.280 Å and 3.340 Å, respectively. (3) Values of chemical lengths of coord. I in the revised Fig. 5a and in the text mentioning them are changed to be the same as the corresponding values in Supplementary Fig. 9a.

Comment 10: The methods section lacks sufficient detail on the DFT calculation: information such as k-mesh, cut-off energy, etc., is missing. Please provide enough information to allow other researchers to reproduce the results.

Response: Thanks for this good suggestion. We have added the information in the revised supplementary information:

The cutoff energy for the plane-wave basis was set to be 520 eV. The Γ -centered $10 \times 6 \times 1$ k-point mesh was used to sample the Brillouin zone (BZ) by employing the Monkhorst-Pack method.

The convergence criteria for energy and atom forces were 10^{-5} eV and 0.02 eV/Å, respectively.

Changes to the manuscript: This content is mentioned in the supporting information on Page 5.

Comment 11: Definitions of abbreviations must be given at their first usage. Several terms, such as "T-type", are used without definition. There are also instances where abbreviations are defined but not used subsequently (e.g., 3 L and TL). Please review the manuscript and rectify these issues.

Response: In the revised manuscript, the terms of "T-type" and "H-type" have been removed, and we define "TL" as the abbreviation of "trilayer" on Page 5, and only use "TL" in the following text, while we keep the annotations of "1 L", "2 L" and "3 L" in Fig. 1 and Fig. 3, because they are more explicit than "monolayer (ML)", "bilayer (BL)" and "trilayer (TL)", respectively.

Comment 12: There are numerous typographical errors in the manuscript that need to be corrected.

Response: All typos have been corrected.

REVIEWER COMMENTS

Reviewer #1 (Remarks to the Author):

The authors have addressed my concerns well and have made substantial improvement in the revised manuscript. The manuscript can be accepted.

Reviewer #3 (Remarks to the Author):

The manuscript has seen significant improvement. Nonetheless, certain areas require further refinement to align with the journal's standards.

1. Figures 1e and f: The authors need to revise the figures to clearly differentiate between samples that have undergone a phase transition to the 2H phase and those that have been oxidized.
2. In connection with the above: If oxidation and phase transition occur concurrently, how did the authors differentiate between them? If both phenomena transpire simultaneously, this should be addressed in the discussion.
3. The authors should incorporate a scale bar and line profile in the optical and AFM images presented in Fig. S4. It would be beneficial to highlight the region where the AFM measurement was conducted within the corresponding optical image.
4. Figures 4 and 5 and the associated text: The response provided by the authors does not meet my expectations. While I concur that a layer-by-layer phase transition might be experimentally accurate, and it's plausible that thinner 2M phase WS_2 is more stable than its bulk counterpart, I have reservations about the basis of the authors' argument. The authors have relied on DFT results indicating that the barrier decreases in a layer-by-layer fashion. As highlighted in my initial comment and acknowledged in the authors' rebuttal, isn't the layer-dependent phase transition barrier reduction in the DFT calculation attributable solely to the increasing the number of the unit cell in the superlattice? Extending this logic suggests that with 40 layers, the phase transition barrier would be lower than the thermal energy at ambient conditions, and in the bulk scenario, the phase change barrier would be nullified, implying the non-existence of a stable bulk 2M/1T' phase. Given that bulk 2M phase WS_2 crystals are accessible to researchers, it's evident that there might be inaccuracies in the DFT calculations or their interpretation. Consequently, relying on the presented DFT results for the argument is flawed. The authors are urged to revisit and reinterpret the DFT findings, considering they might be outcomes of optimization.
5. Pertaining to my 11th comment: Utilizing the abbreviation "3L" consistently throughout the manuscript would enhance its readability.

6. In reference to my 12th comment: The manuscript should avoid using two-byte characters. Additionally, the term "1H WS₂" is not accurate in crystallography; a monolayer semiconducting WS₂ is not hexagonal.

Reviewer #4 (Remarks to the Author):

The authors reported layer-dependent phase transformation of 2M-WS₂ to 2H-WS₂ with mainly Raman spectroscopy studies combined with PL, XRD, HRSTEM, AFM and DFT simulations. The most significant finding of this study is that the monolayer 1T'-WS₂ exemplify higher phase stability compared with its thicker counterparts. The authors attributed the cause of such monolayer phase stability to the stiffened intralayer bonds and the enhanced thermal conductivity. I found the results are interesting. However, there are multiple places that need improvements before I could recommend this manuscript for acceptance. A general concern is that the laser power reported in this work does not reflect the laser intensity that actually hits each sample. Since this work is mainly based on Raman measurement and laser power is a very important parameter to study the phase stability, the laser power should be reported accurately. The laser power that hits the sample can be significantly different from the output number of the instrument depending on the optical path of the specific instrument and the status of the laser on each day of the experiments. The laser intensity on the sample of the same laser source would be different if the objective is different. A more accurate way to report laser intensity is to use a power meter to measure the laser power right in front of the sample on each day of the experiments and convert it to laser intensity. The laser power values from the instrument settings reported in this manuscript makes the laser-power coefficients less meaningful and less useful as a reference for the future follow-up studies. Therefore, it is important to use laser intensity rather than laser power from the instrument settings in this study. Other comments are listed as followings:

- 1) Although ML in Fig1a6 seem to be faint, the optical contrast of each image in Figure 1a are different. I would recommend normalizing the optical contrast change of each layer number region as compared to the background region to provide a more quantified result.
- 2) On Page 6 line 108, do you mean "refraction index" at the place of "infracion index"?
- 3) I found there are missing scale bars on multiple optical microscopy images, AFM images, and PL maps, such as Figure 1c4-5, Figure 2i-m, Supplementary Figure 4, 5, 6, 8, 10, and 13.
- 4) Are Figure 1c3 and the optical microscopy and AFM images from Supplementary Figure 4, 5, 6, 8, 10, and 13 taken before or after heating? The experiment conditions should be clearly addressed. The supplementary AFM images (Supp Figure 4 1L, 3L, 6L, Supp Figure 5 2L, 3L, 4L, 20L, and Supp Figure 6 3L and 5L) show multiple samples' surfaces are not smooth. The white dots on the sample surfaces suggest either the samples are oxidized or contaminated or of low quality. Therefore, it is important to specify if the images were taken before or after the heating and the authors should explain why some samples surfaces are not smooth.
- 5) On page 8, the authors claim that "High-resolution XPS (Supplementary Fig. 7a-b) shows that W 4f peak and valance band edge of 2M WS₂ locate about 1.1 eV and 0.85 eV lower than that of 2H WS₂,

respectively, indicating W element in 2M WS₂ has lower oxidation state than that in 2H WS₂. This is believed to be caused by existing extra electrons at 2M WS₂ surface". It is incorrect to use XPS to identify the change of oxidation state if W atoms from each sample are not in the same chemical environment, in this case, distorted octahedral and trigonal prismatic coordination. Although XPS is a surface technique, a proper measurement could reflect bulk properties unless the samples are not homogeneous. The authors should make more comments to explain why extra electrons exist at the 2M WS₂ surface and how are the sample charge balanced with the extra electrons. What about the inside underneath the surface?

On page 9, "Nevertheless, the O 1s peak (Supplementary Fig. 7c) barely changes during the 2M to 2H phase transition. These results indicate oxygen molecules speed up phase transition by extracting electrons from 2M WS₂ during heating in the air, without chemically oxidizing the WS₂." I found this is confusing, does authors mean the O1s peaks come from oxygen molecules? Why are they not shifting to lower binding energies if they took electrons from 2M WS₂? I also noticed that there is a significant amount of O 1s peaks on Supplementary Fig2. Since XPS experiments are carried out under high vacuum, it is unlikely all the O signals are from the adsorbed O₂ molecules from air. It seems more likely that the surface of the sample is slightly oxidized. In Supplementary Fig.7, the shifting of W peaks is not monotonically as increasing the heating time. The 130 C 45 min heated sample seems to be closer to 2M WS₂ while the 15 min heated sample contains more 2H phase. This result is inconsistent. I suggest the authors to remove the top surface layers with scotch tape right before loading the sample into the XPS chamber to expose a fresh layer and redo the XPS experiments to avoid potential surface contamination. In addition, I would suggest to avoid using the Ar plasma gun to clean the surface since the energy from it could change the surface structure and lead to inaccurate results in this case. Also, the authors should specify the above XPS experiment conditions and how the peaks are calibrated in the methods section.

6) Figure 2d-f shows the HRSTEM images of the 1T' and the lateral heterostructure intermediate phases. Since the areas shown in the Figure 2d-f are very small, I am wondering how many samples and how much areas were studied to reach such a conclusion. Did the authors observe structure changes from 1T' to 1H during the EM studies since electron beams from TEM carries enough energy to introduce phase transformation as well. The lateral heterostructure shown in Figure 2 could be beam damage.

7) Figure 2i and the text about it are confusing. Why are the right part of the flakes disappeared after laser punching. What laser power is this? Why is it so destructive? Is it the same power as other measurements? This makes the other Raman results rather concerning.

8) Supplementary Fig. 11 seems to have a very low signal to noise ratio. The (11-1) peaks seem to shift to higher angle before disappearing. Please make more comments on this.

9) From the optical microscopy images of Supplementary Fig. 12, the flakes seem to be less homogenous as they transform from 2M to 2H. Can the authors explain this?

10) Supplementary Fig. 14 a, the Ag5 peak seems to be mislabeled.

11) On page 20, line 481, "Thes" seems to be a typo.

12) There are multiple experimental details missing in the methods section. Especially, on the Raman measurement, What's the grating numbers and exposure times, and how many integrations were used? Are they the same throughout the experiments of this studies or the settings are changed with different layer numbers to achieve the best signal to noise ratio. In that case, how does that affect the results of thickness dependent Raman modes studies.

Point-by-Point Responses to Reviewers' Comments

Referee: 1

Comments: *The authors have addressed my concerns well and have made substantial improvement in the revised manuscript. The manuscript can be accepted.*

Response: Thanks for the positive feedback.

Referee: 3

Comments: *The manuscript has seen significant improvement. Nonetheless, certain areas require further refinement to align with the journal's standards.*

Response: Thanks for the positive feedback and we have further revised the manuscript accordingly.

Comment 1: *Figures 1e and f: The authors need to revise the figures to clearly differentiate between samples that have undergone a phase transition to the 2H phase and those that have been oxidized.*

Response: Thanks for this good suggestion. In the revised Fig. 1e (attached below), we use squares to label the data points of 2M to 2H phase transition and use triangles to label the data points of 1T' WS₂ being oxidized by the air. The revised Fig. 1f (attached below) is labeled as "2M to 2H transition in the air".

Changes to the manuscript: Fig. 1e and 1f are revised as:

Revised Fig. 1. (e) 2M to 2H (or 1T' to 1H) phase transition temperatures as a function of WS₂ layer thickness measured in the air or Ar atmosphere. In the heating program, temperature was elevated by 5 °C and held for 1 min or 15 min in each step. The transition temperature of ML WS₂ in the air

specially means the temperature at which it begins to be oxidized. (f) The power of laser required to activate 2M to 2H phase transition as a function of WS₂ layer thickness measured in the air.

Comment 2: In connection with the above: If oxidation and phase transition occur concurrently, how did the authors differentiate between them? If both phenomena transpire simultaneously, this should be addressed in the discussion.

Response: Thanks for this good comment. We have observed that monolayered 1T' WS₂ is oxidized and decomposed in the air after heated to 350 °C without transforming to 1H phase, while bilayered and multilayered 2M WS₂ samples transform to 2H phase in the air after heated to specific temperatures without being oxidized. Thus, air oxidation and phase transition don't occur concurrently. To make it clear to readers, we summarize the air oxidation and phase transition as:

In our experiments of heating in the air, we never observe oxidation of BL or multilayered 2M WS₂ flakes before 2M to 2H phase transition, while we never observe 1T' to 1H phase transition on a ML 1T' WS₂ flake before it was oxidized and decomposed. This means 1T' to 1H phase transition temperature of ML 1T' WS₂ is higher than the air oxidation and decomposition temperature.

Changes to the manuscript: This content is mentioned in the manuscript on Page 6.

Comment 3: The authors should incorporate a scale bar and line profile in the optical and AFM images presented in Fig. S4. It would be beneficial to highlight the region where the AFM measurement was conducted within the corresponding optical image.

Response: Following this suggestion, we have added scale bars in the optical and AFM images of the revised Supplementary Fig. 4-6, 8 and 10, and the AFM measured regions are labeled by dotted squares in the corresponding optical images. Please find the revised figures in the revised supporting information.

Changes to the manuscript: Supplementary Fig. 4-6, 8 and 10 are revised accordingly.

Comment 4: Figures 4 and 5 and the associated text: The response provided by the authors does not meet my expectations. While I concur that a layer-by-layer phase transition might be experimentally accurate, and it's plausible that thinner 2M phase WS₂ is more stable than its bulk counterpart, I have reservations about the basis of the authors' argument. The authors have relied on DFT results

indicating that the barrier decreases in a layer-by-layer fashion. As highlighted in my initial comment and acknowledged in the authors' rebuttal, isn't the layer-dependent phase transition barrier reduction in the DFT calculation attributable solely to the increasing the number of the unit cell in the superlattice? Extending this logic suggests that with 40 layers, the phase transition barrier would be lower than the thermal energy at ambient conditions, and in the bulk scenario, the phase change barrier would be nullified, implying the non-existence of a stable bulk 2M/1T' phase. Given that bulk 2M phase WS₂ crystals are accessible to researchers, it's evident that there might be inaccuracies in the DFT calculations or their interpretation. Consequently, relying on the presented DFT results for the argument is flawed. The authors are urged to revisit and reinterpret the DFT findings, considering they might be outcomes of optimization.

Response: We appreciate the reviewer's deep insight which has led us to be clearer about this question. The energies presented in Fig. 4 (previous manuscript) are average energies per asymmetric unit that contains a W atom and two S atoms. In another word, we normalized the calculated energies by the number of asymmetric units. We agree the as-exhibited layer-dependent trend is attributed to the increase of number of WS₂ asymmetric units. As shown in the revised Fig. 5a (attached below), the total phase transition barriers of 1 L, 2 L and 3 L WS₂ supercells (structures shown in the Revised Fig. S18, attached below) are 1.83 eV, 1.87 eV and 1.86 eV, respectively, which are quite close. This is associated with the layer-by-layer phase transition mechanism. Since the 2 L or 3 L WS₂ supercell in transition state contains only one highly distorted layer that has similar structure to the transition state 1 L WS₂, with the other layer(s) basically maintaining 1T' phase, the energy differences between transition states and initial 2M or 1T' phases are close for 1 L, 2 L and 3 L WS₂ supercells. However, this does not mean 1 L, 2 L and 3 L WS₂ should have similar phase transition temperature. 1 L, 2 L and 3 L WS₂ supercells defined in our DFT calculation contain two, four and six W atoms and four, eight and twelve S atoms, respectively, as shown in the Revised Fig. S18 (attached below). Since each cell of thicker WS₂ involves more W and S atoms, thicker WS₂ has higher specific heat ($\text{J K}^{-1} \text{mol}^{-1}$). Accordingly, to climb over the same energy barrier, thicker WS₂ requires fewer degrees of temperature rise. That is why thicker WS₂ has lower phase transition temperature.

Revised Fig. 5. Energy profiles of 2M to 2H phase transition. (a) Total energies and (b) average energies of 1 L, 2 L and 3 L WS₂ supercells in the initial 2M (or 1T') phase (coord. I), transition states (coord. III for 1 L and 3 L WS₂ and coord. IV for 2 L WS₂), final 2H (or 1H) phase (coord. VII) and other intermediate configurations. Energies in b are normalized by the numbers of WS₂ layers. Ea^{1L} , Ea^{2L} and Ea^{3L} are energy barriers of 2M to 2H (1T' to 1H) phase transitions for 1 L, 2 L and 3 L WS₂ supercells, respectively. $\bar{E}a^{1L}$, $\bar{E}a^{2L}$ and $\bar{E}a^{3L}$ are average energy barriers per layer of 1T' to 1H phase transitions for 1 L, 2 L and 3 L WS₂, respectively.

Revised Supplementary Fig. 18. Structures of 1 L 1T' (left), and 2 L (middle) and 3 L (right) 2M WS₂ supercells applied in theoretical calculations. Color code: blue and orange spheres represent W and S, respectively.

To make the DFT-calculated energy barriers of 1 L, 2 L and 3 L WS₂ comparable, it is better to normalize the calculated phase transition barriers by the number of layers. The as-obtained normalized energy profile is presented in the revised Fig. 5b (attached above), which shows the same trend as the Fig. 4 in the previous manuscript. This result demonstrates that thicker WS₂ has smaller average phase transition barrier per layer, which supports the fact that thicker 2M WS₂ can initiate the layer-by-layer phase transition at a lower temperature.

We understand the Reviewer's concern that the average phase transition barrier per layer for bulk WS₂ would be even smaller than thermal energy in ambient condition, if the total barrier of a bulk WS₂ supercell is also close to that of monolayered WS₂. However, it should be noted that the total transition barriers of 2 L and 3 L WS₂ are slightly higher than that of 1 L WS₂, as shown in the Revised Fig. 5a (attached above). Moreover, bulk WS₂ lattice in the transition state probably contains multiple highly distorted layers, rather than only one as 2 L and 3 L WS₂. Therefore, it is more reasonable to conjecture bulk WS₂ has significantly larger total transition barrier than monolayered WS₂. That is to say, with increase of thickness, total transition barrier is increasing and the average barrier per layer decreases slower and slower. This also agrees with the experimental result that phase transition temperatures of 2M WS₂ thicker than 6-layer are almost the same.

To address the reviewer's concerns and make the manuscript more readable, we present the structures of WS₂ supercells used in DFT calculations in the revised Fig. S18, and we discuss the phase transition energy barrier after the layer-by-layer transition mechanism. We present both the total transition energy barriers and average barriers per layer in the revised Fig. 5 and discuss it as:

The energy difference between lattices of transition state and the initial 2M phase is the energy barrier for 2M to 2H phase transition. As shown in Fig. 5a, the total phase transition barriers of 1 L, 2 L and 3 L WS₂ supercells are 1.83 eV, 1.87 eV and 1.86 eV, respectively, which are quite close. This is because the 2 L or 3 L WS₂ supercell in transition state contains only one highly distorted layer that has similar structure to the transition state 1 L WS₂, with the other layer(s) basically maintaining 1T' phase. After we normalize the total phase transition barriers by the number of layers, as shown in Fig. 5b, the average barrier per layer of 1 L WS₂ (1.83 eV) is apparently higher than 0.93 eV of 2 L WS₂ and 0.62 eV of 3 L WS₂, theoretically confirming thinner 2M WS₂ has higher phase stability. It is the heterostructure configurations that result in lower average energies per layer for multilayered WS₂ than

ML WS₂ when they are in transition states. This means thicker 2M WS₂ has smaller phase transition barrier per layer, agreeing with the fact that thicker WS₂ can initiate the layer(s)-by-layer(s) 2M to 2H phase transition at a lower temperature.

Changes to the manuscript: This content is mentioned in the manuscript on Page 21.

Comment 5: Pertaining to my 11th comment: Utilizing the abbreviation “3 L” consistently throughout the manuscript would enhance its readability.

Response: Following this suggestion, we use the abbreviation “2 L” and “3 L” for “two-layered” and “three-layered”, respectively, throughout the revised manuscript.

Changes to the manuscript: All the words of “BL” and “TL” are changed to “2 L” and “3 L” in the manuscript, respectively.

Comment 6: In reference to my 12th comment: The manuscript should avoid using two-byte characters. Additionally, the term “1H WS₂” is not accurate in crystallography; a monolayer semiconducting WS₂ is not hexagonal.

Response: ① and ② are changed to [1] and [2], respectively. A monolayered semiconducting WS₂ has a hexagonal rotary-inversion axis, so it is hexagonal. The term “1H” is commonly used to describe the structures of monolayered semiconducting WS₂ and MoS₂ in the literature (*Chem. Soc. Rev.*, 2020, 49, 3952–3980; *Chem. Eur. J.* 2018, 24, 15942–15954; *Chem. Soc. Rev.*, 2015, 44, 2702–2712; *Nat. phys.*, 2017, 13, 931–937; *Science*, 2014, 346, 1344–1347).

Changes to the manuscript: This content is mentioned in the manuscript on Page 12.

Referee: 4

Comments: *The authors reported layer-dependent phase transformation of 2M-WS₂ to 2H-WS₂ with mainly Raman spectroscopy studies combined with PL, XRD, HRSTEM, AFM and DFT simulations. The most significant finding of this study is that the monolayer 1T'-WS₂ exemplifies higher phase stability compared with its thicker counterparts. The authors attributed the cause of such monolayer phase stability to the stiffened intralayer bonds and the enhanced thermal conductivity. I found the results are interesting. However, there are multiple places that need improvements before I could recommend this manuscript for acceptance. A general concern is that the laser power reported in this work does not reflect the laser intensity that actually hits each sample. Since this work is mainly based on Raman measurement and laser power is a very important parameter to study the phase stability, the laser power should be reported accurately. The laser power that hits the sample can be significantly different from the output number of the instrument depending on the optical path of the specific instrument and the status of the laser on each day of the experiments. The laser intensity on the sample of the same laser source would be different if the objective is different. A more accurate way to report laser intensity is to use a power meter to measure the laser power right in front of the sample on each day of the experiments and convert it to laser intensity. The laser power values from the instrument settings reported in this manuscript makes the laser-power coefficients less meaningful and less useful as a reference for the future follow-up studies. Therefore, it is important to use laser intensity rather than laser power from the instrument settings in this study. Other comments are listed as followings:*

Response: Thanks for the positive feedback and the good suggestion of quantifying laser intensity. In our experiments, we used the same instrument with the same objective lens (magnification of 100 and NA = 0.9) for all Raman measurements. We measured the laser intensity using a power meter and the obtained laser intensities as function of instrument settings was shown in the Responsive Fig. 1 (attached below). According to this function, we have adjusted all the data regarding to laser-power-dependent Raman measurements based on the calculated actual laser intensity exposed by the WS₂ samples.

Responsive Fig. 1. The measured actual power intensity as function of instrumental set output.

Method details about quantification of laser intensity is described in Part II of the supporting information as:

For laser-power-dependent Raman measurements, the exfoliated 2M WS₂ flakes on the Si/SiO₂ or holey Si/SiO₂ or PDMS substrates were measured by setting the laser at specific output powers. The exact laser power exposed by a WS₂ sample was measured by a power meter (Thorlabs), and all the data obtained from laser-power-dependent Raman measurements were based on the actual laser intensity irradiating on the WS₂ samples.

Changes to the manuscript: Details about quantification of laser power are mentioned in the supporting information on Page 4. Data presented in Fig. 1f, Fig. 3e-f, h and Supplementary Fig. 8, 14-16 and the related text are revised according to the actual laser intensity exposed by the WS₂ sample.

Comment 1: Although ML in Fig1a6 seems to be faint, the optical contracts of each image in Figure 1a are different. I would recommend normalizing the optical contrast change of each layer number region as compared to the background region to provide a more quantified result.

Response: Thanks for this good suggestion. ML in Fig. 1a6 is actually disappeared because of air oxidation. We have normalized the optical contrast for all the images in Fig. 1a, and attached the revised Fig. 1a below:

Revised Fig. 1a. Optical images of a piece of exfoliated 2M WS₂ flake with different thicknesses areas at different temperatures in the air. All the scale bars correspond to 2 μm.

Changes to the manuscript: Optical contrast of Fig. 1a is modified.

Comment 2: On Page 6 line 108, do you mean “refraction index” at the place of “infracation index”?

Response: “Infracation index” is changed to “refractive index”.

Changes to the manuscript: This content is mentioned in the manuscript on Page 6.

Comment 3: I found there are missing scale bars on multiple optical microscopy images, AFM images, and PL maps, such as Figure 1c4-5, Figure 2i-m, Supplementary Figure 4, 5, 6, 8, 10, and 13.

Response: We have added scale bars in the mentioned images accordingly.

Comment 4: Are Figure 1c3 and the optical microscopy and AFM images from Supplementary Figure 4, 5, 6, 8, 10, and 13 taken before or after heating? The experiment conditions should be clearly addressed. The supplementary AFM images (Supp Figure 4 1L, 3L, 6L, Supp Figure 5 2L, 3L, 4L, 20L, and Supp Figure 6 3L and 5L) show multiple samples’ surfaces are not smooth. The white dots on the sample surfaces suggest either the samples are oxidized or contaminated or of low quality. Therefore, it is important to specify if the images were taken before or after the heating and the authors should explain why some samples surfaces are not smooth.

Response: All the mentioned images were taken before heating and these messages have been specified in the corresponding revised figure captions. The mentioned AFM images show some WS₂ flakes are not smooth. Since those WS₂ flakes are obtained from mechanical exfoliation, the white dots in the corresponding AFM images can be attributed to adhesive residue from scotch tapes.

Changes to the manuscript: The messages of “at room temperature” are added in the captions of the mentioned figures.

Comment 5:

(i) On page 8, the authors claim that “High-resolution XPS (Supplementary Fig. 7a-b) shows that W 4f peak and valance band edge of 2M WS₂ locate about 1.1 eV and 0.85 eV lower than that of 2H WS₂, respectively, indicating W element in 2M WS₂ has lower oxidation state than that in 2H WS₂. This is believed to be caused by existing extra electrons at 2M WS₂ surface”. It is incorrect to use XPS to identify the change of oxidation state if W atoms from each sample are not in the same chemical environment, in this case, distorted octahedral and trigonal prismatic coordination. Although XPS is a surface technique, a proper measurement could reflect bulk properties unless the samples are not homogeneous. The authors should make more comments to explain why extra electrons exist at the 2M WS₂ surface and how are the sample charge balanced with the extra electrons. What about the inside underneath the surface?

(ii) On page 9, “Nevertheless, the O 1s peak (Supplementary Fig. 7c) barely changes during the 2M to 2H phase transition. These results indicate oxygen molecules speed up phase transition by extracting electrons from 2M WS₂ during heating in the air, without chemically oxidizing the WS₂.” I find this is confusing, does authors mean the O 1s peaks come from oxygen molecules? Why are they not shifting to lower binding energies if they took electrons from 2M WS₂? I also noticed that there is a significant amount of O 1s peaks on Supplementary Fig2. Since XPS experiments are carried out under high vacuum, it is unlikely all the O signals are from the adsorbed O₂ molecules from air. It seems more likely that the surface of the sample is slightly oxidized. In Supplementary Fig.7, the shifting of W peaks is not monotonically as increasing the heating time. The 130 °C 45 min heated sample seems to be closer to 2M WS₂ while the 15 min heated sample contains more 2H phase. This result is inconsistent. I suggest the authors to remove the top surface layers with scotch tape right before loading the sample into the XPS chamber to expose a fresh layer and redo the XPS experiments to avoid potential surface contamination. In addition, I would suggest to avoid using the Ar plasma gun to clean the surface since the energy from it could change the surface structure and lead to inaccurate results in this case. Also, the authors should specify the above XPS experiment conditions and how the peaks are calibrated in the methods section.

Response: (i) We realize that the statement of “extra electrons” is misleading, and it is more appropriate to describe 2M WS₂ as that there is enrichment of electrons at the surface. This has been proved by previous X-ray absorption near edge structure (XANES) measurements, where the edge position of the W L₃-edge XANES spectrum for 2M-WS₂ shifts to lower energy compared with that of 2H WS₂, as shown in the attached Responsive Fig. 2 (*Nat. Mater.*, 2021, 20, 1113–1120). This result has also proved 2M WS₂ has a lower valance state than 2H WS₂ (*Nat. Mater.*, 2021, 20, 1113–1120). Similar result was also observed on 1T' MoS₂ (*Nat. Chem.*, 2018, 10, 638–643). Thus, there would be no doubt to declare W element in 2M WS₂ has lower valance state than that in 2H WS₂ and there is enrichment of electrons at the 2M WS₂ surface. Since 2M WS₂ has no band gap, thermal energy in ambient conditions could excite electron transition from valance band to conducting band. Moreover, [WS₆] slabs are negatively charged in the precursor K_{0.7}WS₂. Extra electrons are possible to be left in 2M WS₂ layers after K atoms are extracted from the interlayer. As 2M WS₂ is metallic conductor, any free electrons can only exist at the surface. These charges can be shielded by external dielectric medium. In our measured XPS results (Revised Supplementary Fig. 7, attached below), shifting of W 4f, S 2p and valance band edge spectra of 2M WS₂ compared to that of 2H WS₂ agrees well with the previously reported XANES measurements (*Nat. Chem.*, 2018, 10, 638–643; *Nat. Mater.*, 2021, 20, 1113–1120), further confirming enrichment of electrons at the surface of 2M WS₂. All the recorded XPS spectra are shifting to higher binding energies during 2M to 2H phase transition, strongly indicating electrons are extracted during the phase change process.

[REDACTED]

Responsive Fig. 2. Normalized W L₃-edge XANES spectra of the 1T' (or 2M)-WS₂ and 2H-WS₂ (*Nat. Mater.*, 2021, 20, 1113–1120).

Revised Supplementary Fig. 7. High-resolution XPS spectra of (a) W 4f-5p, (b) S 2p, (c) valence band and (d) O 1s of 2M, intermediate and 2H phases bulk WS₂. Intermediate phases WS₂ were obtained by heating 2M WS₂ in the air at 130 °C for 15 min, 30 min, and 45 min, respectively. 2H phase WS₂ were obtained by heating 2M WS₂ in the air at 250 °C for 20 min.

Additionally, we thank the reviewer for pointing out that the shifting of W 4f spectra is not monotonically as heating time increases in the previous version Supplementary Fig. 7. We realize that the “15 min” and “45 min” samples are mis-labeled. This error has been corrected in the Revised Supplementary Fig. 7.

(ii) During 2M to 2H phase transition when WS₂ is heated at 130 °C in the air for different time, the measured XPS spectra of W 4f and S 2p (Revised Supplementary Fig. 7, attached above) show no sign of formation of W–O or S–O bonds, and the O 1s spectra (Revised Supplementary Fig. 7, attached above) barely changes. By these results, we can conclude the WS₂ sample is not oxidized by the air during the heating process.

To find out the origin of O 1s XPS peaks, we measured XPS on scotch tape exfoliated 2M WS₂, as suggested by the reviewer. However, the obtained XPS spectrum (red curve of the Responsive Fig. 3a, attached below) shows even higher intensities of O 1s peaks. This is because the substrate (SiO₂/Si wafer) provides significant signals of O 1s spectra, since the mechanical exfoliated WS₂ flakes are too few to fully cover the substrate. We also cleaned the 2M WS₂ sample by Ar⁺ ions beam. We find that WS₂ still shows significantly high intensity O 1s peaks after the cleaning, seen from the blue curve in the Responsive Fig. 3a (attached below). Accordingly, the O 1s peak can't be merely from surface adsorbed H₂O and O₂ molecules.

We speculate there could be H₂O and O₂ intercalated in the interlayer of WS₂ and bonded with S atoms. Our 2M WS₂ samples are prepared by extracting interlayer K elements from the precursor K_{0.7}WS₂ in K₂Cr₂O₇/H₂SO₄ aqueous solution. We have confirmed K elements have been completely removed, since the obtained 2M WS₂ exhibits no signal of K 2p spectrum (Responsive Fig. 3b, attached below). However, it is quite possible that H₂O and air intercalated into the WS₂ interlayer. As shown in the Responsive Fig. 4 (attached below), thermogravimetric analysis of bulk 2M WS₂ powder in N₂ atmosphere shows around 1% weight loss from room temperature to 300 °C, which can be attributed to removing of intercalated foreign molecules. We find intensities of O 1s XPS spectra of 2M WS₂ are significantly decreased after annealing at 200 °C for 1 h and are further weakened after annealing at 300 °C for 1h, as shown in the Responsive Fig. 3c (attached below). These results indicate the O 1s XPS peaks are probably contributed by the intercalated H₂O and O₂. Since the temperature (130 °C) at which we study 2M to 2H phase transition is relatively low, the intercalated foreign molecules can't be completely removed, thus significant O 1s peaks can be observed.

The reviewer mentioned a good question why O 1s peak was not shifting to lower binding energies if oxygen-containing species took electrons from 2M WS₂. It should be noted that XPS was carried out *ex situ* in our experiment. Charge transfer between WS₂ samples and ambient H₂O and O₂ molecules have already been finished during heating process, and there is no charged H₂O or O₂ molecules existing at WS₂ surface as XPS is measured, so it is impossible to observe red-shifted O 1s spectra.

Responsive Fig. 3. (a) XPS surveys of the synthesized bulk 2M WS₂, and the 2M WS₂ after cleaning by Ar⁺ ions, and exfoliated 2M WS₂ flakes on a SiO₂/Si substrate. (b) High-resolution XPS spectra of K 2p of the synthesized bulk 2M WS₂. (c) XPS surveys of the synthesized bulk 2M WS₂, and the 2M WS₂ after annealing in Ar atmosphere for 1h at 200 °C and 300 °C, respectively.

Responsive Fig. 4. Thermogravimetric analysis of the synthesized bulk 2M WS₂ in N₂ atmosphere.

To make our discussion clear and convincing, in the revised manuscript, we firstly introduce the fact that there is enrichment of electrons at the surface of 2M WS₂ by referring to literatures, and then demonstrate O₂ molecules in the air can extract electrons from 2M WS₂ based on the shifting of W 4f, S 2p peaks and valance band edge during heating in the air. Finally, we point out WS₂ is not oxidized by the air during the heating process, because there is no XPS peak attributed to W–O or S–O bond. Discussion about XPS results is changed to:

Previous X-ray absorption fine structure measurements have confirmed enrichment of electrons on 2M or 1T' WS₂ surface^{3,6}. It has also been demonstrated that donation of electrons helps to stabilize 1T' TMDs, while extraction of electrons promotes 1T' to 1H phase transition^{20,21,39-41}. Thus, higher 2M

to 2H (or 1T' to 1H) phase transition temperature measured in the inert atmosphere can be attributed to getting rid of electron acceptors, such as O₂ and H₂O molecules in the air. High-resolution XPS (Supplementary Fig. 7) shows that W 4f, S 2p peaks and valence band edge of 2M WS₂ shift to higher energies after 2M to 2H phase transition is activated by heating in the air, and higher extent of phase transition leads to larger range of shifting. Meanwhile, no peak attributed to W–O or S–O chemical bond can be found in the W 4f and S 2p spectra and O 1s spectra (probably contributed by adsorbed or intercalated H₂O and O₂ molecules) barely change during the 2M to 2H phase transition. These results indicate air speeds up 2M to 2H phase transition during heating processes by extracting electrons from 2M WS₂ without chemically oxidizing it.

Changes to the manuscript: This content is mentioned in the manuscript on Page 8-9.

Comment 6: Figure 2d-f shows the HRSTEM images of the 1T' and the lateral heterostructure intermediate phases. Since the areas shown in Figure 2d-f are very small, I am wondering how many samples and how many areas were studied to reach such a conclusion. Did the authors observe structure changes from 1T' to 1H during the TEM studies since electron beams from TEM carry enough energy to introduce phase transformation as well. The lateral heterostructure shown in Figure 2 could be beam damage.

Response: We never observe structure change from 1T' to 1H during the TEM studies. In fact, electron beams are able to drive 1H to 1T and 1T' change (*Nat. Nanotech.*, 2014, 9, 391-396), so we don't need to worry the 1T' to 1H change observed in TEM is caused by electron beams. In our experiment, we measured TEM using two WS₂ samples with intermediate phase and recorded three regions with varied distances of W-atom lines that can be attributed to 1T'/1H heterostructures. One of them is presented in Fig. 2e, and we attached the other two in the Responsive Fig. 5 below:

Responsive Fig. 5. HRTEM images of intermediate phase WS₂.

Comment 7: Figure 2i and the text about it are confusing. Why did the right part of the flakes disappear after laser punching. What laser power is this? Why is it so destructive? Is it the same power as other measurements? This makes the other Raman results rather concerning.

Response: The right part of the flake is removed after punching with a tungsten probe rather than laser. This is specified in the corresponding figure caption.

Comment 8: Supplementary Fig. 11 seems to have a very low signal to noise ratio. The (11 $\bar{1}$) peaks seem to shift to higher angle before disappearing. Please make more comments on this.

Response: Supplementary Fig. 11 shows zoomed-in view of WS₂ XRD spectra in the range of 31° ~ 43°, and the presented peaks have quite low intensities because most of WS₂ flakes are lying along *bc* plane. The (11 $\bar{1}$) peaks seem to shift to higher angle before disappearing, which can be attributed to in-plane gliding and out-of-plane displacements of W and S atoms. This is discussed in the manuscript as:

High-index XRD diffractions including 2M (11 $\bar{1}$) (002) (31 $\bar{1}$) (601) (311) are also disappearing with (11 $\bar{1}$) diffraction shifting to higher degrees as heating time increases, as shown in Supplementary Fig. 11. This can be attributed to in-plane gliding and out-of-plane displacements of W and S atoms during the 2M to 2H phase transition process.

Changes to the manuscript: This content is mentioned in the manuscript on Page 12.

Comment 9: From the optical microscopy images of Supplementary Fig. 12, the flakes seem to be less homogenous as they transform from 2M to 2H. Can the authors explain this?

Response: We notice that this WS₂ flake contains part areas of additional layer, as labeled by dotted squares in the Responsive Fig. 6 (attached below). Non-homogenous thickness is commonly seen in mechanically exfoliated TMD flakes. Non-homogenous thickness causes non-concerted 2M to 2H phase transition in this WS₂ flake.

Responsive Fig. 6. AFM images and optical images (insets at top-right corners) of a piece of a 6 L WS₂ flake in (a) 2M, (b) an intermediate and (c) 2H phases. Thicker areas are labeled with dotted squares.

Comment 10: Supplementary Fig. 14 a, the Ag5 peak seems to be mislabeled.

Response: Thanks for pointing out this error. We have changed it in the Revised Fig. 14a.

Changes to the manuscript:

Comment 11: On page 20, line 481, “Thes” seems to be a typo.

Response: This typo is revised in the manuscript.

Comment 12: There are multiple experimental details missing in the methods section. Especially, on the Raman measurement, What’s the grating numbers and exposure times, and how many integrations were used? Are they the same throughout the experiments of these studies or the settings are changed with different layer numbers to achieve the best signal to noise ratio. In that case, how does that affect the results of thickness dependent Raman modes studies.

Response: We describe more details about sample preparation, Raman measurements and computational method in the supporting information. For readers' convenience, we add reminders that details can be found in the supporting information to the "Method" part of the revised manuscript.

For the Raman measurements, the grating is 600 grooves per millimeter, and the exposure time is 10s with two times integrations. These settings are kept the same for acquisition of all the Raman spectra. All general settings of Raman measurements are described in Part I of the supporting information as:

Micro-Raman spectra were taken on the 2M WS₂ using a confocal microscope Raman system (WITec, Alpha300R) with optical microscope (Nikon). The measurements were performed with laser excitation wavelength of 532 nm and 100× objective lens (NA = 0.9). The output power of laser is set to 1 mW for acquisitions of all spectra except for the laser-power-dependent Raman measurements. The Raman signals from the sample were introduced to an electron multiplying charge coupled device (CCD) detector (Andor) through a grating with 600 grooves per millimeter. The CCD integration was set to two times with exposure for 10 s in each time for acquisitions of all spectra. The Si peak at 520 cm⁻¹ was used as a reference for calibration of wavenumber.

Changes to the manuscript: This content is mentioned in the supporting information on Page 4.

REVIEWER COMMENTS

Reviewer #3 (Remarks to the Author):

The manuscript has seen notable improvements, and the authors have provided comprehensive responses to the concerns raised by the reviewers and myself, which I appreciate. However, there remain areas that require further attention:

Regarding Comment 3 (Second Point): It is essential to include the AFM height profile. This data is crucial for a complete understanding of the findings.

Regarding Comment 4 (Second Point): While I acknowledge the detailed explanation provided, it does not fully address my concern about the arbitrariness of standardizing the activation energy based on layer number. The observation in Fig. 5a, suggesting that the phase transition barrier remains constant or slightly increases with increasing layer number, appears valid. However, normalizing this barrier per layer still seems to lack a scientific basis. If the phase transition barrier was not dependent on the number of layers and layer-by-layer transition occurred, it implies that the barrier experienced by an individual layer remains independent (or slightly increase) of the layer number. Additionally, my previous concern about the apparent contradiction—the barrier normalizing by layer number leads to a zero phase transition barrier in the bulk limit, but, experimentally, the bulk crystal still has the barrier—remains unaddressed. This issue significantly undermines the rationale for dividing the phase transition barrier by the number of layers. The authors must revisit and substantiate the validity of normalizing the phase transition barrier obtained via DFT by layer number, addressing this contradiction. Furthermore, the argument that an increase in the number of layers—and consequently in heat capacity—would enhance the likelihood of overcoming the activation barrier is not convincing. In a state of thermal equilibrium, the relevance of heat capacity to this process is unclear.

Unless these issues are addressed satisfactorily, I find it challenging to endorse the manuscript for acceptance, despite its improvements and the depth of the discussion provided.

Reviewer #4 (Remarks to the Author):

I appreciate the efforts that the authors spent in address my comments. The manuscript has been significantly improved. I recommend it for publication at this stage.

Point-by-Point Responses to Reviewers' Comments

Referee: 3

Comments: The manuscript has seen notable improvements, and the authors have provided comprehensive responses to the concerns raised by the reviewers and myself, which I appreciate. However, there remain areas that require further attention. Unless these issues are addressed satisfactorily, I find it challenging to endorse the manuscript for acceptance, despite its improvements and the depth of the discussion provided.

Response: Thanks for the positive feedback. We have carefully addressed the remaining issues and details are seen below.

Comment 1: It is essential to include the AFM height profile. This data is crucial for a complete understanding of the findings.

Response: We agree it is important to include AFM height profiles. In the revised manuscript, we present the AFM height profiles of the WS₂ flakes shown in Fig. 1 b1 as Supplementary Fig. 3f (attached below), and we depict the AFM height profiles of other WS₂ flakes in their corresponding AFM images (Fig. 2i-k, Supplementary Fig. 4-6, 8, 10, 12), right at the WS₂ edges together with the annotations of thicknesses.

Revised Supplementary Fig. 3. PL spectra of (a) 2 L (b) 3 L (c) 4 L 2M WS₂ in Fig. 1a heated at different temperatures in the air. (d) Raman and (e) PL spectra of ML 1T' WS₂ in Fig. 1a heated at different temperatures in the air, evidencing the ML 1T' WS₂ is stable before 350 °C in the air. (f) Height profiles of the AFM image shown in Fig. 1 b1.

Changes to the manuscript: Fig. 2i-k and Supplementary Fig. 3-6, 8, 10, 12 and the corresponding captions are modified.

Comment 2: While I acknowledge the detailed explanation provided, it does not fully address my concern about the arbitrariness of standardizing the activation energy based on layer number. The observation in Fig. 5a, suggesting that the phase transition barrier remains constant or slightly increases with increasing layer number, appears valid. However, normalizing this barrier per layer still seems to lack a scientific basis. If the phase transition barrier was not dependent on the number of layers and layer-by-layer transition occurred, it implies that the barrier experienced by an individual layer remains independent (or slightly increase) of the layer number. Additionally, my previous concern about the apparent contradiction—the barrier normalizing by layer number leads to a zero phase transition barrier in the bulk limit, but, experimentally, the bulk crystal still has the barrier—remains unaddressed. This issue significantly undermines the rationale for dividing the phase transition barrier by the number of layers. The authors must revisit and substantiate the validity of normalizing the phase transition barrier obtained via DFT by layer number, addressing this contradiction. Furthermore, the argument that an increase in the number of layers—and consequently in heat capacity—would enhance the likelihood of overcoming the activation barrier is not convincing. In a state of thermal equilibrium, the relevance of heat capacity to this process is unclear.

Response: Thanks for the continued insightful comment on the issue about theoretical calculation. We carefully checked the details of our previous calculating methods and found that the functions of van der Waals (vdW) interactions were not successfully included. Because of the absence of vdW interaction, the individual layers in 2 L and 3 L WS₂ supercells experience nearly the same condition as that of 1 L WS₂ supercell during the simulated phase transitions. At transition state, each of the 1 L and 2 L and 3 L WS₂ supercells has a deformed layer with nearly the same molecular geometry, while other layers in 2 L and 3 L WS₂ supercells almost maintain the 1T' structure. Consequently, the total phase transition barriers of 1 L, 2 L and 3 L WS₂ supercells, i. e., energy differences between the corresponding transition and initial states, are calculated to be quite similar (shown in previous Fig. 5a). In fact, vdW-interaction-absent calculations have also led to exaggerated interlayer spacings. As shown in previous Fig. 4c, interlayer spacings of the initial and final states WS₂ are calculated to be 3.348 Å and 4.082 Å, respectively, which are much larger than the experimentally measured data,

2.492 Å for 2M WS₂ (*Adv. Mater.*, 2019, 31, 1901942) and 3.019 Å for 2H WS₂ (*J. Solid State Chem.*, 1987, 70, 207-209).

We then re-calculate the phase transitions of 2 L and 3 L WS₂ supercells, ensuring vdW interaction is included using the D₃ method of Grimme. The as-calculated interlayer spacings of initial and final states WS₂ decrease to 2.300 Å and 2.927 Å (Revised Fig. 4c, attached below), respectively, which match the previously reported interlayer spacings of 2M and 2H WS₂ (*Adv. Mater.*, 2019, 31, 1901942; *J. Solid State Chem.*, 1987, 70, 207-209). The calculated molecular geometries of 2 L and 3 L WS₂ in intermediate states involve 1T'/1H hetero-structures (Revised Supplementary Fig. 26 and Fig. 4c, attached below), meaning the layer-by-layer phase transition mechanism is still valid. The total phase transition barriers of 2 L and 3 L WS₂ supercells are 1.95 eV and 2.10 eV, respectively, which are distinctly higher than that of 1 L WS₂ (1.83 eV), as shown in the Revised Fig. 5a (attached below). It can be concluded that total phase transition barriers are increasing with the number of layers, rather than maintaining at a same level as described in our previous manuscripts. However, the normalized barrier per layer decreases from 1.83 eV to 0.97 eV and 0.70 eV, as the number of layers increases from 1 L to 2 L and 3 L (Revised Fig. 5b, attached below), exhibiting the same trend as our previously reported.

Revised Supplementary Fig. 26. The molecular geometries of 3 L WS₂ supercell at the initial phase state (coord. I), transition state (coord. IV), final phase state (coord. VII) and other intermediate configurations, viewing from the *a* direction. The highly deformed layer at the transition state is marked by a red dotted square. Layers turned from 1T' to 1H types of structures are marked by blue dotted squares. Lattice edges are depicted in different states, referring to which interlayer dislocations can be seen. Color code: blue and orange spheres represent W and S, respectively.

Revised Fig. 4. Simulation of 2M to 2H phase transition of WS₂ using rectangular supercells. Representative molecular geometries of the 1 L WS₂ supercell in the initial 1T' phase (coord. I), transition state (coord. III), an intermediate configuration (coord. IV) and the final 1H phase (coord. VII), viewing from the (a) *c* direction and (b) *a* direction. Process from coord. I to coord. III corresponds to deformation of 1T'-type lattice and process from coord. IV to coord. VII corresponds to relaxation of 1H-type lattice. (c) Molecular geometries of the 2 L WS₂ supercell at the initial phase state (coord. I), transition state (coord. IV), an intermediate configuration (coord. V) and the final phase state (coord. VII), viewing from the *a* direction. The highly deformed layer at the transition state is marked by a red dotted square. The layer turned from 1T' to 1H type of structures is marked by a blue dotted square. Interlayer spacings are denoted in different states, showing interlayer expansion from coord. I to coord. VII. Lattice edges are depicted in different states, referring to which interlayer dislocation between top and bottom layers can be seen. Color code: blue and orange spheres represent W and S, respectively.

Revised Fig. 5. Energy profiles of 2M to 2H phase transition. (a) Total energies and (b) average energies of 1 L, 2 L and 3 L WS₂ supercells at the initial state (coord. I), transition states (coord. III for 1 L WS₂ and coord. IV for 2 L and 3 L WS₂), final states (coord. VII) and other intermediate configurations. Energies in (b) are normalized by the corresponding numbers of WS₂ layers. E_a^{1L} , E_a^{2L} and E_a^{3L} are total phase transition energy barriers for the 1 L, 2 L and 3 L WS₂ supercells, respectively. \bar{E}_a^{1L} , \bar{E}_a^{2L} and \bar{E}_a^{3L} are average transition barriers per layer for the 1 L, 2 L and 3 L WS₂ supercells, respectively. (c) The average transition barrier per layer as function of the number of layers.

The higher total phase transition barrier for thicker WS₂ is attributed to larger structural difference between the corresponding transition and initial states. Structural parameters of WS₂ layers at the initial 1T' phase state and the transition states in the 1 L, 2 L and 3 L supercells are summarized in Supplementary Table 2 (attached below). It is seen that all the transition state 1 L, 2 L and 3 L WS₂ supercells contain one highly deformed layer, while other layers in 2 L and 3 L supercells are also distorted from the initial 1T' phase. Moreover, structural variation between the deformed layer and the initial 1T' phase lattice gets larger from 1 L to 2 L and to 3 L WS₂. On the one hand, changing of W-atom line distances is getting larger. On the other hand, the deformed layer in 3 L WS₂ have two broken W-S bonds, while deformed layers in 1 L and 2 L WS₂ include only one. However, since the phase transition follows layer-by-layer mechanism, the number of deformed layers doesn't increase from 1 L to 3 L WS₂, so the average transition barrier per layer still decreases with the number of layers.

It should be noted that the average transition barrier per layer decreases more and more slowly as the number of layers increases from 1 L to 2 L and 3 L, seen from the Revised Fig. 5c (attached above), implying the average barriers per layer for thick WS₂ flakes would converge to a certain value, rather than down to zero. This agrees with our experimental result that phase transition temperatures of WS₂ flakes thicker than 5 L converge to 120 °C. To solidify this trend, we tried to run the calculations for 5 L and 10 L WS₂ supercells in the past month. Unfortunately, we didn't get reasonable results from

those calculations because our computing resource is not capable of handling such systems involving too complicated vdW interactions.

Supplementary Table 2. Molecular geometries WS₂ layers at the initial 1T' phase state and the transition states in the 1 L, 2 L and WS₂ supercells. Highly deformed layers at the transition states are marked by red dotted squares.

VdW interaction plays important roles in the phase transition process of multilayered WS₂. On the one hand, vdW-interaction linked layers tend to simultaneously deform during phase transition, resulting in increasing of total transition barrier with the number of layers. On the other hand, the existence of vdW interaction can constrain the freedom of lattice deformation, to minimize the layer stacking energy. Consequently, phase transition of multilayered WS₂ occurs layer(s) by layer(s). For a relatively thick WS₂ (≥ 5 L), transition state lattice might include two or more highly deformed layers

that are widely separated, since vdW interactions between them are small. Once the number of deformed layers existing at the transition state turns to be proportionate to the total number of WS₂ layers, the average phase transition barrier per layer will converge to a constant value for thick and bulk WS₂ flakes. Based on the above results and discussions, the concern that normalizing transition barrier by number of layers can lead to zero barrier for the bulk WS₂ can be dispelled.

Supplementary Fig. 18. Structures of initial state rectangular 1 L (left), and 2 L (middle) and 3 L (right) WS₂ supercells applied in theoretical calculations. 1 L, 2 L and 3 L WS₂ supercells are defined to contain one, two and three 1T' layers, respectively. Color code: blue and orange spheres represent W and S, respectively.

Finally, let us elucidate the reasonability of normalizing total transition barriers by the number of layers. As discussed in our last responsive letter, the as-defined 1 L, 2 L and 3 L WS₂ supercells (Supplementary Fig. 18, attached above) contain one, two and three WS₂ layers, respectively.

Accordingly, each cell of thicker WS₂ involves more W and S atoms. Suppose the 1 L, 2 L and 3 L WS₂ supercells are all heated from room temperature to a certain degree centigrade, total heat taken by 2 L and 3 L WS₂ is double and triple that by 1 L WS₂, respectively. In another word, with temperature raised by certain degrees, heat taken by a WS₂ supercell is proportionate to the number of layers. This is what we mean by mentioning thicker supercells have higher heat capacity in our last responsive letter. As the total phase transition barrier is not proportionately increasing with the number of layers (Revised Fig. 5a, attached above), phase transition of a thicker supercell will be activated at a lower temperature. It is easier to understand this logic by normalizing the total transition barrier by the number of layers, where the smaller average barrier per layer leads to the lower phase transition temperature.

In the revised manuscript, we updated the data about structures of the 2 L and 3 L supercells, regarding Fig. 4c, Supplementary Fig. 21-27 and Supplementary Table 2, according to the newly calculated results, which are discussed as:

The simulations of 2 L and 3 L WS₂ supercells reveal layer-by-layer phase transition mechanism, in good agreement with the XRD and Raman spectra of intermediate phases WS₂ (Fig. 2g-h and 2l-m). For 2 L WS₂, the top layer changes from the initial 1T' structure (coord. I) to a deformed structure at coord. IV, where the transition state is reached, and turns to 1H-type structure at coord. V that then relaxes to coord. VII. The bottom layer is gradually distorted from the initial 1T' structure until coord. IV and changes to 1H structure from coord. VI to VII, as shown in Fig. 4c and Supplementary Fig. 21-22. Similarly, the bottom layer of 3 L WS₂ first transforms to 1H structure, followed by the mid layer and top layer, seen from Supplementary Fig. 23-26. 1T'/1H heterostructures are seen at coord. V for 2 L (Fig. 4c) and 3 L WS₂ (Supplementary Fig. 26). At transition states, both 2 L and 3 L WS₂ contain a highly deformed layer, while other layers are also distorted from the initial 1T' phase, as shown in Supplementary Table 2. The deformed layers in 2 L and 3 L WS₂ have larger structural variations from the initial 1T' lattice compared to that in 1 L WS₂. The two W-atom line distances get closer in transition state 2 L and 3 L WS₂ than 1 L WS₂ (Supplementary Table 2). In addition, the deformed layer in 3 L WS₂ have two broken W-S bonds, while deformed layers in 1 L and 2 L WS₂ include only one (Supplementary Table 2).

Phase transition energy barriers are discussed as:

The total phase transition barrier of a WS₂ supercell is defined as the energy difference between the corresponding transition and initial states. As thicker WS₂ supercell involves larger structural difference between the corresponding transition and initial states, the total phase transition barrier increases from 1.83 eV to 1.95 eV and 2.10 eV, as the number of layers increases from 1 L to 2 L and 3 L, as shown in Fig. 5a. After normalizing the total phase transition barriers by the number of layers, average transition barriers per layer of 1 L, 2 L and 3 L WS₂ supercells are calculated to be 1.83 eV, 0.97 eV and 0.70 eV (Fig. 5b), respectively, exhibiting the average transition barrier decreases with the increase of layer number. This result theoretically confirms thinner 2M WS₂ has higher phase stability. It should be noted that the average transition barrier per layer decreases more and more slowly as the number of layers increases from 1 L to 2 L and 3 L (Fig. 5c), implying the average transition barriers per layer for thick WS₂ flakes would converge to a certain value. This agrees to the fact that phase transition temperatures of WS₂ flakes thicker than 5 L converge to 120 °C (Fig. 1e).

The role of vdW interaction in phase transition of multilayered WS₂ is discussed as:

VdW interaction also plays important roles in the phase transition process of multilayered WS₂. On the one hand, vdW-interaction linked layers tend to simultaneously deform during phase transition, resulting in increasing of total transition barrier with the number of layers. On the other hand, the existence of vdW interaction can constrain the freedom of lattice deformation, to minimize the layer stacking energy. Consequently, phase transition of multilayered WS₂ occurs layer(s) by layer(s). In fact, WS₂ lattices always tend to minimize the vdW-interaction-associated stacking energy by optimizing the stack configuration. Interlayer expansion and dislocation during phase transition have been demonstrated by the simulated molecular geometries (Fig. 4c and Supplementary Fig. 26) and physical characterizations (Fig. 2g-k and Supplementary Fig. 12) of multilayered WS₂. For a relatively thick WS₂ (≥ 5 L), transition state lattice might include two or more highly deformed layers that are widely separated, since vdW interactions between them are small. Once the number of deformed layers existing at the transition state turns to be proportionate to the total number of WS₂ layers, the average phase transition barrier per layer will converge to a constant value for thick and bulk WS₂ flakes.

Additionally, the section of “Abstract”, last paragraphs of “Introduction” and “Discussion” are modified to include the contents about layer(s)-by-layer(s) phase transition and transition energy barrier. We also attached the theoretically calculated files in the supplementary materials in case readers are interested in referring to.

Changes to the manuscript: This content is mentioned in the manuscript on Page 18-22.

Referee: 4

Comments: I appreciate the efforts that the authors spent in addressing my comments. The manuscript has been significantly improved. I recommend it for publication at this stage.

Response: Thanks for the positive response.

REVIEWERS' COMMENTS

Reviewer #3 (Remarks to the Author):

The manuscript is well improved. Now it is ready for publish.